# Provable Causal State Representation under Asynchronous Diffusion Model for POMDPs

## Abstract

A major challenge in applying reinforcement learning (RL) to real-world scenarios is managing high-dimensional, noisy perception input signals. Identifying and utilizing representations that contain sufficient and essential information for decision-making tasks is key to computational efficiency and generalization of RL by reducing bias in decision-making processes. In this paper, we present a new RL framework, named *Causal State Representation under Asynchronous Diffusion Model (CSR-ADM)*, which accommodates and enhances any RL algorithm for partially observable Markov decision processes (POMDPs) with perturbed inputs. A new asynchronous diffusion model is proposed to denoise both reward and observation spaces, and integrated with the bisimulation technology to capture causal state representations in POMDPs. Notably, the causal state is the coarsest partition of the denoised observations. We link the causal state to a causal feature set and provide theoretical guarantees by deriving the upper bound on value function approximation between the noisy observation space and the causal state space, demonstrating equivalence to bisimulation under the Lipschitz assumption. To the best of our knowledge, CSR-ADM is the first framework to approximate causal states with diffusion models, substantiated by a comprehensive theoretical foundation. Extensive experiments on Roboschool tasks show that CSR-ADM outperforms state-of-the-art methods, significantly improving the robustness of existing RL algorithms under varying scales of random noise.

## 1 Introduction

Reinforcement learning (RL), a method for autonomous learning, has demonstrated extensive applications (Schrittwieser et al., 2020; Silver et al., 2017), where an agent learns by interacting with the environment to maximize long-term cumulative rewards through trial and error. However, classical RL methods face challenges when the state of the environment cannot be fully observed. Partially observable Markov decision processes (POMDPs) were introduced to handle the situations with incomplete observations. A major challenge of POMDPs is the robustness of observations against such perturbation on the state space, which may result from sensor errors or mismatches between statistic datasets and the real environment. Enhancing the robustness of the trained RL policy against state perturbations is crucial for improving the interpretability and efficiency of making decisions, leading to a causal representation of states.

Recently, research for causal state representation (CSR) learning has been developed to extract abstract features from perturbed observations. Utilizing these abstract representations rather than the raw data has demonstrated more efficient decision-making capability for Markov decision processes (MDPs) (Lesort et al., 2018) and POMDPs (Zhang et al., 2019). Representative methods along this line include bisimulation-based methods (Zhang et al., 2020), Kalman filters (Zois et al., 2014), ordinary differential equations (ODE)-based recurrent models (Zhao et al., 2024), world models (Ha & Schmidhuber, 2018), a connection between predictive state representations (PSRs) and bisimulation via causal states (Zhang et al., 2019), and others (Lanier et al., 2024; Chen et al., 2023a). However, these methods do not consider perturbations, which limits the deployment of relevant representative algorithms. Therefore, by properly modeling and estimating the underlying transition dynamics and rewards with noise, it is possible to effectively reduce interactions with the environment, for either model-based or model-free RL (Hafner et al., 2019; 2020).

Despite the effectiveness of the above methods, existing state representations for RL tend to output an unimodal distribution over the action space, which is likely trapped in a locally optimal solution with poor performance due to its limited expressiveness of complex distributions. Given that generative models are powerful in learning complicated multimodal distributions, several algorithms with generative models for CSR in POMDPs have emerged, such as deep variational reinforcement learning (Igl et al., 2018) and structured sequential variational auto-encoder (Huang et al., 2022).

However, methods aligned with generative models, such as variational autoencoder typically generate samples by learning the latent representations of data, rather than directly addressing noise, thus their effectiveness in handling noise may be relatively limited. In contrast, the diffusion model (Sohl-Dickstein et al., 2015; Song et al., 2020; Ho et al., 2020) can remove noise better while preserving important features in data by iteratively transforming noisy samples into high-quality real samples. The diffusion model offers a better choice when the simultaneous denoising and preservation of important features are required. Recently, diffusion-based generative models have been increasingly used in decision-making problems as trajectory generators or state representation (Janner et al., 2022; Ajay et al., 2022; Zhihe & Xu, 2023a). Although the diffusion model shows its promising and potential applications to POMDP tasks, previous works have overlooked the causal relationships (e.g., bisimulation). Moreover, it is a matter of deliberation whether it is reasonable to achieve diffusion model-based denoising by the same step. Thus, a natural question arising is:

*How can we apply diffusion models to enhance causal state representation for reducing decision-making biases in perturbed POMDPs?*

## 1.1 Contribution

In this paper, we aim to enhance decision-making in deep reinforcement learning (DRL) for perturbed POMDPs, characterized by partial and noisy observations. We introduce an innovative approach, *Causal State Representation under Asynchronous Diffusion Model (CSR-ADM)*, which is applicable to any RL algorithm. Our contributions are summarized as follows:

**Algorithm Design**: We develop a new causal state representation for perturbed POMDPs to improve DRL decision-making amidst noisy and incomplete observations. This representation extends bisimulation, traditionally applied in MDPs, to POMDPs, facilitating the evaluation of causality in DRL inputs. We also propose a novel diffusion model that characterizes the conditional probability distribution of transition dynamics and rewards under varying noise intensities. This model serves as a criterion for assessing the causality of bisimulation relationships and mitigates observation noise through new adjustable asynchronous forward and backward propagation. Notably, our asynchronous diffusion model is adept at handling disturbances across variables of different scales and can be implemented as a standalone module for effective denoising.

**Theoretical Analysis**: We establish the theoretical guarantees of CSR-ADM in perturbed POMDPs by deriving the upper bound on the value function approximation (VFA) between the noisy observation and the causal state spaces. By assessing the distribution estimation error using the Wasserstein-1 distance for the proposed asynchronous diffusion model, we demonstrate that the model tightens the upper bound on VFA and hence contributes to DRL decision-making for POMDPs.

**Extensive Simulation**: We conduct extensive simulations across six environments under perturbed POMDPs to demonstrate the performance of CSR-ADM. Considering that our approach can accommodate any RL algorithm, we present simulations where CSR-ADM is combined with soft actor-critic (SAC) and compare it against the other four baselines. We also perform ablation studies to investigate the impact of key parameters, i.e., noise intensity and the magnitude of environmental noise. Experimental results show that CSR-ADM enhances RL's decision-making under incomplete and noisy observations and rewards.

## 1.2 Related Work

**Causal state representation**   To enhance the performance of decision-making under perturbed POMDPs, several recent studies have focused on deriving causal state representations for decision-making generalization through the technique of representation learning. For instance, Zhang et al. (2019) proposed an algorithm to approximate causal states in POMDPs. Utilizing domain-invariant causal features, Bica et al. (2021) proposed Invariant Causal Imitation Learning (ICIL) to address

distribution shifts. Additionally, some works (Lee et al., 2019; Menda et al., 2019; Loquercio et al., 2020) proposed ensemble representations that leverage multi-modal sensor inputs to boost generalizability for self-driving agents under uncertainty quantification. The PlanT framework (Renz et al., 2023) serves as a learnable planner module grounded in object-centric representations. Moreover, the realm of RL has witnessed advancements in state representation through self-supervised learning approaches, including hierarchical skill decomposition (Akrour et al., 2018), time-contrastive learning (Sermanet et al., 2018), and deep bisimulation metric learning (Zhang et al., 2020; Dadashi et al., 2021). However, there is a lack of consideration of perturbation-based causal state representations.

**RL with diffusion model**   The diffusion model was originally proposed as an generative model for image generation (Sohl-Dickstein et al., 2015; Ho et al., 2020). Recently, it has been adopted in decision-making for state-based tasks, especially for perturbed states. In RL, diffusion models can be utilized not only for direct decision-making (Ajay et al., 2022; Janner et al., 2022; Zhihe & Xu, 2023a; Wang et al., 2022; Zhang et al., 2024; Li et al., 2023) but also for effective denoising and distribution estimation. For instance, DMBP (Zhihe & Xu, 2023a) utilizes the diffusion model as a denoiser (against state observation perturbations) rather than a generator, for robust training of RL agents. The DIPO (Yang et al., 2023) utilizes the diffusion model to address the denoising problem in model-free RL. Moreover, Fu et al. (2024) presented a sharp statistical theory of distribution estimation using a conditional diffusion model. However, the current studies do not differentiate whether data used for training contains noise or not, hence limiting the effectiveness of denoising.

## 2   PROBLEM FORMULATION

**RL in POMDP**   For RL, some environments are generally modeled as POMDPs in the form of $\mathcal{M} = (\mathcal{S}, \mathcal{A}, \mathcal{O}, \gamma, F, G, H)$, where $\gamma$ is the discount factor. Assume a sequence of samples $\{\langle \mathbf{o}_t, \mathbf{a}_t, r_t \rangle\}_{t=1}^{T}$, where $\mathbf{o}_t \in \mathcal{O}$ represents the sensory signal (e.g., high-dimensional images) at time $t$ with $\mathcal{O}$ indicting to the observation space. $\mathbf{a}_t \in \mathcal{A}$ represents the action chosen at time $t$ with action space $\mathcal{A}$, and $r_t \in [0, 1]$ denotes the reward. We use $\mathbf{s}_t = \{s_{1,t}, s_{2,t}, \cdots, s_{d,t}\} \in \mathcal{S}$ to denote the $d$-dimensional true state, where $\mathcal{S}$ is the state space with $d$ dimensions. Therefore, we can describe the environment model as follows:

$$\mathbf{o}_t = F(\mathbf{s}_t, \mathbf{e}_t) \iff P(\mathbf{o}_t \mid \mathbf{s}_t), \tag{1a}$$

$$r_t = G(\mathbf{s}_{t-1}, \mathbf{a}_{t-1}, \varepsilon_t) \iff P(r_t \mid \mathbf{s}_{t-1}, \mathbf{a}_{t-1}), \tag{1b}$$

$$\mathbf{s}_t = H(\mathbf{s}_{t-1}, \mathbf{a}_{t-1}, \eta_t) \iff P(\mathbf{s}_t \mid \mathbf{s}_{t-1}, \mathbf{a}_{t-1}), \tag{1c}$$

where $F$, $G$, and $H$ represent the observation function, reward function, and transition function, respectively; $\mathbf{e}_t$, $\varepsilon_t$, and $\eta_t$ are the associated independent and identically distributed (i.i.d.) random noises. The POMDP consists of states $\mathbf{s}_t$. Given $\mathbf{a}_{t-1}$ and $\mathbf{s}_{t-1}$, $\mathbf{s}_t$ is independent of the states and actions that occurred before time $t-1$. Additionally, the action $\mathbf{a}_{t-1}$ directly affects the state $\mathbf{s}_t$, rather than the observation signal $\mathbf{o}_t$. The reward is also influenced by both the state and action. In particular, the observation signal $\mathbf{o}_t$ is generated from a base state corrupted by random noises. We consider noise $\epsilon_t$ in the reward function to capture noise, e.g., measurement errors.

**Causal state representation and bisimulation**   There exist structural relationships among different dimensions of $\mathbf{s}_t$, so that action $\mathbf{a}_{t-1}$ may not affect all dimensions of $\mathbf{s}_t$ and reward $r_t$ may be unaffected by all dimensions of $\mathbf{s}_{t-1}$. As illustrated in Figure 1, we take $d = 3$ as an example, i.e., $\mathbf{s}_t = [s_{1,t}, s_{2,t}, s_{3,t}]^{\mathrm{T}}$. State $s_{3,t-1}$ affects $s_{2,t}$, but there is no connection between $\mathbf{a}_{t-1}$ and $s_{3,2}$. Only $s_{2,t-1}$ has an edge toward $r_t$.

Causal state representation has been explored as a method to differentiate pertinent information from irrelevant details (Li et al., 2006), aiming to generate a more compact representation that facilitates decision-making and planning. As a type of causal state representation, states and observations are considered bisimilar if they yield the same expected reward and have equivalent distributions over subsequent bisimilar states and observations (Givan et al., 2003). To this end, we assert that they exhibit a bisimulation relationship, providing a mathematically rigorous definition of how two environments can yield the same outcome. Based on the environment's dynamics $P(\mathbf{s}_{t+1}, r_{t+1}|\mathbf{s}_t, \mathbf{a}_t)$, the similarity between environments can be expressed by the similarity between their state transition and reward functions. Following (Castro et al., 2009), we define the equivalence in POMDP as:

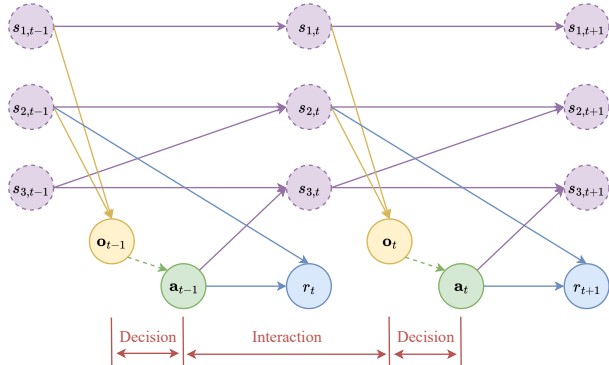

Figure 1: System Model Diagram: Taking $d = 3$ as an example, solid-line circular nodes represent observed variables, while dashed-line circular nodes represent unobserved variables; solid lines represent causal relationships, while dashed lines represent decision dependencies.

**Definition 1 (Causal state representation under bisimulation)** *Given a POMDP* $\mathcal{M} = (\mathcal{S}, \mathcal{A}, \mathcal{O}, F, G, H)$ *and the function of the state space into observation space* $F : \mathcal{S} \to \mathcal{O}$, *any pair of state and observation* $\{\mathbf{s}_t \in \mathcal{S}, \mathbf{o}_t \in \mathcal{O}\}$ *is F-trajectory equivalent if and only if*

- *For any* $\mathbf{a} \in \mathcal{A}$, $P\left(r_{t+1} \mid \mathbf{s}_t, \mathbf{a}\right) = P\left(r_{t+1} \mid F^{-1}(\mathbf{o}_t), \mathbf{a}\right)$,

- *For any* $\mathbf{a} \in \mathcal{A}$, $P\left(\mathbf{s}_{t+1} \mid \mathbf{s}_t, \mathbf{a}\right) = P\left(\mathbf{s}_{t+1} \mid F^{-1}\left(\mathbf{o}_t\right), \mathbf{a}\right)$.

**Goal** By denoising states and rewards, estimating environment dynamics, and extracting causal states, we aim to represent causal states under perturbed POMDP. We also wish to design a diffusion model considering the differentiation of noise intensity within data.

## 3 ALGORITHM DESIGN

In this section, we propose the *Causal State Representation under Asynchronous Diffusion Model (CSR-ADM)* framework to achieve effective causal state representation. Specifically, we design an asynchronous diffusion model to simultaneously denoise the states and rewards through the environment dynamics estimation. Additionally, we learn an approximate causal state representation based on bisimulation. Here, we present the procedure for CSR-ADM training in Algorithm 1. A diagram of the proposed approach is shown in Figure 4 in Appendix A. As a causal state presentation framework, CSR-ADM can be adapted to any RL algorithm.

---

**Algorithm 1:** Hybrid asynchronous diffusion model and bisimulation guided RL (CSR-ADM)

1 **Parameter:** Discount factor $\gamma$, forward stepsize $K$, and noise intensity $\delta$;
2 **Initialize:** Observation denoise model $\theta$, reward denoise model $\phi$, bisimulation model $\zeta$, start observation $\mathbf{o}_1$, and empty replay memory $\mathcal{D}$;
3 **for** *Episode* $t = 1, \ldots, T$ **do**
4     Compute the (approximate) denoised causal state $\hat{\mathbf{s}}_t$ from $\mathbf{o}_t$ using observation denoise model $\theta$ and bisimulation model $\zeta$;
5     Select action $\mathbf{a}_t \sim \pi(\hat{\mathbf{s}}_t)$, and obtain reward $r_{t+1}$ and new observation $\mathbf{o}_{t+1}$;
6     Store transition $(\mathbf{o}_t, \mathbf{a}_t, r_{t+1}, \mathbf{o}_{t+1})$ in replay memory $\mathcal{D}$;
7     Sample a batch of transitions randomly from $\mathcal{D}$ as $\mathcal{B}$;
8     Obtain states $\hat{\mathbf{s}}_t$ and $\hat{\mathbf{s}}_{t+1}$ from observations $\mathbf{o}_t$ and $\mathbf{o}_{t+1}$ in $\mathcal{B}$, respectively;
9     Take gradient descent on $\hat{\mathcal{L}}_{\text{State}}(\theta) + \hat{\mathcal{L}}_{\text{BiState}}(\zeta)$;
10    Take gradient descent on $\hat{\mathcal{L}}_{\text{Rew}}(\phi) + \hat{\mathcal{L}}_{\text{BiRew}}(\zeta)$;

**Output:** Policy $\pi$

---

**Asynchronous diffusion model** The objective of the asynchronous diffusion model is to derive $P\left(\hat{\mathbf{s}}_{t+1} \mid \hat{\mathbf{s}}_t, \mathbf{a}_t\right)$ and $P\left(\widehat{r}_{t+1} \mid \hat{\mathbf{s}}_t, \mathbf{a}_t\right)$ from perturbed sample $(\mathbf{o}_t, \mathbf{a}_t, r_{t+1}, \mathbf{o}_{t+1})$, where $\hat{\mathbf{s}}_t$ and $\hat{\mathbf{s}}_{t+1}$ denote the causal states estimated under denoised observations, and $\widehat{r}_{t+1}$ represents the denoised reward at time $t + 1$. Existing diffusion model-based RL algorithms typically use $\mathbf{o}_{t+1}$ and $r_{t+1}$ as the initial training data (Zhihe & Xu, 2023b), implying that the distribution fitted by the diffusion model is affected by noise. Consequently, there is a gap for improvement in the existing diffusion model-based denoising algorithms.

Considering the differentiation of noise intensity, we design an asynchronous diffusion model to achieve effective denoising of states and rewards and estimate environmental dynamics, assuming that $\mathbf{o}_{t+1}$ and $r_{t+1}$ are superimposed by $\delta$-steps Gaussian noise. For clarity, we use $t$ to indicate the RL iteration and $k$ to indicate the diffusion model's step. To obtain the denoised causal state $\hat{\mathbf{s}}_{t+1}$, we use $r_{t+1}$ and $\tilde{\mathbf{s}}_{t+1} = \zeta(\mathbf{o}_{t+1})$ as part of the inputs to the asynchronous diffusion model, along with $\hat{\mathbf{s}}_t$ and $\mathbf{o}_t$, where $\tilde{\mathbf{s}}_{t+1}$ represents the causal state with noise. Given the above assumption, we denote these inputs as $\mathbf{x}^\delta$, corresponding to the results after a $\delta$-step forward process. Considering the initial conditional distribution as $P(\mathbf{x}^\delta|\hat{\mathbf{s}}_t, \mathbf{a}_t)$, we proceed to analyze the denoised conditional distribution $P(\mathbf{x}^0|\hat{\mathbf{s}}_t, \mathbf{a}_t)$.

For each asynchronous diffusion model update, we consider adding noise progressively, which is represented by a forward Ornstein–Uhlenbeck (OU) process, as follows:

$$d\mathbf{x}^k = -0.5\mathbf{x}^k dk + d\mathbf{w}^k \quad \text{with} \quad \mathbf{x}^\delta \sim P(\mathbf{x}^\delta|\hat{\mathbf{s}}_t, \mathbf{a}_t) \quad \text{for} \quad k \geq \delta; \tag{2}$$

$$d\mathbf{x}^k = -0.5\mathbf{x}^k dk + d\mathbf{w}^k \quad \text{with} \quad \mathbf{x}^0 = \left(\mathbf{x}^\delta - \sqrt{1 - \overline{\alpha}_\delta}\epsilon\right)/\sqrt{\overline{\alpha}_\delta} \quad \text{for} \quad k \geq 0, \tag{3}$$

where $\mathbf{w}^k$ is a Wiener process, $\beta_1, \beta_2, \cdots, \beta_K$ provide a predefined variance schedule, $\alpha_j = 1 - \beta_j$, $\overline{\alpha}_i = \prod_{j=0}^{i} \alpha_j$, and $\epsilon$ follows a standard normal distribution. In the infinite-time limit, $\mathbf{x}^\infty$ follows a standard Gaussian distribution. At any finite step $k$, we denote $P(\mathbf{x}^k|\hat{\mathbf{s}}_t, \mathbf{a}_t)$ as the marginal conditional distribution of each result $\mathbf{x}^k$ produced by the forward process conditioned on the denoised causal states and actions.

The forward process terminates at a sufficiently large step $K$ and the reverse process is defined to generate samples by reversing results per step in (2) as

$$d\overline{\mathbf{x}}^k = \left[0.5\overline{\mathbf{x}}^k + \nabla \log p(\overline{\mathbf{x}}^k|\hat{\mathbf{s}}_t, \mathbf{a}_t)\right] dk + d\overline{\mathbf{w}}^k \quad \text{with} \quad \overline{\mathbf{x}}^0 \sim P(\mathbf{x}^K|\hat{\mathbf{s}}_t, \mathbf{a}_t), \tag{4}$$

where $\overline{\mathbf{w}}^k$ and $\overline{\mathbf{x}}^k$ is a time-reversed Wiener and reverse process, respectively. $\nabla \log p(\overline{\mathbf{x}}^k|\hat{\mathbf{s}}_t, \mathbf{a}_t)$ is the unknown conditional score function and needs to be estimated utilizing conditional score networks. We refer to $\hat{\varphi}(\mathbf{x}, \hat{\mathbf{s}}_t, \mathbf{a}_t, t)$ as such the estimator of the conditional score $\nabla \log p(\mathbf{x}^k|\hat{\mathbf{s}}_t, \mathbf{a}_t)$.

According to classifier-free guidance, a widely adopted method for training $\hat{\varphi}$ proposed by Ho et al. (2020), we obtain the loss function for our asynchronous diffusion model, as given by

$$\ell(\mathbf{x}, \hat{\mathbf{s}}_t, \mathbf{a}_t; \varphi) = \int_{k_0}^{K} \frac{1}{K - k_0} \mathbb{E}_{\tau, \mathbf{x}^k \sim N(\sqrt{\alpha_k}\hat{\mathbf{x}}^0, \sigma_k^2 I)} \left[\left\|\varphi(\mathbf{x}^k, \tau(\hat{\mathbf{s}}_t, \mathbf{a}_t), k) - \nabla_{\mathbf{x}^k} \log p(\mathbf{x}^k|\mathbf{x})\right\|_2^2\right] dk$$

$$+ \int_{\delta}^{K} \frac{1}{K - \delta} \mathbb{E}_{\tau, \mathbf{x}^k \sim N(\sqrt{\alpha_k}\mathbf{x}, \sigma_k^2 I)} \left[\left\|\varphi(\mathbf{x}^k, \tau(\hat{\mathbf{s}}_t, \mathbf{a}_t), k) - \nabla_{\mathbf{x}^k} \log p(\mathbf{x}^k|\mathbf{x})\right\|_2^2\right] dk, \tag{5}$$

where $p(\mathbf{x}^k|\mathbf{x})$ is the Gaussian transition kernel of the forward process (2), i.e., $\nabla \log p(\mathbf{x}^k|\mathbf{x}^0) = -(\mathbf{x}^k - \sqrt{\alpha_k}\mathbf{x}_0)/\sigma_k^2$. Let $\tau \sim \text{Unif}\{\emptyset, \text{id}\}$ be a mask signal, where $\emptyset$ means that we ignore the guidance $(\hat{\mathbf{s}}_t, \mathbf{a}_t)$ and $\text{id}$ denotes otherwise. We consider the uniform distribution on $\tau$, which means $\mathbb{P}(\tau = \emptyset) = \mathbb{P}(\tau = \text{id}) = 0.5$. Moreover, we consider an early-stopping step $k_0$ similar to Nichol & Dhariwal (2021), in order to prevent the blow-up of score functions.

Recall the assumption of adequate mask signal $\tau$ and sampling on $\mathbf{x}^k$ in (5). Consequently, the classifier-free guidance aims to minimize the empirical risk as follows:

$$\arg\min_{\varphi} \hat{\mathcal{L}}(\varphi) = \mathbb{E}_{\{\mathbf{o}_t, \mathbf{a}_t, r_t, \mathbf{o}_{t+1}\}} [\ell(\mathbf{x}_i, \hat{\mathbf{s}}_i, \mathbf{a}_i; \varphi)] = \frac{1}{n} \sum_{i}^{n} [\ell(\mathbf{x}_i, \hat{\mathbf{s}}_i, \mathbf{a}_i; \varphi)], \tag{6}$$

with $n$ being the sample size. By substituting $\hat{\mathbf{s}}_{t+1}$ and $\theta$ (resp. $r_{t+1}$ and $\phi$) for $\mathbf{x}$ and $\varphi$ in (6), we can similarly obtain the training objective for the states and rewards, respectively, as follows:

$$\hat{\mathcal{L}}_{\text{State}}(\theta) = \mathbb{E}_{\{\mathbf{o}_t, \mathbf{a}_t, r_t, \mathbf{o}_{t+1}\}} [\ell(\zeta(\mathbf{o}_{t+1}), \zeta(\mathbf{o}_t), \mathbf{a}_t; \theta)]; \tag{7}$$

$$\hat{\mathcal{L}}_{\text{Rew}}(\phi) = \mathbb{E}_{\{\mathbf{o}_t, \mathbf{a}_t, r_t, \mathbf{o}_{t+1}\}} [\ell(r_{t+1}, \zeta(\mathbf{o}_t), \mathbf{a}_t; \phi)]. \tag{8}$$

**Bisimulation** We extend the concept of bisimulation to POMDPs to achieve effective causal state representation, specifically estimating $P(\hat{\mathbf{s}}_t \mid \mathbf{o}_t)$. Based on the Wasserstein metric (see Appendix B.1.1), a new bisimulation metric is of particular relevance, as defined below:

**Definition 2 (Bisimulation metric)** *Given $f$, $g$, and constant $c_{\mathrm{R}}, c_{\mathrm{T}} \in (0,1)$ for POMDPs, for any pair of state and observation $\{\mathbf{s}_t \in \mathcal{S}, \mathbf{o}_t \in \mathcal{O}\}$, the following metric exists and is unique,*

$$d\left(\mathbf{s}_t, F^{-1}(\mathbf{o}_t)\right) := \max_{\mathbf{a} \in \mathcal{A}}(c_{\mathrm{R}} W_p(d)\left(P\left(r \mid \mathbf{s}_t, \mathbf{a}\right), P\left(r \mid F^{-1}(\mathbf{o}_t), \mathbf{a}\right)\right)$$
$$+ c_{\mathrm{T}} W_p(d)\left(P\left(\mathbf{s}' \mid \mathbf{s}_t, \mathbf{a}\right), P\left(\mathbf{s}' \mid F^{-1}(\mathbf{o}_t), \mathbf{a}\right)\right)), \tag{9}$$

*where $W_p$ denotes the Wasserstein distance between probability distributions.*

A distance of zero for a given pair indicates bisimilarity. We employ a recurrent neural network (RNN) to fit $P(\hat{\mathbf{s}}_t \mid \mathbf{o}_t)$, i.e., $\hat{\mathbf{s}}_t = \zeta(\mathbf{o}_t)$. When the diffusion model accurately predicts the future observations, $\hat{\mathbf{s}}_t$ serves as a sufficient statistic for the latent variables. In practice, we use empirical implementations to estimate the state representation minimizing the objective loss:

$$\hat{\mathcal{L}}_{\text{BiState}}(\zeta) = \frac{1}{2} \mathbb{E}_{\{\mathbf{o}_t, \mathbf{a}_t, r_t, \mathbf{o}_{t+1}\}} \left[ W_d\left(P\left(\hat{\mathbf{s}}_{t+1} \mid \hat{\mathbf{s}}_t, \mathbf{a}_t\right), \theta\left(\zeta\left(\mathbf{o}_t\right), \mathbf{a}_t\right)\right)\right] \tag{10a}$$

$$\hat{\mathcal{L}}_{\text{BiRew}}(\zeta) = \frac{1}{2} \mathbb{E}_{\{\mathbf{o}_t, \mathbf{a}_t, r_t, \mathbf{o}_{t+1}\}} \left[ W_d\left(P\left(r_{t+1} \mid \hat{\mathbf{s}}_t, \mathbf{a}_t\right), \phi\left(\zeta\left(\mathbf{o}_t\right), \mathbf{a}_t\right)\right)\right]. \tag{10b}$$

Consequently, we implement causal state representation and assist reinforcement learning decisions, by iteratively optimizing $\hat{\mathcal{L}}_{\text{State}}(\theta) + \hat{\mathcal{L}}_{\text{BiState}}(\zeta)$ and $\hat{\mathcal{L}}_{\text{Rew}}(\phi) + \hat{\mathcal{L}}_{\text{BiRew}}(\zeta)$.

## 4 THEORETICAL GUARANTEES

We proceed to bound the value function difference between any pairs of observations and states under causal state representation when the proposed asynchronous diffusion model is employed. We start with some assumptions. Let $\delta$ denote the noise intensity. We mathematically reformulate the assumption considering the noise intensity of the input data, as follows:

**Assumption 1** *The sampled distribution $p_{\text{data}}$ is the result of the noiseless distribution $p(\mathbf{x}_{\text{true}}|\hat{\mathbf{s}}_t, \mathbf{a}_t)$ after $\delta$ steps of the forward process, i.e.,*

$$p_{\text{data}}(\mathbf{x}^\delta|\hat{\mathbf{s}}_t, \mathbf{a}_t) = \int_{\mathbb{R}^d} p(\mathbf{x}_{\text{true}}|\hat{\mathbf{s}}_t, \mathbf{a}_t) \frac{1}{\sigma_\delta^d (2\pi)^{d/2}} \exp\left(-\frac{\left\|\sqrt{\alpha_\delta}\mathbf{x}_{\text{true}} - \mathbf{x}^\delta\right\|^2}{2\sigma_\delta^2}\right) d\mathbf{x}_{\text{true}}. \tag{11}$$

We further introduce a mild tail condition on the initial conditional data distribution as Assumption 2, which pertains solely to the regularity of the original data distribution and does not place constraints on the resulting conditional score function. In other words, we assume an additional bounded Hölder norm condition (see Appendix B.1.2 for details) on true data distribution, as follows:

**Assumption 2** *Let $C_2$ and $C$ be two positive constants. For a fixed radius $B$, define the function $f \in \mathcal{H}^b(\mathbb{R}^d \times [0,1]^{d_y}, B)$. We assume $f(\mathbf{x}_{\text{true}}, \hat{\mathbf{s}}_t, \mathbf{a}_t) \geq C$ for all $(\mathbf{x}_{\text{true}}, \hat{\mathbf{s}}_t, \mathbf{a}_t)$ and the true conditional density function $p(\mathbf{x}_{\text{true}}|\hat{\mathbf{s}}_t, \mathbf{a}_t) = \exp(-C_2 \|\mathbf{x}_{\text{true}}\|_2^2 / 2) \cdot f(\mathbf{x}_{\text{true}}, \hat{\mathbf{s}}_t, \mathbf{a}_t).$*

Since a provable tight relationship implies theoretical guarantees in VFA, a key characteristic of bisimulation metrics is their connection to value functions. To generalize the VFA bound, we assume the existence and uniqueness of $p$-Wasserstein bisimulation metric for any pair of states to measure their similarity.

**Assumption 3 ($p$-Wasserstein bisimulation metric)** *For any given $c_{\mathrm{R}}, c_{\mathrm{T}} \in (0,1)$, $c_{\mathrm{R}} + c_{\mathrm{T}} < 1$, $\forall (\mathbf{s}_i, \mathbf{s}_j) \in \mathcal{S} \times \mathcal{S}$, and $p \geq 1$, we assume that the bisimulation metric in (12) exists and is unique:*

$$d\left(\mathbf{s}_i, \mathbf{s}_j\right) := \max_{\mathbf{a} \in \mathcal{A}}\left(c_{\mathrm{R}} W_p(d)(P(r \mid \mathbf{s}_i, \mathbf{a}), P(r \mid \mathbf{s}_j, \mathbf{a})) + c_{\mathrm{T}} W_p(d)(P(\mathbf{s}' \mid \mathbf{s}_i, \mathbf{a}), P(\mathbf{s}' \mid \mathbf{s}_j, \mathbf{a}))\right). \tag{12}$$

Notably, Assumption 3 does not restrict the state, action, or observation spaces to be finite (or any other conditions). Under Assumptions 1–3, we analyze the theoretical guarantee of CSR-ADM

under POMDPs. The analysis is divided into four steps, including (i) establishing the upper bound of VFA for causal states overlooking observations; (ii) refining the upper bound to the observations and causal states under any model approximations; (iii) analyzing the model approximation under a specific model, i.e., the asynchronous diffusion model; and (iv) combining the results in (ii) and (iii) and deriving the upper bound of VFA under the asynchronous diffusion model.

**Step 1: $p$-Wasserstein value difference bound for any pairs of states** Similar to the bounds developed in previous work (Castro, 2020; Ferns et al., 2011) for policy-independent bisimulation metrics, the following bound holds for on-policy bisimulation metrics: $|V^\pi(\mathbf{s}_i) - V^\pi(\mathbf{s}_j)| \leq d(\mathbf{s}_i, \mathbf{s}_j)$ with $d(\mathbf{s}_i, \mathbf{s}_j)$ defined in Assumption 3, where $V^\pi(\mathbf{s}) = \mathbb{E}_\pi[\sum_{i=0}^\infty \gamma^t r_{t+i+1}|s_t = s]$. With the proof provided in Appendix B.2.1, we can establish the value difference bound as follows.

**Theorem 1 ($p$-Wasserstein value difference bound)** *For the on-policy bisimulation metric defined in (12), given any $c_\mathrm{T} \in [\gamma, 1)$, $c_\mathrm{R} \in (0,1)$, $c_\mathrm{R} + c_\mathrm{T} < 1$, and $p \geq 1$, the bisimulation distance between two states provides the upper bound on the discrepancy in their values:*

$$c_\mathrm{R}|V^\pi(\mathbf{s}_i) - V^\pi(\mathbf{s}_j)| \ \leq \ d(\mathbf{s}_i, \mathbf{s}_j), \ \forall(\mathbf{s}_i, \mathbf{s}_j) \in \mathcal{S} \times \mathcal{S}. \tag{13}$$

In this sense, the bisimulation metric in (12) represents the upper bound of the value gap.

**Step 2: Value difference bound for any pairs of observation and state** Consider $p = 1$ for our analysis. We demonstrate the validity of Assumption 3 in Remark 1, with the proof provided in Appendix B.3.1. More general cases will be proved in our future research.

**Remark 1** *If both the policy and the environment are deterministic or $p = 1$, Assumption 3 holds.*

Recall the definitions of reward function and transition function are independent of $\zeta$ and $\theta$. We consider the influence of the model errors on the value function with the optimal policy-dependent bisimulation distance, as summarized in Theorem 2 with the proof provided in Appendix B.2.2.

**Theorem 2 (Value difference bound with model errors)** *Let the reward function be bounded as $r \in [0,1]$ and $\zeta : \hat{\mathcal{S}} \to \mathcal{S}$ a function mapping estimated states (i.e., denoised observations) to causal states such that $\zeta(\hat{\mathbf{s}}_i) = \zeta(\hat{\mathbf{s}}_j)$ is equivalent to $\widehat{d}_\zeta(\hat{\mathbf{s}}_i, \hat{\mathbf{s}}_j) = \|\zeta(\hat{\mathbf{s}}_i) - \zeta(\hat{\mathbf{s}}_j)\|_q \leq 2\,\widehat{\epsilon}$. For $c_\mathrm{R} \in (0,1)$, $c_\mathrm{T} \in [\gamma, 1)$, $c_\mathrm{R} + c_\mathrm{T} < 1$, and $p = 1$, then:*

$$|V^\pi(\mathbf{s}) - \widetilde{V}^\pi(\zeta(\hat{\mathbf{s}}))| \leq \frac{1}{c_\mathrm{R}(1-\gamma)} \left( 2\,\widehat{\epsilon} + \mathcal{E}_\zeta + \frac{2c_\mathrm{R}}{1 - c_\mathrm{T} - c_\mathrm{R}} \mathcal{E}_\phi + \frac{2c_\mathrm{T}}{1 - c_\mathrm{T} - c_\mathrm{R}} \mathcal{E}_\theta \right), \forall \mathbf{s} \in \mathcal{S}, \tag{14}$$

*where $\mathcal{E}_\zeta := \left\| \widehat{d}_\zeta - \widehat{d} \right\|_\infty$ is the bisimulation metric learning error, $\mathcal{E}_\phi := W_1(d)\left(P\left(r \mid \mathbf{s}, \mathbf{a}\right), P\left(r \mid \hat{\mathbf{s}}, \mathbf{a}\right)\right)$ is the reward approximation error, and $\mathcal{E}_\theta := W_1(d)\left(P\left(\mathbf{s}' \mid \mathbf{s}, \mathbf{a}\right), P\left(\mathbf{s}' \mid \hat{\mathbf{s}}, \mathbf{a}\right)\right)$ is the state transition model error.*

By Theorem 2, we can quantify the upper bound of the value gap under arbitrary model errors. This can be extended to different probability density estimation models to establish specific convergence properties. The theorem facilitates analyzing the impact of the proposed asynchronous diffusion model on the value gap.

**Step 3: Distribution estimation under asynchronous diffusion model** Since $\mathcal{E}_\theta$ and $\mathcal{E}_\phi$ are based on the same asynchronous diffusion model architecture, we define the approximation error of the conditional probability as $\varphi(\mathbf{x}^k, \hat{\mathbf{s}}_t, \mathbf{a}_t, k)$, where $\mathbf{x}^k$ can be replaced by either $\hat{\mathbf{s}}_{t+1}^k$ or $r_{t+1}^k$. Under Assumption 2, we can measure the asynchronous diffusion model's distribution estimation by considering the initialization error, score estimation error, and discretization error, and provide the sample complexity bounds for each of the three errors using the Wasserstein-1 distance. We present the approximation theory for estimating the conditional score utilizing ReLU neural networks as the subsequent theorem, with its proof provided in Appendix B.2.3.

**Theorem 3 (Approximation error by asynchronous diffusion model)** *Under Assumptions 1 and 2, for any fixed $(\mathbf{s}^\star, \mathbf{a}^\star)$, the terminal step $K = \frac{2b}{2d_s + d_a + 2b} \log n$, and the early-stopping step $k_0 =$*

$n^{-\frac{4b}{2d_s+d_a+2b}-1}$, *the estimated error of the conditional probability of noiseless data is given by*

$$\mathbb{E}_{\{\mathbf{x}_t,\hat{\mathbf{s}}_t,\mathbf{a}_t\}_{t=1}^n}\Big[W_1(p(\mathbf{x}_t|\hat{\mathbf{s}}_t,\mathbf{a}_t),\hat{p}(\mathbf{x}_t^{k_0}|\hat{\mathbf{s}}_t,\mathbf{a}_t))\Big]=\mathcal{T}(\mathbf{s}^\star,\mathbf{a}^\star)O\left(n^{-\frac{b}{2d_s+d_a+2b}}(\log n)^{\max(19/2,(b+2)/2)}\right),$$

*where b is the degree of smoothness in Hölder norm; $d_s$ and $d_a$ represent the dimensions of state and action, respectively; $\mathcal{T}(\mathbf{s}^\star,\mathbf{a}^\star)$ is distribution coefficient.*

As $n \to \infty$, the distribution estimation measured by Wasserstein-1 distance converges, i.e., $\mathbb{E}_{\{\mathbf{x}_t,\hat{\mathbf{s}}_t,\mathbf{a}_t\}_{t=1}^n}\Big[W_1(p(\mathbf{x}_t|\hat{\mathbf{s}}_t,\mathbf{a}_t),\hat{p}(\mathbf{x}_t^{k_0}|\hat{\mathbf{s}}_t,\mathbf{a}_t))\Big] \to 0$, corroborating the effective distribution estimation capability offered by the proposed asynchronous diffusion model.

**Step 4: Wasserstein value difference bound under asynchronous diffusion model** The bisimulation metric learning can be achieved by various machine learning models, such as RNN, whose convergence rate of $\mathcal{E}_\zeta$ is $\mathcal{O}\left(n^{-\frac{2p_R}{2p_R+d_s+1}}(\log n)^6\right)$ with the model size $p_G$, as proved by Kohler & Krzyżak (2023). According to the results in Theorems 2 and 3, we establish the final theoretical guarantee as follows.

**Theorem 4 (Value difference bound with asynchronous diffusion model)** *Consider the same conditions as in Theorems 2 and 3, then:* $\forall \mathbf{s} \in \mathcal{S}$,

$$\mathbb{E}_{\{\mathbf{o}_t,\mathbf{a}_t,r_t,\mathbf{o}_{t+1}\}}\left|V^\pi(\mathbf{s})-\widetilde{V}^\pi(\zeta(\mathbf{s}))\right| \leq 2\widehat{\epsilon} + \frac{1}{c_R(1-\gamma)}\Bigg(\mathcal{O}\left(n^{-\frac{2p_R}{2p_R+d_s+1}}(\log n)^6\right)$$

$$+ \frac{2c_R+2c_T}{1-c_T-c_R}\mathcal{T}(\mathbf{s}^\star,\mathbf{a}^\star)\mathcal{O}\left(n^{-\frac{b}{2d_s+d_a+2b}}(\log n)^{\max\{19/2,(b+2)/2\}}\right)\Bigg). \quad (15)$$

Therefore, we have established the asymptotic convergence of the proposed algorithm, see Appendix B.2.4 for details. As $n \to \infty$, the estimated causal state $\widetilde{V}^\pi(\zeta(\mathbf{s}))$ in (15) converges to within $2\hat{\epsilon}$-neighborhood of the ground-truth causal state $V^\pi(\mathbf{s})$, i.e., the neighborhood region of the ground-truth causal state $V^\pi(\mathbf{s})$ with the radius of $\hat{\epsilon}$. Specifically, $(c_R, c_T)$ ensures a trade-off between the reward approximation error and the state transition model error, while $(c_R+c_T, 1)$ guarantees a balance between the approximation error of the noisy distribution and the approximation error of bisimulation.

**Computational cost.** We evaluate the additional computational cost of the CSR-ADM compared to typical RL algorithms. Chen et al. (2024) analyzed the computational cost of a diffusion model to be $\widetilde{\mathcal{O}}(\text{poly}\log d)$, where $d$ is the dimension of the input data. Considering our definition of noise intensity, the loss function of the asynchronous diffusion model (see Eq. (5)) is twice that of a standard diffusion model, directly doubling the computational cost. Therefore, the computational cost of the causal state representation is $\widetilde{\mathcal{O}}(\text{poly}\log\max\{|\mathcal{A}|,|\mathcal{O}|\})$ in CSR-ADM.

## 5 EXPERIMENTS

We provide an evaluation of Roboschool environments (Brockman et al., 2016) under standard POMDP implementation by Ni et al. (2022), looking at tasks that typically occlude some part of the observation. There are six environments, i.e., {Hopper, Ant, Walker}-{P, V}, where "-P" stands for observing positions and angles only, and "-V" stands for observing velocities only. For more information about environments, see Appendix C.1. To demonstrate the robustness of the proposed CSR-ADM, we train CSR-ADM with the same hyper-parameters for all six tasks, where we provide the hyper-parameters in Appendix C.2. Considering that the proposed CSR-ADM framework can accommodate any RL algorithm, we extend CSR-ADM to a typical RL algorithm, i.e., SAC. We evaluate all experiments with $600,000$ iterations and apply smoothing operations for each return.

### 5.1 COMPARISON WITH THE BASELINES

By comparing the results with SAC (Fujimoto et al., 2018), DMBP (Zhihe & Xu, 2023a) (only considering denoise), and DBC (Zhang et al., 2021) (only considering bisimulation), we demonstrate

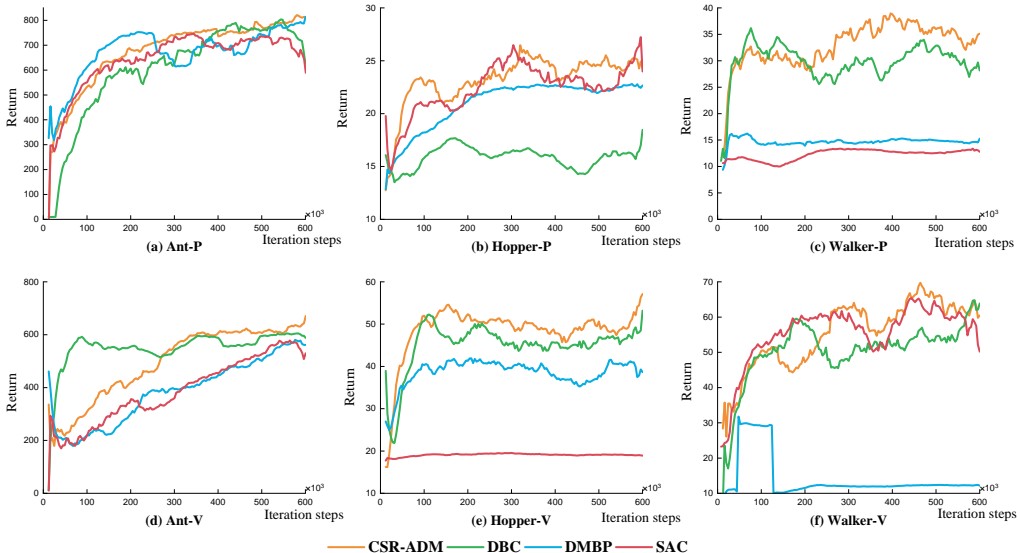

Figure 2: Comparison of the performances of CSR-ADM framework in this paper with three baselines on six environmsnts including Ant-P, Ant-V, Hopper-P, Hopper-V, Walker-P, and Walker-V.

the scalability and effectiveness of CSR-ADM. Specifically, we set both the reward and observation to be affected by Gaussian noise with zero mean, a variance of one, and a scale of two. Additionally, in CSR-ADM, we configure the noise intensity $\delta = 2$ to evaluate the impact of the noise.

As shown in Figure 2, the proposed approach demonstrates superior performance across six environments. Specifically, as compared to DMBP (only considering the denoise functionality in Walker-V) or DBC (solely focusing on bisimulation in Hopper-P), CSR-ADM exhibits superior generalization capabilities. In particular, CSR-ADM improves the return compared to SAC, DMBP, and DBC at least by 14.18%, 29.42%, and 136.63% across the six environments, respectively. Furthermore, although the proposed approach requires learning more parameters than the other algorithms, it achieves better performance in the early stages of training in five out of six environments.

## 5.2 ABLATION STUDY

We conduct ablation studies on all six environments, with three types of modules disabled, i.e., CSR-ADM without bisimulation, CSR-ADM only with reward denoise (i.e., SAC with reward denoise), and CSR-ADM only with observation denoise (i.e., SAC with observation denoise). Specifically, the noise in environments and noise intensity are configured the same as the experiments of comparison with the baselines above.

As shown in Figure 3, we present the performance of the proposed approach in ablation experiments across six environments. By comparing the performance of four cases, it is evident that both bisimulation and the asynchronous diffusion model yield positive contributions to the return. Interestingly, in most environments, the case considering only reward denoising significantly underperforms compared to the case focusing solely on state denoising. This disparity can be attributed to the observation having a higher dimensionality than the reward, resulting in its noise having a greater overall impact. Additionally, the environments of Hopper-V and Walker-P exhibit higher sensitivity to noise, which is also reflected in Table 1.

## 5.3 INFLUENCE ON KEY PARAMETERS

We examine the performance under three noise scales with varying noise intensities across six environments, as shown in Table 1, where bold numbers correspond to the optimal results for the same environment and noise scale. A common conclusion across the six environments is that for noise scale of 0.1, the optimal noise intensity is $\delta = 1$; for noise scale of 0.5, the optimal noise intensity is $\delta = 2$; and if the noise scale increases to 1, the optimal noise intensity becomes unstable, fluctuating

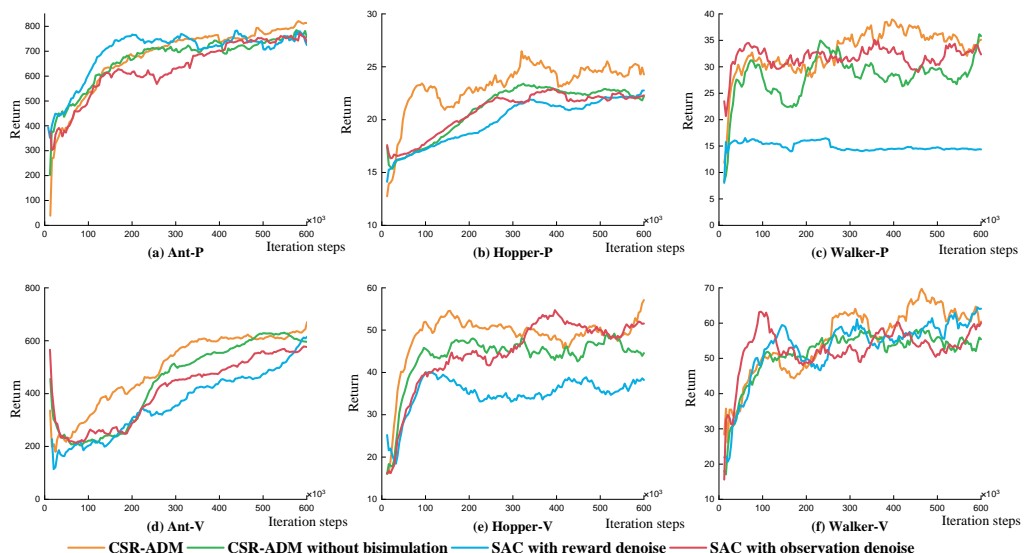

Figure 3: Ablation studies of CSR-ADM framework in this paper on six environments including Ant-P, Ant-V, Hopper-P, Hopper-V, Walker-P, and Walker-V.

Table 1: Returns at different noise intensities with various noise scales in six environments including Ant-P, Ant-V, Hopper-P, Hopper-V, Walker-P, and Walker-V.

| Noise scale | Noise intensity | Ant-P | Ant-V | Hopper-P | Hopper-V | Walker-P | Walker-V |
|---|---|---|---|---|---|---|---|
| | $\delta = 1$ | **790.8** | **573.8** | **214** | **183** | **285.1** | **65.44** |
| 0.1 | $\delta = 2$ | 764.2 | 499.1 | 153.3 | 161 | 221.3 | 52.68 |
| | $\delta = 3$ | 694 | 466 | 122.4 | 128.7 | 215.4 | 58.05 |
| | $\delta = 1$ | 727.7 | 465.4 | 23.87 | 45.58 | 31.04 | 58.15 |
| 0.5 | $\delta = 2$ | **789.3** | **615.4** | **24.18** | **50.17** | **34.01** | **65.24** |
| | $\delta = 3$ | 670.9 | 560.3 | 21.94 | 43.3 | 31.23 | 61.03 |
| | $\delta = 1$ | 569.2 | **538.2** | 24.93 | 45.26 | 35.25 | 50.45 |
| 1 | $\delta = 2$ | 597.3 | 533.8 | **26.2** | 48.16 | **35.8** | 53.36 |
| | $\delta = 3$ | **648.3** | 528.4 | 25.53 | **48.96** | 33.3 | **62.26** |

between 1 and 3, suggesting that higher noise intensities may be necessary for measurement. This indicates that noise intensity can reflect the impact of noise. As the noise scale increases from 0.1 to 0.5, half of the environments exhibit relatively stable returns, and when the noise scale rises from 0.5 to 1, the returns of CSR-ADM across all six environments show no significant change. This suggests that the proposed approach can maintain relatively stable performance in high-noise environments.

## 6 CONCLUSION

In conclusion, this paper introduces the *Causal State Representation under Asynchronous Diffusion Model (CSR-ADM)*, a novel framework that effectively addresses the challenges posed by high-dimensional and noisy input signals in RL applied to POMDPs. By integrating an innovative asynchronous diffusion model for denoising both rewards and observations with bisimulation technology, CSR-ADM captures essential causal state representations, which are crucial for decision-making tasks. Our theoretical analysis provides solid guarantees regarding the approximation of value functions between noisy observation spaces and causal state spaces, reinforcing the framework's robustness. Empirical results from extensive experiments on Roboschool tasks confirm that CSR-ADM surpasses existing state-of-the-art methods, significantly enhancing the performance and robustness of RL algorithms in the presence of varying levels of random noise. This work not only contributes a new approach to improving computational efficiency and generalization in RL but also sets a solid foundation for future research on causal state representation techniques in noisy environments.

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

## A   CSR-ADM STRUCTURE

The proposed framework consists of three modules and can be extended to any RL algorithm. Specifically, CSR-ADM employs an asynchronous diffusion model to denoise states and rewards separately. Subsequently, it approximates causal states based on the denoised states and rewards with bisimulation. Finally, the approximated causal states and denoised rewards are used as a set of samples inputted into the RL algorithm for decision-making. It should be noted that the asynchronous diffusion model algorithm denoises observations, which are then input into the bisimulation metric learning model to extract causal states. The detailed structure is shown in Figure 4.

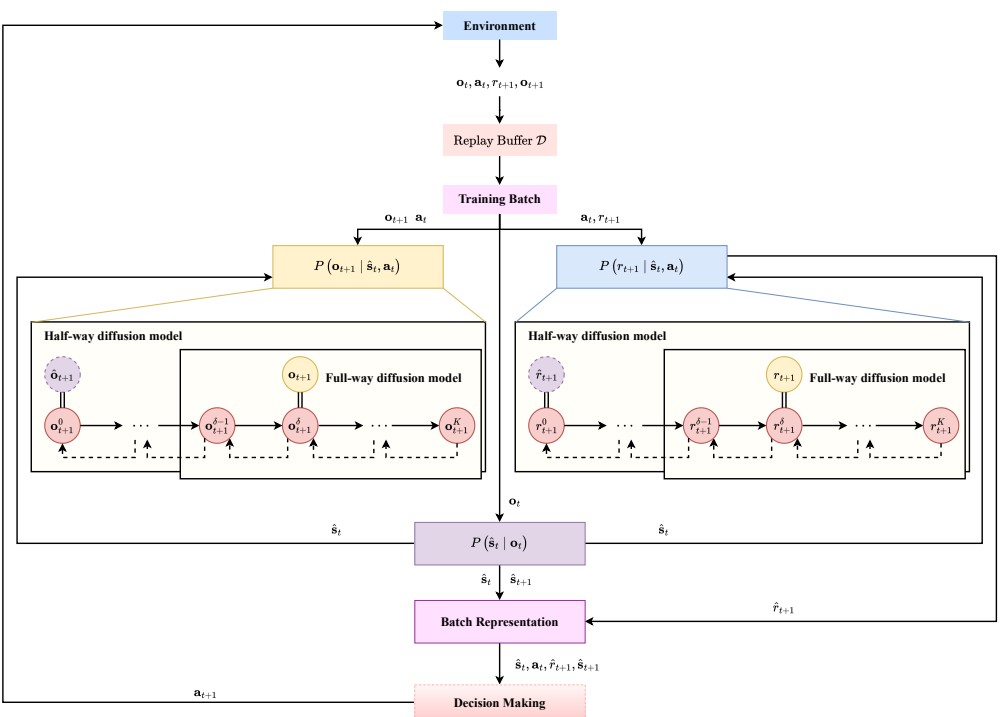

Figure 4: Overview diagram of the proposed CSR-ADM including dynamics estimating under asynchronous diffusion model and causal state representation under bisimulation.

## B   THEORETICAL GUARANTEE

### B.1   ADDITIONAL NOTATION AND BASIC FACTS

#### B.1.1   WASSERSTEIN DISTANCES

**Definition 3 (Wasserstein metric (Villani, 2008))** *Let $d : X \times X \to [0, \infty)$ be a distance function and $\Omega$ be the set of all joint distributions with marginals $\mu$ and $\lambda$ over the space $X$. Then, the Wasserstein metric is given by*

$$W_p(d)(\mu, \lambda) = \left( \inf_{\omega \in \Omega} \mathbb{E}_{(x_1, x_2) \sim \omega} [d(x_1, x_2)^p] \right)^{\frac{1}{p}}. \tag{16}$$

**Definition 4 (Dual formulation of the Wasserstein metric (Villani, 2008))** *Let $d : X \times X \to [0, \infty)$ be a distance function, and $\mu$ and $\lambda$ be marginals over the space $X$. Then, a dual formulation*

*of the Wasserstein metric is given by*

$$W_p(d)(\mu, \lambda) = \left( \sup_{\zeta \oplus \psi \leq d^p} \mathbb{E}_{x_1 \sim \mu}[\zeta(x_1)] + \mathbb{E}_{x_2 \sim \lambda}[\psi(x_2)] \right)^{\frac{1}{p}}, \tag{17}$$

*where $\zeta \oplus \psi \leq d^p$ is equivalent to $\zeta(x) + \psi(y) \leq d(x,y)^p, \ \forall (x,y) \in X \times X$.*

This dual formulation takes a simple form for $p = 1$, which is

$$W_1(d)(\mu, \lambda) = \sup_{f \in \text{Lip}_{1,d}(X)} \mathbb{E}_{x_1 \sim \mu}[f(x_1)] - \mathbb{E}_{x_2 \sim \lambda}[f(x_2)], \tag{18}$$

where $\text{Lip}_{1,d}(X)$ denotes 1-Lipschitz function $f : X \to \mathbb{R}$ such that $|f(x_1) - f(x_2)| \leq d(x_1, x_2)$. Note that the 2-Wasserstein metric $W_2(\|\cdot\|_2)$ (or simply $W_2$) has a closed-form for Gaussian distributions (Olkin & Pukelsheim, 1982):

$$W_2(\mathcal{N}(\mu_i, \Sigma_i), \mathcal{N}(\mu_j, \Sigma_j))^2 = \|\mu_i - \mu_j\|_2^2 + \|\Sigma_i - \Sigma_j\|_{\mathcal{F}}^2, \tag{19}$$

where $\|\cdot\|_{\mathcal{F}}$ denotes the Frobenius norm. We can observe in (19) that for point masses (i.e., $\Sigma_i, \Sigma_j \to 0$), the 2-Wasserstein metric is equivalent to the Euclidean distance between the two points.

**Lemma 1 ($p$-Wasserstein Inequality (Villani, 2008))** *For any two distributions $\mu$ and $\lambda$, if $p \leq q$:*

$$W_p(\mu, \lambda) \leq W_q(\mu, \lambda). \tag{20}$$

**Lemma 2 (Bounds on the Wasserstein distances (Santambrogio, 2015))** *For any two distributions $\mu$ and $\lambda$ over a space $X$, for all $p \geq 1$:*

$$W_1(\mu, \lambda) \leq W_p(\mu, \lambda) \leq \text{diam}(X)^{\frac{p-1}{p}} W_1(\mu, \lambda)^{\frac{1}{p}}. \tag{21}$$

### B.1.2 HÖLDER NORM

**Definition 5 (Hölder norm)** *Let $b = m + \gamma > 0$ be a degree of smoothness, where $m = \lfloor b \rfloor$ is an integer and $\gamma \in [0, 1)$. For a function $f : \mathbb{R}^d \to \mathbb{R}$, its Hölder norm is defined as*

$$\|f\|_{\mathcal{H}^b(\mathbb{R}^d)} := \max_{\mathbf{s}: \|\mathbf{s}\|_1 < m} \sup_{\mathbf{x}} |\partial^{\mathbf{s}} f(\mathbf{x})| + \max_{\mathbf{s}: \|\mathbf{s}\|_1 = s} \sup_{\mathbf{x} \neq \mathbf{z}} \frac{|\partial^{\mathbf{s}} f(\mathbf{x}) - \partial^{\mathbf{s}} f(\mathbf{z})|}{\|\mathbf{x} - \mathbf{z}\|_{\infty}^{\gamma}},$$

*where $\mathbf{s}$ is a multi-index. We say a function $f$ is $b$-Hölder, if and only if $\|f\|_{\mathcal{H}^b(\mathbb{R}^d)} < \infty$.*

We define a Hölder ball of radius $B > 0$ for some constant $B$ as

$$\mathcal{H}^b(\mathbb{R}^d, B) = \left\{ f : \mathbb{R}^d \to \mathbb{R} \Big| \|f\|_{\mathcal{H}^b(\mathbb{R}^d)} < B \right\}.$$

### B.1.3 NOTATION ABOUT ASYNCHRONOUS DIFFUSION MODEL

Given a score approximator $\varphi$, we aim to bound the following conditional score:

$$\mathcal{R}(\varphi) = \int_{k_0}^{K} \frac{1}{K - k_0} \mathbb{E}_{\mathbf{x}^k, \mathbf{y}} \left\| \varphi(\mathbf{x}^k, \mathbf{y}, k) - \nabla \log p(\mathbf{x}^k | \mathbf{y}) \right\|_2^2 dk$$

$$+ \int_{\delta}^{K} \frac{1}{K - \delta} \mathbb{E}_{\mathbf{x}^k, \mathbf{y}} \left\| \varphi(\mathbf{x}^k, \mathbf{y}, k) - \nabla \log p(\mathbf{x}^k | \mathbf{y}) \right\|_2^2 dk.$$

Due to the structure of classifier-free guidance, we first consider the following mixed score error:

$$\mathcal{R}_{\star}(\varphi) = \int_{k_0}^{K} \frac{1}{K - k_0} \mathbb{E}_{\mathbf{x}^k, \mathbf{y}, \tau} \left\| \varphi(\mathbf{x}^k, \tau \mathbf{y}, k) - \nabla \log p(\mathbf{x}^k | \tau \mathbf{y}) \right\|_2^2 dk$$

$$+ \int_{\delta}^{K} \frac{1}{K - \delta} \mathbb{E}_{\mathbf{x}^k, \mathbf{y}, \tau} \left\| \varphi(\mathbf{x}^k, \tau \mathbf{y}, k) - \nabla \log p(\mathbf{x}^k | \tau \mathbf{y}) \right\|_2^2 dk = \mathcal{R} + \mathcal{R}_0, \tag{22}$$

where the conditional score error $\mathcal{R}$ and the unconditional score error $\mathcal{R}_0$ are defined as

$$\mathcal{R} = \frac{1}{2}\int_{k_0}^{K} \frac{1}{K-k_0} \mathbb{E}_{\mathbf{x}^k,\mathbf{y}} \left\| \varphi(\mathbf{x}^k,\mathbf{y},k) - \nabla \log p(\mathbf{x}^k|\mathbf{y}) \right\|_2^2 dk$$

$$+ \frac{1}{2}\int_{\delta}^{K} \frac{1}{K-\delta} \mathbb{E}_{\mathbf{x}^k,\mathbf{y}} \left\| \varphi(\mathbf{x}^k,\mathbf{y},k) - \nabla \log p(\mathbf{x}^k|\mathbf{y}) \right\|_2^2 dk;$$

$$\mathcal{R}_0 = \frac{1}{2}\int_{k_0}^{K} \frac{1}{K-k_0} \mathbb{E}_{\mathbf{x}^k} \left\| \varphi(\mathbf{x}^k,\varnothing,k) - \nabla \log p(\mathbf{x}^k) \right\|_2^2 dk$$

$$+ \frac{1}{2}\int_{\delta}^{K} \frac{1}{K-\delta} \mathbb{E}_{\mathbf{x}^k} \left\| \varphi(\mathbf{x}^k,\varnothing,k) - \nabla \log p(\mathbf{x}^k) \right\|_2^2 dk,$$

which naturally give rise to the inequality $\mathcal{R}(\varphi) \leq 2\mathcal{R}_\star(\varphi)$. Thus, we only need to analyze the bound of $\mathcal{R}_\star(\varphi)$. In practice, we minimize an equivalent loss of $\mathcal{R}_\star$, denoted by $\ell(\varphi)$, which is written as

$$\ell(\varphi) := \int_{k_0}^{K} \frac{1}{K-k_0} \mathbb{E}_{\hat{\mathbf{x}}^0,\mathbf{y}} \left[ \mathbb{E}_{\tau,\mathbf{x}^k|\hat{\mathbf{x}}^0} \left[ \left\| \varphi(\mathbf{x}^k,\tau\mathbf{y},k) - \nabla \log p(\mathbf{x}^k|\hat{\mathbf{x}}^0) \right\|_2^2 \right] \right] dk$$

$$+ \int_{\delta}^{K} \frac{1}{K-\delta} \mathbb{E}_{\mathbf{x}^\delta,\mathbf{y}} \left[ \mathbb{E}_{\tau,\mathbf{x}^k|\mathbf{x}^\delta} \left[ \left\| \varphi(\mathbf{x}^k,\tau\mathbf{y},k) - \nabla \log p(\mathbf{x}^k|\mathbf{x}^\delta) \right\|_2^2 \right] \right] dk, \quad (23)$$

where $\hat{\mathbf{x}}^0 = \frac{1}{\sqrt{\bar{\alpha}_\delta}}\left(\mathbf{x}^\delta - \sqrt{1-\bar{\alpha}_\delta}\epsilon\right)$ and $\epsilon$ follows a standard normal distribution. According to Lemma C.3 in Vincent (2011), (22) differs from (23) by a constant independent of $\mathbf{s}$. Now, we consider training the model with $n$ samples $\{\mathbf{x}_i,\mathbf{y}_i\}_{t=1}^n$ by minimizing the corresponding empirical loss, i.e.,

$$\hat{\ell}(\varphi) = \frac{1}{n}\sum_{i=1}^{n} \ell(\mathbf{x}_i,\mathbf{y}_i,\mathbf{s}), \quad (24)$$

where

$$\ell(\mathbf{x},\mathbf{y};\varphi) := \int_{k_0}^{K} \frac{1}{K-k_0} \mathbb{E}_{\tau,\mathbf{x}^k|\hat{\mathbf{x}}^0} \left[ \left\| \varphi(\mathbf{x}^k,\tau\mathbf{y},k) - \nabla \log p(\mathbf{x}^k|\hat{\mathbf{x}}^0) \right\|_2^2 \right] dk$$

$$+ \int_{\delta}^{K} \frac{1}{K-\delta} \mathbb{E}_{\tau,\mathbf{x}^k|\mathbf{x}^\delta} \left[ \left\| \varphi(\mathbf{x}^k,\tau\mathbf{y},k) - \nabla \log p(\mathbf{x}^k|\mathbf{x}^\delta) \right\|_2^2 \right] dk. \quad (25)$$

Moreover, in order to derive a bounded covering number of our ReLU network function class, we use a truncated loss $\ell^{\mathrm{tr}}(\mathbf{s},\mathbf{x},\mathbf{y})$ defined as:

$$\ell^{\mathrm{tr}}(\mathbf{x},\mathbf{y};\varphi) := \ell(\mathbf{x},\mathbf{y};\varphi)\mathbb{I}\left\{\|\mathbf{x}\|_\infty \leq R\right\}.$$

Accordingly, we denote the truncated domain of the score function by $\mathcal{D} = [-R,R]^d \times [0,1]^{d_y} \cup \{\varnothing\}$. We consider the truncated loss function class defined as

$$\mathcal{S}(R) = \left\{ \ell(\cdot,\cdot;\varphi) : \mathcal{D} \to \mathbb{R} \,\middle|\, \mathbf{s} \in \mathcal{F} \right\}. \quad (26)$$

## B.2  PROOF OF KEY THEOREMS

### B.2.1  PROOF OF THEOREM 1

We prove (13) in Theorem 1 by mathematical induction. Consider the following updates:

$$V^{(t+1)}(\mathbf{s}_i) = \max_{\mathbf{a}\in\mathcal{A}} \left( \int_{r\in\mathcal{R}} r(\mathbf{s}_i,\mathbf{a})\,P(r\mid\mathbf{s}_i,\mathbf{a})\,dr + \gamma \int_{\mathbf{s}'\in\mathcal{S}} P(\mathbf{s}'\mid\mathbf{s}_i,\mathbf{a})\,V^{(t)}(\mathbf{s}')d\mathbf{s}' \right) \quad (27)$$

$$d^{(t+1)}(\mathbf{s}_i,\mathbf{s}_j) = \max_{\mathbf{a}\in\mathcal{A}} \Big( c_{\mathrm{R}}W_p(d^{(t)})\left(P(r\mid\mathbf{s}_i,\mathbf{a}),P(r\mid\mathbf{s}_j,\mathbf{a})\right)$$

$$+ c_{\mathrm{T}}W_p(d^{(t)})\left(P(\mathbf{s}'\mid\mathbf{s}_i,\mathbf{a}),P(\mathbf{s}'\mid\mathbf{s}_j,\mathbf{a})\right) \Big). \quad (28)$$

We need to show that the following holds $\forall t \in \mathbb{N}$:

$$c_{\mathrm{R}} \left| V^{(t)}(\mathbf{s}_i) - V^{(t)}(\mathbf{s}_j) \right| \leq d^{(t)}(\mathbf{s}_i, \mathbf{s}_j), \ \forall (\mathbf{s}_i, \mathbf{s}_j) \in \mathcal{S} \times \mathcal{S}. \tag{29}$$

Then, (13) holds when $t \to \infty$. The base case for mathematical induction, $t = 0$, holds since:

$$\left| V^{(0)}(\mathbf{s}_i) - V^{(0)}(\mathbf{s}_j) \right| = d^{(0)}(\mathbf{s}_i, \mathbf{s}_j) = 0, \ \forall (\mathbf{s}_i, \mathbf{s}_j) \in \mathcal{S} \times \mathcal{S}.$$

Assuming (29) holds at $t$. Then, in the general case for $t + 1$:

$$c_{\mathrm{R}} |V^{(t+1)}(\mathbf{s}_i) - V^{(t+1)}(\mathbf{s}_j)|$$

$$= c_{\mathrm{R}} \left| \max_{\mathbf{a} \in \mathcal{A}} \left( \int_{r \in \mathcal{R}} r(\mathbf{s}_i, \mathbf{a}) P(r \mid \mathbf{s}_i, \mathbf{a}) dr + \gamma \int_{\mathbf{s}' \in \mathcal{S}} P(\mathbf{s}' \mid \mathbf{s}_i, \mathbf{a}) V^{(t)}(\mathbf{s}') d\mathbf{s}' \right) \right.$$

$$\left. - \max_{\mathbf{a} \in \mathcal{A}} \left( \int_{r \in \mathcal{R}} r(\mathbf{s}_j, \mathbf{a}) P(r \mid \mathbf{s}_j, \mathbf{a}) dr + \gamma \int_{\mathbf{s}' \in \mathcal{S}} P(\mathbf{s}' \mid \mathbf{s}_j, \mathbf{a}) V^{(t)}(\mathbf{s}') d\mathbf{s}' \right) \right|$$

$$\leq c_{\mathrm{R}} \left| \max_{\mathbf{a} \in \mathcal{A}} \left( \int_{r \in \mathcal{R}} r(\mathbf{s}_i, \mathbf{a}) P(r \mid \mathbf{s}_i, \mathbf{a}) dr - \int_{r \in \mathcal{R}} r(\mathbf{s}_j, \mathbf{a}) P(r \mid \mathbf{s}_j, \mathbf{a}) dr \right. \right.$$

$$\left. \left. + \gamma \int_{\mathbf{s}' \in \mathcal{S}} \left( P(\mathbf{s}' \mid \mathbf{s}_i, \mathbf{a}) - P(\mathbf{s}' \mid \mathbf{s}_j, \mathbf{a}) \right) V^{(t)}(\mathbf{s}') d\mathbf{s}' \right) \right|$$

$$\leq c_{\mathrm{R}} \max_{\mathbf{a} \in \mathcal{A}} \left| \int_{r \in \mathcal{R}} r(\mathbf{s}_i, \mathbf{a}) P(r \mid \mathbf{s}_i, \mathbf{a}) dr - \int_{r \in \mathcal{R}} r(\mathbf{s}_j, \mathbf{a}) P(r \mid \mathbf{s}_j, \mathbf{a}) dr \right|$$

$$+ c_{\mathrm{R}} \gamma \max_{\mathbf{a} \in \mathcal{A}} \left| \int_{\mathbf{s}' \in \mathcal{S}} \left( P(\mathbf{s}' \mid \mathbf{s}_i, \mathbf{a}) - P(\mathbf{s}' \mid \mathbf{s}_j, \mathbf{a}) \right) V^{(t)}(\mathbf{s}') d\mathbf{s}' \right|$$

$$= c_{\mathrm{R}} \max_{\mathbf{a} \in \mathcal{A}} \left| \int_{r \in \mathcal{R}} r(\mathbf{s}_i, \mathbf{a}) P(r \mid \mathbf{s}_i, \mathbf{a}) dr - \int_{r \in \mathcal{R}} r(\mathbf{s}_j, \mathbf{a}) P(r \mid \mathbf{s}_j, \mathbf{a}) dr \right|$$

$$+ c_{\mathrm{T}} \max_{\mathbf{a} \in \mathcal{A}} \left| \int_{\mathbf{s}' \in \mathcal{S}} \left( P(\mathbf{s}' \mid \mathbf{s}_i, \mathbf{a}) - P(\mathbf{s}' \mid \mathbf{s}_j, \mathbf{a}) \right) \frac{c_{\mathrm{R}} \gamma}{c_{\mathrm{T}}} V^{(t)}(\mathbf{s}') d\mathbf{s}' \right|. \tag{30}$$

Notice that by the induction hypothesis, $c_{\mathrm{R}} V^{(t)}(\mathbf{s})$ is a 1-Lipschitz function with respect to the distance function $d^{(t)}$, i.e., $c_{\mathrm{R}} V^{(t)}(\mathbf{s}) \in \mathrm{Lip}_{1, d^{(t)}}$. Since $\gamma \leq c_{\mathrm{T}}$ by assumption, $\frac{c_{\mathrm{R}} \gamma}{c_{\mathrm{T}}} V^{(t)}(\mathbf{s})$ is also 1-Lipschitz. With the assumption of $r(\mathbf{s}, \mathbf{a}) \in \mathrm{Lip}_{1, d^{(t)}}$, using the dual form of the $W_1$ metric in (18):

$$c_{\mathrm{R}} |V^{(t+1)}(\mathbf{s}_i) - V^{(t+1)}(\mathbf{s}_j)|$$

$$\leq c_{\mathrm{R}} \max_{\mathbf{a} \in \mathcal{A}} \left( W_1(d^{(t)})(P(r \mid \mathbf{s}_i, \mathbf{a}), P(r \mid \mathbf{s}_j, \mathbf{a})) \right) + c_{\mathrm{T}} \max_{\mathbf{a} \in \mathcal{A}} \left( W_1(d^{(t)})(P(\mathbf{s}' \mid \mathbf{s}_i, \mathbf{a}), P(\mathbf{s}' \mid \mathbf{s}_j, \mathbf{a})) \right) \tag{31}$$

$$\leq c_{\mathrm{R}} \max_{\mathbf{a} \in \mathcal{A}} \left( W_p(d^{(t)})(P(r \mid \mathbf{s}_i, \mathbf{a}), P(r \mid \mathbf{s}_j, \mathbf{a})) \right) + c_{\mathrm{T}} \max_{\mathbf{a} \in \mathcal{A}} \left( W_p(d^{(t)})(P(\mathbf{s}' \mid \mathbf{s}_i, \mathbf{a}), P(\mathbf{s}' \mid \mathbf{s}_j, \mathbf{a})) \right)$$

$$= d^{(t+1)},$$

where the last inequality is due to Lemma 1.

### B.2.2 Proof of Theorem 2

To prove Theorem 2, we start with the following lemmas.

**Lemma 3 (Value difference bound with causal state)** *Let $\zeta : \hat{\mathcal{S}} \to \mathcal{S}$ be a function mapping estimated states (i.e., denoised observations) to causal states such that $\zeta(\hat{\mathbf{s}}_i) = \zeta(\hat{\mathbf{s}}_j)$ is equivalent to $d(\hat{\mathbf{s}}_i, \hat{\mathbf{s}}_j) \leq 2\epsilon$. For $c_{\mathrm{R}}, c_{\mathrm{T}} \in [0, 1)$ and $c_{\mathrm{R}} + c_{\mathrm{T}} < 1$:*

$$|V^\pi(\mathbf{s}_i) - \widetilde{V}^\pi(\zeta(\hat{\mathbf{s}}_i))| \leq \frac{2\epsilon}{c_{\mathrm{R}}(1 - \gamma)}, \ \forall \mathbf{s}_i \in \mathcal{S}. \tag{32}$$

Proof found in Appendix B.3.2.

**Lemma 4 (Boundedness condition for convergence)** *Assume $\mathcal{S}$ is compact. If the support of an approximate dynamics model $\widehat{\mathcal{P}}$, i.e., $\mathcal{S}' = \mathrm{supp}(\widehat{P})$, is a closed subset of $\mathcal{S}$, then there exists a unique on-policy bisimulation metric $\widehat{d}$ of the form (12), and this metric is bounded:*

$$\mathrm{supp}(\widehat{P}) \subseteq \mathcal{S} \Rightarrow \mathrm{diam}(\mathcal{S}; \widehat{d}) \leq \frac{c_{\mathrm{R}}}{1 - c_{\mathrm{T}}}(r_{\max} - r_{\min}). \tag{33}$$

Proof found in Appendix B.3.3.

**Lemma 5 (Bisimulation distance error)** *Let $c_{\mathrm{T}} \in [0, 1)$ and $c_{\mathrm{R}} \geq 0$. Assume $\mathrm{supp}(\widehat{\mathcal{P}}) \subseteq \mathcal{S}$ and $1 - (c_{\mathrm{R}} + c_{\mathrm{T}})a_p > 0$. Then,*

$$\left\| d - \widehat{d} \right\|_{\infty} \leq \frac{2c_{\mathrm{R}}}{1 - (c_{\mathrm{R}} + c_{\mathrm{T}})a_p}\mathcal{E}_\phi + \frac{2c_{\mathrm{T}}}{1 - (c_{\mathrm{R}} + c_{\mathrm{T}})a_p}\mathcal{E}_\theta + \frac{(c_{\mathrm{R}} + c_{\mathrm{T}})(a_p - 1)}{1 - (c_{\mathrm{R}} + c_{\mathrm{T}})a_p}\mathrm{diam}(\mathcal{S}; d), \tag{34}$$

*where $a_p = 2^{(p-1)/p}$ and $\mathrm{diam}(\mathcal{S}; d) \leq \frac{c_{\mathrm{R}}}{1-c_{\mathrm{T}}}(r_{\max} - r_{\min})$ based on Lemma 4.*

Proof found in Appendix B.3.4.

For the remainder of this section, we assume $p = 1$.

**Corollary 1 (Bisimulation distance error with $p = 1$)** *Let $p = 1$, with the remaining conditions as in Lemma 5. Then*

$$\left\| d - \widehat{d} \right\|_{\infty} \leq \frac{2c_{\mathrm{R}}}{1 - c_{\mathrm{R}} - c_{\mathrm{T}}}\mathcal{E}_\phi + \frac{2c_{\mathrm{T}}}{1 - c_{\mathrm{R}} - c_{\mathrm{T}}}\mathcal{E}_\theta. \tag{35}$$

When $p = 1$, we have $a_p = a_1 = 1$, giving the expression above.

Corollary 1 bounds the error between the true on-policy bisimulation distance and the optimal *approximate* bisimulation distance (i.e., the best distance function we can achieve with our encoder, given the error in our forward dynamics model). However, we wish to bound the error in the value function in terms of $\widehat{d}_\zeta$, not just $\widehat{d}$ (to take the error of the encoder $\zeta$ into account, as well as that of the dynamics model).

First, we can bound the true bisimulation distance in terms of the encoder and model error. Using Corollary 1 and the definition of bisimulation encoder, there is

$$\left\| d - \widehat{d}_\zeta \right\|_{\infty} \leq \left\| d - \widehat{d} \right\|_{\infty} + \left\| \widehat{d}_\zeta - \widehat{d} \right\|_{\infty} \leq \frac{2c_{\mathrm{R}}}{1 - c_{\mathrm{R}} - c_{\mathrm{T}}}\mathcal{E}_\phi + \frac{2c_{\mathrm{T}}}{1 - c_{\mathrm{R}} - c_{\mathrm{T}}}\mathcal{E}_\theta + \mathcal{E}_\zeta. \tag{36}$$

Thus, if we can relate $d$ to the value function, we can also do so for $\widehat{d}_\zeta$, as a function of model error.

Finally, we look at bounding the difference in the state value function, using the *approximate* bisimulation distance defined through the learned encoder. Let $\widehat{\epsilon}$ be the aggregation radius in $\zeta$-space (meaning the maximum diameter with respect to $\widehat{d}_\zeta$ per partition subset, or equivalence class, is at most $2\widehat{\epsilon}$):

$$\sup_{\mathbf{s}_i, \mathbf{s}_j \in \mathcal{S}} \|\zeta(\mathbf{s}_i) - \zeta(\mathbf{s}_j)\|_q \leq 2\widehat{\epsilon}.$$

Notice that $\widehat{\epsilon}$ bounds the maximal diameter of the partition cells with respect to the *learned* metric, using $\zeta$, rather than the ground truth bisimulation distance.

From the proof of Lemma 3, it readily follows that

$$
(1-\gamma)|V(\mathbf{s}) - \widetilde{V}(\zeta(\hat{\mathbf{s}}))| \leq \frac{c_{\mathrm{R}}^{-1}}{\xi(\zeta(\hat{\mathbf{s}}))} \int\limits_{\mathbf{z}\in\zeta(\hat{\mathbf{s}})} d(\mathbf{s},\mathbf{z})d\xi(\mathbf{z})
$$

$$
\leq \frac{c_{\mathrm{R}}^{-1}}{\xi(\zeta(\hat{\mathbf{s}}))} \int\limits_{\mathbf{z}\in\zeta(\hat{\mathbf{s}})} \widehat{d}_\zeta(\mathbf{s},\mathbf{z}) + \underbrace{|d(\mathbf{s},\mathbf{z}) - \widehat{d}_\zeta(\mathbf{s},\mathbf{z})|_\infty}_{A_3} d\xi(\mathbf{z})
$$

$$
\leq \frac{c_{\mathrm{R}}^{-1}}{\xi(\zeta(\hat{\mathbf{s}}))} \int\limits_{\mathbf{z}\in\zeta(\hat{\mathbf{s}})} 2\widehat{\epsilon} + A_3 \, d\xi(\mathbf{z})
$$

$$
= c_{\mathrm{R}}^{-1}(2\widehat{\epsilon} + A_3)
$$

$$
\leq \frac{1}{c_{\mathrm{R}}} \left( 2\widehat{\epsilon} + \mathcal{E}_\zeta + \frac{2c_{\mathrm{R}}}{1 - c_{\mathrm{R}} - c_{\mathrm{T}}}\mathcal{E}_\phi + \frac{2c_{\mathrm{T}}}{1 - c_{\mathrm{R}} - c_{\mathrm{T}}}\mathcal{E}_\theta \right),
$$

where the last inequality exists due to (36).

### B.2.3   PROOF OF THEOREM 3

For conciseness, we denote $\mathbf{y} = (\hat{\mathbf{s}}_t, \mathbf{a}_t)$. Notice that we have the following decomposition:

$$
W_1(p(\mathbf{x}|\mathbf{y}), \hat{p}(\mathbf{x}^{k_0}|\mathbf{y})) \leq W_1(p(\mathbf{x}|\mathbf{y}), p(\mathbf{x}^{k_0}|\mathbf{y})) + W_1(p(\mathbf{x}^{k_0}|\mathbf{y}), p'(\mathbf{x}^{k_0}|\mathbf{y}))
$$
$$
+ W_1(p'(\mathbf{x}^{k_0}|\mathbf{y}), \hat{p}(\mathbf{x}^{k_0}|\mathbf{y})). \tag{37}
$$

Here, $W_1(p(\mathbf{x}|\mathbf{y}), p(\mathbf{x}^{k_0}|\mathbf{y}))$ follows from the correspondence between the forward and backward processes, $W_1(p(\mathbf{x}^{k_0}|\mathbf{y}), p'(\mathbf{x}^{k_0}|\mathbf{y}))$ follows from the definitions of $\mathbf{x}$ and $\mathbf{x}'$ (with the only difference in the initial distribution), where the latter denotes the result obtained by the true distribution.

We use another backward process as a transition term between $\mathbf{x}'_k$ and $\overline{\mathbf{x}}'_k$, which is defined as

$$
d\overline{\mathbf{x}}'_k = \left[ \frac{1}{2}\overline{\mathbf{x}}'_k + \nabla \log p_{K-k}(\overline{\mathbf{x}}'_k|\mathbf{y}) \right] dk + d\hat{\mathbf{w}}_k \quad \text{with} \quad \overline{\mathbf{x}}'_0 \sim N(0, I). \tag{38}
$$

We denote the conditional distribution of $\overline{\mathbf{x}}'_k$ on $\mathbf{y}$ as $p'_{K-k}(\cdot|\mathbf{y})$. We then bound the three terms in (37), as follows.

**Bound the first term** $W_1(p(\mathbf{x}|\mathbf{y}), p(\mathbf{x}^{k_0}|\mathbf{y}))$. Let $X \sim p(\mathbf{x}|\mathbf{y})$ and $Z \sim N(0, I)$. Then,

$$
W_1(p(\mathbf{x}|\mathbf{y}), p(\mathbf{x}^{k_0}|\mathbf{y})) \leq \mathbb{E}[\|X - \sqrt{\alpha_{k_0}}X + \sigma_{k_0}Z\|] \leq (1 - \sqrt{\alpha_{k_0}})\mathbb{E}[\|X\|] + \sigma_{k_0}\mathbb{E}[\|Z\|]
$$
$$
\leq (1 - \sqrt{\alpha_{k_0}})\sqrt{d} + \sigma_{k_0}\sqrt{d} \lesssim \sqrt{k_0}, \tag{39}
$$

where the last inequality holds due to $\frac{\sigma_k}{\sqrt{\alpha_k}} = \mathcal{O}\left(\sqrt{k}\right)$ when $k = o(1)$.

**Bound the second term** $W_1(p(\mathbf{x}^{k_0}|\mathbf{y}), p'(\mathbf{x}^{k_0}|\mathbf{y}))$. Since $\overline{\mathbf{x}}'_k$ and $\overline{\mathbf{x}}_k$ are obtained through the same backward SDE, but with different initial distributions, by Data Processing Inequality and Pinsker's Inequality (see e.g., Lemma 2 in Canonne (2022)), we have

$$
W_1(p(\mathbf{x}^{k_0}|\mathbf{y}), p'(\mathbf{x}^{k_0}|\mathbf{y})) \lesssim \mathrm{TV}(p(\mathbf{x}^{k_0}|\mathbf{y}), p'(\mathbf{x}^{k_0}|\mathbf{y})) \lesssim \sqrt{\mathrm{KL}(p(\mathbf{x}^{k_0}|\mathbf{y})\|p'(\mathbf{x}^{k_0}|\mathbf{y}))}
$$
$$
\lesssim \sqrt{\mathrm{KL}(p(\mathbf{x}^K|\mathbf{y})\|N(0,I))} \lesssim \sqrt{\mathrm{KL}(p(\mathbf{x}|\mathbf{y})\|N(0,I))}\exp(-K).
$$

Therefore, we obtain

$$
W_1(p(\mathbf{x}^{k_0}|\mathbf{y}), p'(\mathbf{x}^{k_0}|\mathbf{y})) \lesssim \exp(-K). \tag{40}
$$

**Bound the last term** $W_1(p'(\mathbf{x}^{k_0}|\mathbf{y}), \hat{p}(\mathbf{x}^{k_0}|\mathbf{y}))$. Although Assumption 2 does not ensure the Novikov's condition holds, according to Chen et al. (2023b), as long as we have a bounded second moment for the score estimation error and finite KL divergence w.r.t. the standard Gaussian, we can still adopt Girsanov's Theorem and bound the KL divergence between any two distributions produced from the same SDE. We restate the lemma in Fu et al. (2024) as follows:

**Lemma 6 (Lemma D.4 in Fu et al. (2024))** *Let $p_0$ be a probability distribution, and let $Y = \{Y_k\}_{k \in [0,K]}$ and $Y' = \{Y'_k\}_{k \in [0,K]}$ be two stochastic processes that satisfy the following SDEs:*

$$dY_k = s(Y_k, k)dt + dW_k, \quad Y_0 \sim p_0;$$
$$dY'_k = s'(Y'_k, k)dk + dW_k, \quad Y'_0 \sim p_0.$$

*We further define the distributions of $Y_k$ and $Y'_k$ as $p_k$ and $p'_k$, respectively. Suppose that*

$$\int_{\mathbf{x}} p_k(\mathbf{x}) \left\| (s - s')(\mathbf{x}, k) \right\|^2 d\mathbf{x} \le C, \qquad \forall k \in [0, K]. \tag{41}$$

*Then, we have*

$$KL(p_K|p'_K) \le \int_0^K \frac{1}{2} \int_{\mathbf{x}} p_k(\mathbf{x}) \left\| (s - s')(\mathbf{x}, k) \right\|^2 d\mathbf{x}dk.$$

Therefore, we obtain

$$W_1(p'(\mathbf{x}^{k_0}|\mathbf{y}), \hat{p}(\mathbf{x}^{k_0}|\mathbf{y})) \lesssim \mathrm{TV}(p'(\mathbf{x}^{k_0}|\mathbf{y}), \hat{p}(\mathbf{x}^{k_0}|\mathbf{y})) \lesssim \sqrt{\mathrm{KL}(p'(\mathbf{x}^{k_0}|\mathbf{y}), \hat{p}(\mathbf{x}^{k_0}|\mathbf{y}))}$$

$$\lesssim \sqrt{\int_{k_0}^K \frac{1}{2} \int_{\mathbf{x}^k} p_k(\mathbf{x}^k|\mathbf{y}) \left\| \hat{\varphi}(\mathbf{x}^k, \mathbf{y}, k) - \nabla \log p(\mathbf{x}^k|\mathbf{y}) \right\|^2 d\mathbf{x}^k dk}. \tag{42}$$

Given the state and action $\mathbf{y}^\star = (\mathbf{s}^\star, \mathbf{a}^\star)$, we can generate an estimated conditional distribution $p(\mathbf{x}^{k_0}|\mathbf{s}^\star, \mathbf{a}^\star)$ using the backward diffusion process, arriving at

$$W_1(p'(\mathbf{x}^{k_0}|\mathbf{y}), \hat{p}(\mathbf{x}^{k_0}|\mathbf{y})) \lesssim \sqrt{\int_{k_0}^K \frac{1}{2} \int_{\mathbf{x}^k} p(\mathbf{x}^k|\mathbf{s}^\star, \mathbf{a}^\star) \left\| \hat{\varphi}(\mathbf{x}, \mathbf{s}^\star, \mathbf{a}^\star, k) - \nabla \log p(\mathbf{x}^k|\mathbf{s}^\star, \mathbf{a}^\star) \right\|^2 d\mathbf{x}dk}$$

$$= \sqrt{\frac{\int_{k_0}^K \mathbb{E}_{\mathbf{x}^k} \left[ \left\| \hat{\varphi}(\mathbf{x}^k, \mathbf{s}^\star, \mathbf{a}^\star, k) - \nabla \log p(\mathbf{x}^k|\mathbf{s}^\star, \mathbf{a}^\star) \right\|^2 \right] dk}{\int_{k_0}^K \mathbb{E}_{\mathbf{x}^k, \mathbf{s}, \mathbf{a}} \left[ \left\| \hat{\varphi}(\mathbf{x}_k, \mathbf{s}, \mathbf{a}, k) - \nabla \log p(\mathbf{x}^k|\mathbf{s}, \mathbf{a}) \right\|^2 \right] dk}} \cdot \sqrt{\frac{K}{2} \mathcal{R}(\hat{\varphi})}$$

$$\le \mathcal{T}(\mathbf{s}^\star, \mathbf{a}^\star) \sqrt{\frac{K}{2} \mathcal{R}(\hat{\varphi})}, \tag{43}$$

where $\mathcal{R}(\hat{\varphi})$ is defined in Appendix B.1.3. Besides, the distribution coefficient $\mathcal{T}(\mathbf{s}^\star, \mathbf{a}^\star)$ is related to the widely used concentrability coefficient – $L_\infty$ density ratio – in RL (Fan et al., 2020). Since we use the score network $\mathcal{F}$ as a smoothing factor, i.e., the network class $\mathcal{F}$ may not be sensitive to certain differences between the query $(\mathbf{s}^\star, \mathbf{a}^\star)$ and the training data, $\mathcal{T}(\mathbf{s}^\star, \mathbf{a}^\star)$ is always smaller than the concentrability coefficient.

When $\mathbf{y} = (\hat{\mathbf{s}}_t, \mathbf{a}_t)$ is unbounded, we can establish the following lemma:

**Lemma 7** *Suppose Assumption 2 holds. Given the ReLU neural network $\mathcal{F}(M_t, W, \kappa, L, P)$, by taking the network size parameter $N = n^{\frac{1}{d+d_y+2b}}$, the early-stopping step $k_0 = n^{-\mathcal{O}(1)}$ and terminal step $K = \mathcal{O}(\log n)$, the empirical loss minimizer $\hat{\mathbf{s}}$ satisfies*

$$\mathbb{E}_{\{\mathbf{x}_t, \mathbf{y}_t\}_{t=1}^n} [\mathcal{R}(\hat{\varphi})] = O\left( \log \frac{1}{k_0} n^{-\frac{2b}{2d_s + d_a + 2b}} (\log n)^{\max(17, b)} \right). \tag{44}$$

The proof of Lemma 7 is provided in Appendix B.3.5.

Taking expectations w.r.t. the samples $\{\mathbf{x}_t, \mathbf{s}_t, \mathbf{a}_t\}_{t=1}^n$ and applying (44), we have

$$\mathbb{E}_{\{\mathbf{x}_t, \hat{\mathbf{s}}_t, \mathbf{a}_t\}_{t=1}^n}\Big[W_1(p'(\mathbf{x}_t^{k_0}|\hat{\mathbf{s}}_t, \mathbf{a}_t), \hat{p}(\mathbf{x}_t^{k_0}|\hat{\mathbf{s}}_t, \mathbf{a}_t))\Big] \lesssim \mathcal{T}(\mathbf{s}^\star, \mathbf{a}^\star)\sqrt{\frac{K}{2}\log\frac{1}{k_0}} n^{-\frac{2b}{2d_s+d_a+2b}}(\log n)^{\max(17,b)}.$$

We take $k_0 = n^{-\frac{4b}{2d_s+d_a+2b}-1}$ and $K = \frac{2\beta}{2d_s+d_a+2b}\log n$ to bound the expected total variation by

$$\mathbb{E}_{\{\mathbf{x}_t, \hat{\mathbf{s}}_t, \mathbf{a}_t\}_{t=1}^n}\Big[W_1(p'(\mathbf{x}_t^{k_0}|\hat{\mathbf{s}}_t, \mathbf{a}_t), \hat{p}(\mathbf{x}_t^{k_0}|\hat{\mathbf{s}}_t, \mathbf{a}_t))\Big] = \mathcal{T}(\mathbf{s}^\star, \mathbf{a}^\star)\mathcal{O}\Big(n^{-\frac{2b}{2d_s+d_a+2b}}(\log n)^{\max(19/2,(b+2)/2)}\Big).$$

**Putting all this together.** We bound the divergence between $\hat{p}(\mathbf{x}^{k_0}|\mathbf{y})$ and the ground-truth conditional data distribution $p(\mathbf{x}|\mathbf{y})$ as

$$\mathbb{E}_{\{\mathbf{x}_t, \hat{\mathbf{s}}_t, \mathbf{a}_t\}_{t=1}^n}\big[W_1(p(\mathbf{x}|\mathbf{y}), \hat{p}(\mathbf{x}^{k_0}|\mathbf{y}))\big] \leq \mathcal{T}(\mathbf{s}^\star, \mathbf{a}^\star)O\left(n^{-\frac{2b}{2d_s+d_a+2b}}(\log n)^{\max(19/2,(b+2)/2)}\right).$$

This proof is complete.

### B.2.4 STATEMENT OF OVERALL CONVERGENCE

To analyze the convergence of $\mathbb{E}_{\{\mathbf{o}_t, \mathbf{a}_t, r_t, \mathbf{o}_{t+1}\}}\left|V^\pi(\mathbf{s}) - \widetilde{V}^\pi(\zeta(\mathbf{s}))\right|$ is in essence to analyze the convergence of $\frac{\ln^{c_1} n}{n^{c_2}}$. This is because $c_1 = 6$ and $c_2 = \frac{2p_R}{2p_R+d_s+1} > 0$ in $\mathcal{O}(n^{-\frac{2p_R}{2p_R+d_s+1}}(\log n)^6)$; and $c_1 = \max\{\frac{19}{2}, \frac{b+2}{2}\} > 0$ and $c_2 = \frac{b}{2d_s+d_a+2b} > 0$ in $\frac{2c_R+2c_T}{1-c_T-c_R}\mathcal{T}(\mathbf{s}^\star, \mathbf{a}^\star)\mathcal{O}\left(n^{-\frac{b}{2d_s+d_a+2b}}(\log n)^{\max(19/2,(b+2)/2)}\right).$

By applying L'Hôpital's rule, it follows that $\lim_{n\to\infty}\frac{\ln^{c_1} n}{n^{c_2}} = \lim_{n\to\infty}\frac{c_1\ln n}{n\times c_2 n^{c_2-1}} = \lim_{n\to\infty}\frac{c_1\ln n}{c_2 n^{c_2}} = \lim_{n\to\infty}\frac{c_1}{n\times c_2^2 n^{c_2-1}} = \lim_{n\to\infty}\frac{c_1}{c_2^2 n^{c_2}} = 0, \forall c_1, c_2 > 0$. As a result, both terms $\mathcal{O}(n^{-\frac{2p_R}{2p_R+d_s+1}}(\log n)^6)$ and $\frac{2c_R+2c_T}{1-c_T-c_R}\mathcal{T}(\mathbf{s}^\star, \mathbf{a}^\star)\mathcal{O}\left(n^{-\frac{b}{2d_s+d_a+2b}}(\log n)^{\max\{19/2,(b+2)/2\}}\right)$ on the RHS of Eq. (15) converge to zero, as $n \to \infty$. The estimated causal state $\widetilde{V}^\pi(\zeta(\mathbf{s}))$ in Eq. (15) converges to within $2\hat{\epsilon}$-neighborhood of the ground-truth causal state $V^\pi(\mathbf{s})$, i.e., the neighborhood region of the ground-truth causal state $V^\pi(\mathbf{s})$ with the radius of $\hat{\epsilon}$. In other words, the asymptotic convergence of the proposed algorithm is established, as $n \to \infty$.

## B.3 AUXILIARY PROOF

### B.3.1 PROOF OF REMARK 1

We first prove that the following fixed-point update is a contraction:

$$d(\mathbf{s}_i, \mathbf{s}_j) := \max_{\mathbf{a}\in\mathcal{A}}\left(c_R W_p(d)\left(P\left(r \mid \mathbf{s}_i, \mathbf{a}\right), P\left(r \mid \mathbf{s}_j, \mathbf{a}\right)\right) + c_T W_p(d)\left(P\left(\mathbf{s}' \mid \mathbf{s}_i, \mathbf{a}\right), P\left(\mathbf{s}' \mid \mathbf{s}_j, \mathbf{a}\right)\right)\right),$$

and invoke the Banach fixed-point theorem to show the existence of a unique metric.

First, consider the case where $p = 1$:

$$d(\mathbf{s}_i, \mathbf{s}_j) - d'(\mathbf{s}_i, \mathbf{s}_j)$$

$$= \max_{\mathbf{a}\in\mathcal{A}}\left(c_R W_1(d)\left(P\left(r \mid \mathbf{s}_i, \mathbf{a}\right), P\left(r \mid \mathbf{s}_j, \mathbf{a}\right)\right) + c_T W_1(d)\left(P\left(\mathbf{s}' \mid \mathbf{s}_i, \mathbf{a}\right), P\left(\mathbf{s}' \mid \mathbf{s}_j, \mathbf{a}\right)\right)\right)$$

$$- \max_{\mathbf{a}\in\mathcal{A}}\left(c_R W_1(d')\left(P\left(r \mid \mathbf{s}_i, \mathbf{a}\right), P\left(r \mid \mathbf{s}_j, \mathbf{a}\right)\right) + c_T W_1(d')\left(P\left(\mathbf{s}' \mid \mathbf{s}_i, \mathbf{a}\right), P\left(\mathbf{s}' \mid \mathbf{s}_j, \mathbf{a}\right)\right)\right)$$

$$\leq \max_{\mathbf{a}\in\mathcal{A}} c_R\left(W_1(d)\left(P\left(r \mid \mathbf{s}_i, \mathbf{a}\right), P\left(r \mid \mathbf{s}_j, \mathbf{a}\right)\right) - W_1(d')\left(P\left(r \mid \mathbf{s}_i, \mathbf{a}\right), P\left(r \mid \mathbf{s}_j, \mathbf{a}\right)\right)\right)$$

$$+ \max_{\mathbf{a}\in\mathcal{A}} c_T\left(W_1(d)\left(P\left(\mathbf{s}' \mid \mathbf{s}_i, \mathbf{a}\right), P\left(\mathbf{s}' \mid \mathbf{s}_j, \mathbf{a}\right)\right) - W_1(d')\left(P\left(\mathbf{s}' \mid \mathbf{s}_i, \mathbf{a}\right), P\left(\mathbf{s}' \mid \mathbf{s}_j, \mathbf{a}\right)\right)\right)$$

$$= \max_{\mathbf{a}\in\mathcal{A}} c_R\left(W_1(d-d'+d')\left(P\left(r \mid \mathbf{s}_i, \mathbf{a}\right), P\left(r \mid \mathbf{s}_j, \mathbf{a}\right)\right) - W_1(d')\left(P\left(r \mid \mathbf{s}_i, \mathbf{a}\right), P\left(r \mid \mathbf{s}_j, \mathbf{a}\right)\right)\right)$$

$$+ \max_{\mathbf{a}\in\mathcal{A}} c_T\left(W_1(d-d'+d')\left(P\left(\mathbf{s}' \mid \mathbf{s}_i, \mathbf{a}\right), P\left(\mathbf{s}' \mid \mathbf{s}_j, \mathbf{a}\right)\right) - W_1(d')\left(P\left(\mathbf{s}' \mid \mathbf{s}_i, \mathbf{a}\right), P\left(\mathbf{s}' \mid \mathbf{s}_j, \mathbf{a}\right)\right)\right)$$

$$\leq \max_{\mathbf{a}\in\mathcal{A}}\left(c_R W_1 \|d-d'\|_\infty\left(P\left(r \mid \mathbf{s}_i, \mathbf{a}\right), P\left(r \mid \mathbf{s}_j, \mathbf{a}\right)\right) + c_T W_1 \|d-d'\|_\infty\left(P\left(\mathbf{s}' \mid \mathbf{s}_i, \mathbf{a}\right), P\left(\mathbf{s}' \mid \mathbf{s}_j, \mathbf{a}\right)\right)\right)$$

$$\leq (c_R + c_T)\|d-d'\|_\infty, \forall(\mathbf{s}_i, \mathbf{s}_j) \in \mathcal{S} \times \mathcal{S}.$$

For $c_\text{R} + c_\text{T} \in [0, 1)$, there exists a unique fixed-point due to the Banach fixed-point theorem.

Next, we consider the case where both $P$ and $\pi$ are deterministic, such that $P$ is a delta distribution. Observe that for point masses, $W_p(d)(\delta(\mathbf{s}_i), \delta(\mathbf{s}_j)) = d(\mathbf{s}_i, \mathbf{s}_j)$, due to Definition 3 of the Wasserstein metric. Then:

$$
\begin{aligned}
&d(\mathbf{s}_i, \mathbf{s}_j) - d'(\mathbf{s}_i, \mathbf{s}_j) \\
&= \max_{\mathbf{a} \in \mathcal{A}} \left( c_\text{R} W_1(d)\left(P\left(r \mid \mathbf{s}_i, \mathbf{a}\right), P\left(r \mid \mathbf{s}_j, \mathbf{a}\right)\right) + c_\text{T} W_1(d)\left(P\left(\mathbf{s}' \mid \mathbf{s}_i, \mathbf{a}\right), P\left(\mathbf{s}' \mid \mathbf{s}_j, \mathbf{a}\right)\right)\right) \\
&\quad - \max_{\mathbf{a} \in \mathcal{A}} \left( c_\text{R} W_1(d')\left(P\left(r \mid \mathbf{s}_i, \mathbf{a}\right), P\left(r \mid \mathbf{s}_j, \mathbf{a}\right)\right) + c_\text{T} W_1(d')\left(P\left(\mathbf{s}' \mid \mathbf{s}_i, \mathbf{a}\right), P\left(\mathbf{s}' \mid \mathbf{s}_j, \mathbf{a}\right)\right)\right) \\
&= \max_{\mathbf{a} \in \mathcal{A}} \left( c_\text{R} \left(d\left(r_{i'}, r_{j'}\right) - d'\left(r_{i'}, r_{j'}\right)\right) + c_\text{T} \left(d\left(r_{i'}, r_{j'}\right) - d'\left(r_{i'}, r_{j'}\right)\right)\right) \\
&\leq (c_\text{R} + c_\text{T}) \left\|d - d'\right\|_\infty, \ \forall (\mathbf{s}_i, \mathbf{s}_j) \in \mathcal{S} \times \mathcal{S}.
\end{aligned}
$$

Then, the fixed point iterations that update the metric as $d^{(n+1)}(\mathbf{s}_i, \mathbf{s}_j) \leftarrow \mathcal{F}(d^{(n)})(\mathbf{s}_i, \mathbf{s}_j)$ can eventually converge for finite MDPs.

### B.3.2 PROOF OF LEMMA 3

Let $\xi$ be a measure on $\mathcal{S}$. Given a partition $\zeta(\hat{\mathbf{s}}) \in \hat{\mathcal{S}}$, i.e., a set of points in $\mathcal{S}$ clustered in an $\epsilon$-neighborhood such that $\xi(\zeta(\hat{\mathbf{s}})) > 0$, we can define the reward function and transition function of a $\xi$-average finite POMDP as $\xi$-average finite MDP in Theorem 3.21 of Ferns et al. (2011):

$$
\widetilde{P}(r|\zeta(\hat{\mathbf{s}}), \mathbf{a}) = \frac{1}{\xi(\zeta(\hat{\mathbf{s}}))} \int_{\mathbf{z} \in \zeta(\hat{\mathbf{s}})} P(r|\mathbf{z}, \mathbf{a}) d\xi(\mathbf{z}), \tag{45}
$$

$$
\widetilde{P}(\zeta(\hat{\mathbf{s}}')|\zeta(\hat{\mathbf{s}}), \mathbf{a}) = \frac{1}{\xi(\zeta(\hat{\mathbf{s}}))} \int_{\mathbf{z} \in \zeta(\hat{\mathbf{s}})} P(\zeta(\hat{\mathbf{s}}')|\mathbf{z}, \mathbf{a}) d\xi(\mathbf{z}). \tag{46}
$$

Then,

$$
\begin{aligned}
&|V(\mathbf{s}) - \widetilde{V}(\zeta(\hat{\mathbf{s}}))| \\
&= \left| \max_{\mathbf{a} \in \mathcal{A}} \left( \int_{r \in \mathcal{R}} r(\mathbf{s}, \mathbf{a}) P(r \mid \mathbf{s}, \mathbf{a}) dr + \gamma \int_{\mathbf{s}' \in \mathcal{S}} P(\mathbf{s}' \mid \mathbf{s}, \mathbf{a}) V(\mathbf{s}') d\mathbf{s}' \right) \right. \\
&\quad \left. - \max_{\mathbf{a} \in \mathcal{A}} \left( \int_{r \in \mathcal{R}} r(\zeta(\hat{\mathbf{s}}), \mathbf{a}) \widetilde{P}(r|\zeta(\hat{\mathbf{s}}), \mathbf{a}) dr + \gamma \int_{\zeta(\hat{\mathbf{s}}')' \in \hat{\mathcal{S}}} \widetilde{P}(\zeta(\hat{\mathbf{s}}')|\zeta(\hat{\mathbf{s}}), \mathbf{a}) \widetilde{V}(\zeta(\hat{\mathbf{s}}')) d\zeta(\hat{\mathbf{s}}') \right) \right| \\
&\leq \left| \max_{\mathbf{a} \in \mathcal{A}} \left( \int_{r \in \mathcal{R}} \left( r(\mathbf{s}, \mathbf{a}) P(r \mid \mathbf{s}, \mathbf{a}) - r(\zeta(\hat{\mathbf{s}}), \mathbf{a}) \widetilde{P}(r|\zeta(\hat{\mathbf{s}}), \mathbf{a}) \right) dr \right. \right. \\
&\quad \left. \left. + \gamma \left( \int_{\mathbf{s}' \in \mathcal{S}} P(\mathbf{s}' \mid \mathbf{s}, \mathbf{a}) V(\mathbf{s}') d\mathbf{s}' - \int_{\zeta(\hat{\mathbf{s}}') \in \hat{\mathcal{S}}} \widetilde{P}(\zeta(\hat{\mathbf{s}}')|\zeta(\hat{\mathbf{s}}), \mathbf{a}) \widetilde{V}(\zeta(\hat{\mathbf{s}}')) d\zeta(\hat{\mathbf{s}}') \right) \right) \right| \\
&\leq \max_{\mathbf{a} \in \mathcal{A}} \underbrace{\left| \int_{r \in \mathcal{R}} \left( r(\mathbf{s}, \mathbf{a}) P(r \mid \mathbf{s}, \mathbf{a}) - r(\zeta(\hat{\mathbf{s}}), \mathbf{a}) \widetilde{P}(r|\zeta(\hat{\mathbf{s}}), \mathbf{a}) \right) dr \right|}_{A_1} \\
&\quad + \max_{\mathbf{a} \in \mathcal{A}} \gamma \underbrace{\left| \int_{\mathbf{s}' \in \mathcal{S}} P(\mathbf{s}' \mid \mathbf{s}, \mathbf{a}) V(\mathbf{s}') d\mathbf{s}' - \int_{\zeta(\hat{\mathbf{s}}') \in \hat{\mathcal{S}}} \widetilde{P}(\zeta(\hat{\mathbf{s}}')|\zeta(\hat{\mathbf{s}}), \mathbf{a}) \widetilde{V}(\zeta(\hat{\mathbf{s}}')) d\zeta(\hat{\mathbf{s}}') \right|}_{A_2}.
\end{aligned} \tag{47}
$$

Therefore, we can obtain

$$A_1 = \left| \int_{r \in \mathcal{R}} \left( r(\mathbf{s}, \mathbf{a}) P(r \mid \mathbf{s}, \mathbf{a}) - r(\zeta(\hat{\mathbf{s}}), \mathbf{a}) \widetilde{P}(r|\zeta(\hat{\mathbf{s}}), \mathbf{a}) \right) dr \right|$$

$$\leq \frac{1}{\xi(\zeta(\hat{\mathbf{s}}))} \int_{\mathbf{z} \in \zeta(\hat{\mathbf{s}})} \left| \int_{r \in \mathcal{R}} \left( r(\mathbf{s}, \mathbf{a}) P(r \mid \mathbf{s}, \mathbf{a}) - r(\zeta(\hat{\mathbf{s}}), \mathbf{a}) \widetilde{P}(r|\zeta(\hat{\mathbf{s}}), \mathbf{a}) \right) dr \right| d\xi(\mathbf{z})$$

$$\leq \frac{c_{\mathrm{R}}^{-1}}{\xi(\zeta(\hat{\mathbf{s}}))} \int_{\mathbf{z} \in \zeta(\hat{\mathbf{s}})} c_{\mathrm{R}} W_1(d) \left( P(r \mid \mathbf{s}, \mathbf{a}), P(r|\mathbf{z}, \mathbf{a}) \right) d\xi(\mathbf{z})$$

$$\leq \frac{c_{\mathrm{R}}^{-1}}{\xi(\zeta(\hat{\mathbf{s}}))} \int_{\mathbf{z} \in \zeta(\hat{\mathbf{s}})} c_{\mathrm{R}} W_p(d) \left( P(r \mid \mathbf{s}, \mathbf{a}), P(r|\mathbf{z}, \mathbf{a}) \right) d\xi(\mathbf{z}), \tag{48}$$

where the penultimate inequality holds because $r(\mathbf{s}, \mathbf{a})$ is 1-Lipschitz and also because of the dual form of the $W_1$ metric, and the last inequality is due to Lemma 1. Similarly, we can have

$$A_2 = \gamma \left| \int_{\mathbf{s}' \in \mathcal{S}} P(\mathbf{s}' \mid \mathbf{s}, \mathbf{a}) V(\mathbf{s}') d\mathbf{s}' - \int_{\zeta(\hat{\mathbf{s}}') \in \hat{\mathcal{S}}} \widetilde{P}(\zeta(\hat{\mathbf{s}}')|\zeta(\hat{\mathbf{s}}), \mathbf{a}) \widetilde{V}(\zeta(\hat{\mathbf{s}}')) d\zeta(\hat{\mathbf{s}}') \right|$$

$$\leq \frac{\gamma}{\xi(\zeta(\hat{\mathbf{s}}))} \int_{\mathbf{z} \in \zeta(\hat{\mathbf{s}})} \left| \int_{\mathbf{s}' \in \mathcal{S}} P(\mathbf{s}' \mid \mathbf{s}, \mathbf{a}) V(\mathbf{s}') d\mathbf{s}' - \int_{\zeta(\hat{\mathbf{s}}') \in \hat{\mathcal{S}}} P(\zeta(\hat{\mathbf{s}}')|\mathbf{z}, \mathbf{a}) \widetilde{V}(\zeta(\hat{\mathbf{s}}')) d\zeta(\hat{\mathbf{s}}') \right| d\xi(\mathbf{z})$$

$$\leq \frac{\gamma}{\xi(\zeta(\hat{\mathbf{s}}))} \int_{\mathbf{z} \in \zeta(\hat{\mathbf{s}})} \left| \int_{\mathbf{s}' \in \mathcal{S}} \left( P(\mathbf{s}' \mid \mathbf{s}, \mathbf{a}) V(\mathbf{s}') - P(\zeta(\hat{\mathbf{s}}')|\mathbf{z}, \mathbf{a}) \widetilde{V}(\zeta(\hat{\mathbf{s}}')) \right) d\mathbf{s}' \right| d\xi(\mathbf{z})$$

$$\leq \frac{\gamma}{\xi(\zeta(\hat{\mathbf{s}}))} \int_{\mathbf{z} \in \zeta(\hat{\mathbf{s}})} \left| \int_{\mathbf{s}' \in \mathcal{S}} \left( P(\mathbf{s}' \mid \mathbf{s}, \mathbf{a}) V(\mathbf{s}') - P(\mathbf{s}'|\mathbf{z}, \mathbf{a}) V(\mathbf{s}') \right) d\mathbf{s}' \right| d\xi(\mathbf{z})$$

$$+ \frac{\gamma}{\xi(\zeta(\hat{\mathbf{s}}))} \int_{\mathbf{z} \in \zeta(\hat{\mathbf{s}})} \left| \int_{\mathbf{s}' \in \mathcal{S}} \left( P(\zeta(\hat{\mathbf{s}}') \mid \mathbf{z}, \mathbf{a}) \left( V(\mathbf{s}') - \widetilde{V}(\zeta(\hat{\mathbf{s}}')) \right) \right) d\mathbf{s}' \right| d\xi(\mathbf{z}). \tag{49}$$

With $\|\cdot\|_\infty$ defined the supremum norm over $\mathcal{S}$, there is

$$A_2 \leq \frac{\gamma}{\xi(\zeta(\hat{\mathbf{s}}))} \int_{\mathbf{z} \in \zeta(\hat{\mathbf{s}})} \left| \int_{\mathbf{s}' \in \mathcal{S}} \left( P(\mathbf{s}' \mid \mathbf{s}, \mathbf{a}) - P(\mathbf{s}'|\mathbf{z}, \mathbf{a}) \right) V(\mathbf{s}') d\mathbf{s}' \right| d\xi(\mathbf{z}) + \left\| V - \widetilde{V} \right\|_\infty$$

$$\leq \frac{c_{\mathrm{R}}^{-1}}{\xi(\zeta(\hat{\mathbf{s}}))} \int_{\mathbf{z} \in \zeta(\hat{\mathbf{s}})} c_{\mathrm{T}} \left| \int_{\mathbf{s}' \in \mathcal{S}} \left( P(\mathbf{s}' \mid \mathbf{s}, \mathbf{a}) - P(\mathbf{s}'|\mathbf{z}, \mathbf{a}) \right) \frac{c_{\mathrm{R}} \gamma}{c_{\mathrm{T}}} V(\mathbf{s}') d\mathbf{s}' \right| d\xi(\mathbf{z}) + \gamma \left\| V - \widetilde{V} \right\|_\infty$$

$$\leq \frac{c_{\mathrm{R}}^{-1}}{\xi(\zeta(\hat{\mathbf{s}}))} \int_{\mathbf{z} \in \zeta(\hat{\mathbf{s}})} c_{\mathrm{T}} W_1(d) \left( P(\mathbf{s}' \mid \mathbf{s}, \mathbf{a}), P(\mathbf{s}'|\mathbf{z}, \mathbf{a}) \right) d\xi(\mathbf{z}) + \gamma \left\| V - \widetilde{V} \right\|_\infty$$

$$\leq \frac{c_{\mathrm{R}}^{-1}}{\xi(\zeta(\hat{\mathbf{s}}))} \int_{\mathbf{z} \in \zeta(\hat{\mathbf{s}})} c_{\mathrm{T}} W_p(d) \left( P(\mathbf{s}' \mid \mathbf{s}, \mathbf{a}), P(\mathbf{s}'|\mathbf{z}, \mathbf{a}) \right) d\xi(\mathbf{z}) + \gamma \left\| V - \widetilde{V} \right\|_\infty, \tag{50}$$

where the penultimate inequality holds because $\frac{c_{\mathrm{R}} \gamma}{c_{\mathrm{T}}} V(\mathbf{s})$ is 1-Lipschitz together with the dual form of the $W_1$ metric, and the last inequality is due to Lemma 1. Hence,

$$|V(\mathbf{s}) - \widetilde{V}(\zeta(\hat{\mathbf{s}}))| \leq \max_{\mathbf{a} \in \mathcal{A}} (A_1 + A_2)$$

$$\leq \frac{c_{\mathrm{R}}^{-1}}{\xi(\zeta(\hat{\mathbf{s}}))} \int_{\mathbf{z} \in \zeta(\hat{\mathbf{s}})} d(\mathbf{s}, \mathbf{z}) d\xi(\mathbf{z}) + \gamma \left\| V - \widetilde{V} \right\|_\infty \tag{51}$$

$$\leq c_{\mathrm{R}}^{-1} 2\epsilon + \gamma \left\| V - \widetilde{V} \right\|_\infty. \tag{52}$$

Thus, taking the supremum on the LHS over the state space $\mathcal{S}$:

$$|V(\mathbf{s}) - \widetilde{V}(\zeta(\hat{\mathbf{s}}))| \leq \frac{2\epsilon}{c_{\mathrm{R}}(1 - \gamma)}, \quad \forall \mathbf{s} \in \mathcal{S}. \tag{53}$$

### B.3.3 PROOF OF LEMMA 4

**Lemma 8 (Diameter of $\mathcal{S}$ is bounded)** *Let $d : \mathcal{S} \times \mathcal{S} \to [0, \infty)$ be any bisimulation metric:*

$$\text{diam}(\mathcal{S}; d) := \sup_{\mathbf{s}_i, \mathbf{s}_j \in \mathcal{S} \times \mathcal{S}} d(\mathbf{s}_i, \mathbf{s}_j) \leq \frac{c_{\mathrm{R}}}{1 - c_{\mathrm{T}}}(r_{\max} - r_{\min}). \tag{54}$$

This lemma is a slight generalization of the distance bounds given in Theorem 3.12 of Ferns et al. (2011), and the proof follows similarly to Ferns et al. (2011):

$$d(\mathbf{s}_i, \mathbf{s}_j) = \max_{\mathbf{a} \in \mathcal{A}} \left( c_{\mathrm{R}} W_p(d) \left( P\left(r \mid \mathbf{s}_i, \mathbf{a}\right), P\left(r \mid \mathbf{s}_j, \mathbf{a}\right)\right) + c_{\mathrm{T}} W_p(d) \left( P\left(\mathbf{s}' \mid \mathbf{s}_i, \mathbf{a}\right), P\left(\mathbf{s}' \mid \mathbf{s}_j, \mathbf{a}\right)\right) \right)$$

$$\leq c_{\mathrm{R}}(r_{\max} - r_{\min}) + c_{\mathrm{T}} \text{diam}(\mathcal{S}; d), \ \forall (\mathbf{s}_i, \mathbf{s}_j) \in \mathcal{S} \times \mathcal{S},$$

due to Lemma 2 (upper bound as $p \to \infty$). Then,

$$\text{diam}(\mathcal{S}; d) \leq c_{\mathrm{R}}(r_{\max} - r_{\min}) + c_{\mathrm{T}} \text{diam}(\mathcal{S}; d) \leq \frac{c_{\mathrm{R}}}{1 - c_{\mathrm{T}}}(r_{\max} - r_{\min}).$$

The existence proof is almost identical to the proof of Remark 1, except that replaces $P$ with an approximate dynamics model $\widehat{P}$. This is possible since $\mathcal{S}$ is compact by assumption such that $\text{supp}(\widehat{P}) \subseteq \mathcal{S}$ is also compact:

$$d(\mathbf{s}_i, \mathbf{s}_j) - d'(\mathbf{s}_i, \mathbf{s}_j)$$

$$= \max_{\mathbf{a} \in \mathcal{A}} \left[ c_{\mathrm{R}} W_1(d) \left( \widehat{P}\left(r \mid \mathbf{s}_i, \mathbf{a}\right), \widehat{P}\left(r \mid \mathbf{s}_j, \mathbf{a}\right)\right) + c_{\mathrm{T}} W_1(d) \left( \widehat{P}\left(\mathbf{s}' \mid \mathbf{s}_i, \mathbf{a}\right), \widehat{P}\left(\mathbf{s}' \mid \mathbf{s}_j, \mathbf{a}\right)\right) \right]$$

$$- \max_{\mathbf{a} \in \mathcal{A}} \left[ c_{\mathrm{R}} W_1(d') \left( \widehat{P}\left(r \mid \mathbf{s}_i, \mathbf{a}\right), \widehat{P}\left(r \mid \mathbf{s}_j, \mathbf{a}\right)\right) + c_{\mathrm{T}} W_1(d') \left( \widehat{P}\left(\mathbf{s}' \mid \mathbf{s}_i, \mathbf{a}\right), \widehat{P}\left(\mathbf{s}' \mid \mathbf{s}_j, \mathbf{a}\right)\right) \right]$$

$$\leq \max_{\mathbf{a} \in \mathcal{A}} c_{\mathrm{R}} \left[ W_1(d) \left( \widehat{P}\left(r \mid \mathbf{s}_i, \mathbf{a}\right), \widehat{P}\left(r \mid \mathbf{s}_j, \mathbf{a}\right)\right) - W_1(d') \left( \widehat{P}\left(r \mid \mathbf{s}_i, \mathbf{a}\right), \widehat{P}\left(r \mid \mathbf{s}_j, \mathbf{a}\right)\right) \right]$$

$$+ \max_{\mathbf{a} \in \mathcal{A}} c_{\mathrm{T}} \left[ W_1(d) \left( \widehat{P}\left(\mathbf{s}' \mid \mathbf{s}_i, \mathbf{a}\right), \widehat{P}\left(\mathbf{s}' \mid \mathbf{s}_j, \mathbf{a}\right)\right) - W_1(d') \left( \widehat{P}\left(\mathbf{s}' \mid \mathbf{s}_i, \mathbf{a}\right), \widehat{P}\left(\mathbf{s}' \mid \mathbf{s}_j, \mathbf{a}\right)\right) \right]$$

$$= \max_{\mathbf{a} \in \mathcal{A}} c_{\mathrm{R}} \left[ W_1(d - d' + d') \left( \widehat{P}\left(r \mid \mathbf{s}_i, \mathbf{a}\right), \widehat{P}\left(r \mid \mathbf{s}_j, \mathbf{a}\right)\right) - W_1(d') \left( \widehat{P}\left(r \mid \mathbf{s}_i, \mathbf{a}\right), \widehat{P}\left(r \mid \mathbf{s}_j, \mathbf{a}\right)\right) \right]$$

$$+ \max_{\mathbf{a} \in \mathcal{A}} c_{\mathrm{T}} \left[ W_1(d - d' + d') \left( \widehat{P}\left(\mathbf{s}' \mid \mathbf{s}_i, \mathbf{a}\right), \widehat{P}\left(\mathbf{s}' \mid \mathbf{s}_j, \mathbf{a}\right)\right) - W_1(d') \left( \widehat{P}\left(\mathbf{s}' \mid \mathbf{s}_i, \mathbf{a}\right), \widehat{P}\left(\mathbf{s}' \mid \mathbf{s}_j, \mathbf{a}\right)\right) \right]$$

$$\leq \max_{\mathbf{a} \in \mathcal{A}} \left[ c_{\mathrm{R}} W_1 \|d - d'\|_\infty \left( \widehat{P}\left(r \mid \mathbf{s}_i, \mathbf{a}\right), \widehat{P}\left(r \mid \mathbf{s}_j, \mathbf{a}\right)\right) + c_{\mathrm{T}} W_1 \|d - d'\|_\infty \left( \widehat{P}\left(\mathbf{s}' \mid \mathbf{s}_i, \mathbf{a}\right), \widehat{P}\left(\mathbf{s}' \mid \mathbf{s}_j, \mathbf{a}\right)\right) \right]$$

$$\leq (c_{\mathrm{R}} + c_{\mathrm{T}}) \|d - d'\|_\infty, \ \forall (\mathbf{s}_i, \mathbf{s}_j) \in \mathcal{S} \times \mathcal{S},$$

which implies $\mathcal{F}$ is a $(c_{\mathrm{R}} + c_{\mathrm{T}})$-contraction. Next, we proceed to prove that the distance is bounded. First, note that due to Lemma 2:

$$\text{supp}(\widehat{P}) \subseteq \mathcal{S} \Rightarrow \sup_{\mathbf{s}_i, \mathbf{s}_j \in \mathcal{S} \times \mathcal{S}} W_p(\widehat{d})(\widehat{P}^\pi(\cdot | \mathbf{s}_i, \mathbf{a}), \widehat{P}^\pi(\cdot | \mathbf{s}_j, \mathbf{a})) \leq \text{diam}(\mathcal{S}; \widehat{d}), \ \forall p \geq 1. \tag{55}$$

Then, similarly to Lemma 8, we have

$$\widehat{d}(\mathbf{s}_i, \mathbf{s}_j) = \max_{\mathbf{a} \in \mathcal{A}} \left( c_{\mathrm{R}} W_p(\widehat{d}) \left( \widehat{P}\left(r | \mathbf{s}_i, \mathbf{a}\right), \widehat{P}\left(r | \mathbf{s}_j, \mathbf{a}\right)\right) + c_{\mathrm{T}} W_p(\widehat{d}) \left( \widehat{P}\left(\mathbf{s}' | \mathbf{s}_i, \mathbf{a}\right), \widehat{P}\left(\mathbf{s}' | \mathbf{s}_j, \mathbf{a}\right)\right) \right)$$

$$\leq c_{\mathrm{R}}(r_{\max} - r_{\min}) + c_{\mathrm{T}} \text{diam}(\mathcal{S}; \widehat{d}), \ \forall (\mathbf{s}_i, \mathbf{s}_j) \in \mathcal{S} \times \mathcal{S},$$

which implies that:

$$\text{diam}(\mathcal{S}; \widehat{d}) \leq c_{\mathrm{R}}(r_{\max} - r_{\min}) + c_{\mathrm{T}} \text{diam}(\mathcal{S}; \widehat{d}) \leq \frac{c_{\mathrm{R}}}{1 - c_{\mathrm{T}}}(r_{\max} - r_{\min}).$$

### B.3.4 PROOF OF LEMMA 5

First, by the Wasserstein triangle inequality (Clement & Desch, 2008), we define the difference for rewards and transitions, respectively:

$$\left| W_p(d) \left( P\left(r \mid \mathbf{s}_i, \mathbf{a}\right), P\left(r \mid \mathbf{s}_j, \mathbf{a}\right)\right) - W_p(d) \left( \widehat{P}\left(r \mid \hat{\mathbf{s}}_i, \mathbf{a}\right), \widehat{P}\left(r \mid \hat{\mathbf{s}}, \mathbf{a}\right)\right) \right| \leq 2\mathcal{E}_\phi; \tag{56}$$

$$\left| W_p(d) \left( P\left(\mathbf{s}' \mid \mathbf{s}_i, \mathbf{a}\right), P\left(\mathbf{s}' \mid \mathbf{s}_j, \mathbf{a}\right)\right) - W_p(d) \left( \widehat{P}\left(\mathbf{s}' \mid \hat{\mathbf{s}}, \mathbf{a}\right), \widehat{P}\left(\mathbf{s}' \mid \hat{\mathbf{s}}, \mathbf{a}\right)\right) \right| \leq 2\mathcal{E}_\theta. \tag{57}$$

Second, the convexity of $d^p$ implies that,

$$W_p\left(\|d-\widehat{d}\|_\infty + d\right)\left(\widehat{P}\left(\mathbf{s}' \mid \mathbf{s}, \mathbf{a}\right), \widehat{P}\left(\mathbf{s}' \mid \mathbf{s}, \mathbf{a}\right)\right)$$

$$= \left(\inf_{\omega\in\Omega} \mathbb{E}_{(\mathbf{s}_i,\mathbf{s}_j)\sim\omega}[(\|d-\widehat{d}\|_\infty + d(\mathbf{s}_i,\mathbf{s}_j))^p]\right)^{\frac{1}{p}}$$

$$\leq \left(\inf_{\omega\in\Omega} 2^{p-1}\mathbb{E}_{(\mathbf{s}_i,\mathbf{s}_j)\sim\omega}[(\|d-\widehat{d}\|_\infty^p + d(\mathbf{s}_i,\mathbf{s}_j)^p]\right)^{\frac{1}{p}}$$

$$\leq a_p\left(\|d-\widehat{d}\|_\infty^p + W_p^p(d)\left(\widehat{P}\left(\mathbf{s}' \mid \mathbf{s}_i, \mathbf{a}\right), \widehat{P}\left(\mathbf{s}' \mid \mathbf{s}_j, \mathbf{a}\right)\right)\right)^{\frac{1}{p}}$$

$$\leq a_p\left(\left[\|d-\widehat{d}\|_\infty + W_p(d)\left(\widehat{P}\left(\mathbf{s}' \mid \mathbf{s}_i, \mathbf{a}\right), \widehat{P}\left(\mathbf{s}' \mid \mathbf{s}_j, \mathbf{a}\right)\right)\right]^p\right)^{1/p}$$

$$= a_p\left(\|d-\widehat{d}\|_\infty + W_p(d)\left(\widehat{P}\left(\mathbf{s}' \mid \mathbf{s}_i, \mathbf{a}\right), \widehat{P}\left(\mathbf{s}' \mid \mathbf{s}_j, \mathbf{a}\right)\right)\right). \tag{58}$$

Similarly, we obtain

$$W_p\left(\|d-\widehat{d}\|_\infty + d\right)\left(\widehat{P}\left(r \mid \mathbf{s}_i, \mathbf{a}\right), \widehat{P}\left(r \mid \mathbf{s}_j, \mathbf{a}\right)\right)$$

$$= \left(\inf_{\omega\in\Omega} \mathbb{E}_{(\mathbf{s}_i,\mathbf{s}_j)\sim\omega}[(\|d-\widehat{d}\|_\infty + d(\mathbf{s}_i,\mathbf{s}_j))^p]\right)^{\frac{1}{p}}$$

$$\leq \left(\inf_{\omega\in\Omega} 2^{p-1}\mathbb{E}_{(\mathbf{s}_i,\mathbf{s}_j)\sim\omega}[(\|d-\widehat{d}\|_\infty^p + d(\mathbf{s}_i,\mathbf{s}_j)^p]\right)^{\frac{1}{p}}$$

$$\leq a_p\left(\|d-\widehat{d}\|_\infty^p + W_p^p(d)\left(\widehat{P}\left(r \mid \mathbf{s}_i, \mathbf{a}\right), \widehat{P}\left(r \mid \mathbf{s}_j, \mathbf{a}\right)\right)\right)^{\frac{1}{p}}$$

$$\leq a_p\left(\left[\|d-\widehat{d}\|_\infty + W_p(d)\left(\widehat{P}\left(r \mid \mathbf{s}_i, \mathbf{a}\right), \widehat{P}\left(r \mid \mathbf{s}_j, \mathbf{a}\right)\right)\right]^p\right)^{1/p}$$

$$= a_p\left(\|d-\widehat{d}\|_\infty + W_p(d)\left(\widehat{P}\left(r \mid \mathbf{s}_i, \mathbf{a}\right), \widehat{P}\left(r \mid \mathbf{s}_j, \mathbf{a}\right)\right)\right). \tag{59}$$

Third, recall that when $\text{supp}(\widehat{P}) \subseteq \mathcal{S}$, due to Lemma 2, we have:

$$W_p(d)\left(\widehat{P}\left(\mathbf{s}' \mid \mathbf{s}_i, \mathbf{a}\right), \widehat{P}\left(\mathbf{s}' \mid \mathbf{s}_j, \mathbf{a}\right)\right) \leq \text{diam}(\mathcal{S}; d) \tag{60}$$

$$W_p(d)\left(\widehat{P}\left(r \mid \mathbf{s}_i, \mathbf{a}\right), \widehat{P}\left(r \mid \mathbf{s}_j, \mathbf{a}\right)\right) \leq \text{diam}(\mathcal{S}; d). \tag{61}$$

Then, the difference in distances can be bounded by:

$$\left| W_p(d)\left(P\left(\mathbf{s}' \mid \mathbf{s}_i, \mathbf{a}\right), P\left(\mathbf{s}' \mid \mathbf{s}_j, \mathbf{a}\right)\right) - W_p(\widehat{d})\left(\widehat{P}\left(\mathbf{s}' \mid \mathbf{s}_i, \mathbf{a}\right), \widehat{P}\left(\mathbf{s}' \mid \mathbf{s}_j, \mathbf{a}\right)\right) \right|$$

$$\leq \left| W_p(\widehat{d})\left(\widehat{P}\left(\mathbf{s}' \mid \mathbf{s}_i, \mathbf{a}\right), \widehat{P}\left(\mathbf{s}' \mid \mathbf{s}_j, \mathbf{a}\right)\right) - W_p(d)\left(\widehat{P}\left(\mathbf{s}' \mid \mathbf{s}_i, \mathbf{a}\right), \widehat{P}\left(\mathbf{s}' \mid \mathbf{s}_j, \mathbf{a}\right)\right) \right| \tag{62}$$

$$\quad + \left| W_p(d)\left(P\left(\mathbf{s}' \mid \mathbf{s}_i, \mathbf{a}\right), P\left(\mathbf{s}' \mid \mathbf{s}_j, \mathbf{a}\right)\right) - W_p(d)\left(\widehat{P}\left(\mathbf{s}' \mid \mathbf{s}_i, \mathbf{a}\right), \widehat{P}\left(\mathbf{s}' \mid \mathbf{s}_j, \mathbf{a}\right)\right) \right|$$

$$\leq \left| W_p(\widehat{d})\left(\widehat{P}\left(\mathbf{s}' \mid \mathbf{s}_i, \mathbf{a}\right), \widehat{P}\left(\mathbf{s}' \mid \mathbf{s}_j, \mathbf{a}\right)\right) - W_p(d)\left(\widehat{P}\left(\mathbf{s}' \mid \mathbf{s}_i, \mathbf{a}\right), \widehat{P}\left(\mathbf{s}' \mid \mathbf{s}_j, \mathbf{a}\right)\right) \right| + 2\mathcal{E}_\theta$$

$$= \left| W_p(\widehat{d} - d + d)\left(\widehat{P}\left(\mathbf{s}' \mid \mathbf{s}_i, \mathbf{a}\right), \widehat{P}\left(\mathbf{s}' \mid \mathbf{s}_j, \mathbf{a}\right)\right) - W_p(d)\left(\widehat{P}\left(\mathbf{s}' \mid \mathbf{s}_i, \mathbf{a}\right), \widehat{P}\left(\mathbf{s}' \mid \mathbf{s}_j, \mathbf{a}\right)\right) \right| + 2\mathcal{E}_\theta$$

$$\leq \left| W_p(\|\widehat{d} - d\|_\infty + d)\left(\widehat{P}\left(\mathbf{s}' \mid \mathbf{s}_i, \mathbf{a}\right), \widehat{P}\left(\mathbf{s}' \mid \mathbf{s}_j, \mathbf{a}\right)\right) - W_p(d)\left(\widehat{P}\left(\mathbf{s}' \mid \mathbf{s}_i, \mathbf{a}\right), \widehat{P}\left(\mathbf{s}' \mid \mathbf{s}_j, \mathbf{a}\right)\right) \right| + 2\mathcal{E}_\theta$$

$$= \left| W_p(\|d - \widehat{d}\|_\infty + d)\left(\widehat{P}\left(\mathbf{s}' \mid \mathbf{s}_i, \mathbf{a}\right), \widehat{P}\left(\mathbf{s}' \mid \mathbf{s}_j, \mathbf{a}\right)\right) - W_p(d)\left(\widehat{P}\left(\mathbf{s}' \mid \mathbf{s}_i, \mathbf{a}\right), \widehat{P}\left(\mathbf{s}' \mid \mathbf{s}_j, \mathbf{a}\right)\right) \right| + 2\mathcal{E}_\theta$$

$$\leq \left| a_p\|d - \widehat{d}\|_\infty + a_p W_p(d)\left(\widehat{P}\left(\mathbf{s}' \mid \mathbf{s}_i, \mathbf{a}\right), \widehat{P}\left(\mathbf{s}' \mid \mathbf{s}_j, \mathbf{a}\right)\right) \right.$$

$$\left. - W_p(d)\left(\widehat{P}\left(\mathbf{s}' \mid \mathbf{s}_i, \mathbf{a}\right), \widehat{P}\left(\mathbf{s}' \mid \mathbf{s}_j, \mathbf{a}\right)\right) \right| + 2\mathcal{E}_\theta$$

$$\leq a_p\|d - \widehat{d}\|_\infty + (a_p - 1)\text{diam}(\mathcal{S}; d) + 2\mathcal{E}_\theta, \tag{63}$$

where the second inequality holds due to (57), the penultimate inequality exists with (58), and the last inequality comes from (60). Similarly, we get

$$\left| W_p(d) \left( P\left( r \mid \mathbf{s}_i, \mathbf{a} \right), P\left( r \mid \mathbf{s}_j, \mathbf{a} \right) \right) - W_p(\widehat{d}) \left( \widehat{P}\left( r \mid \mathbf{s}_i, \mathbf{a} \right), \widehat{P}\left( r \mid \mathbf{s}_j, \mathbf{a} \right) \right) \right|$$

$$\leq \left| W_p(\widehat{d}) \left( \widehat{P}\left( r \mid \mathbf{s}_i, \mathbf{a} \right), \widehat{P}\left( r \mid \mathbf{s}_j, \mathbf{a} \right) \right) - W_p(d) \left( \widehat{P}\left( r \mid \mathbf{s}_i, \mathbf{a} \right), \widehat{P}\left( r \mid \mathbf{s}_j, \mathbf{a} \right) \right) \right|$$

$$+ \left| W_p(d) \left( P\left( r \mid \mathbf{s}_i, \mathbf{a} \right), P\left( r \mid \mathbf{s}_j, \mathbf{a} \right) \right) - W_p(d) \left( \widehat{P}\left( r \mid \mathbf{s}_i, \mathbf{a} \right), \widehat{P}\left( r \mid \mathbf{s}_j, \mathbf{a} \right) \right) \right|$$

$$\leq \left| W_p(\widehat{d}) \left( \widehat{P}\left( r \mid \mathbf{s}_i, \mathbf{a} \right), \widehat{P}\left( r \mid \mathbf{s}_j, \mathbf{a} \right) \right) - W_p(d) \left( \widehat{P}\left( r \mid \mathbf{s}_i, \mathbf{a} \right), \widehat{P}\left( r \mid \mathbf{s}_j, \mathbf{a} \right) \right) \right| + 2\mathcal{E}_\phi$$

$$= \left| W_p(\widehat{d} - d + d) \left( \widehat{P}\left( r \mid \mathbf{s}_i, \mathbf{a} \right), \widehat{P}\left( r \mid \mathbf{s}_j, \mathbf{a} \right) \right) - W_p(d) \left( \widehat{P}\left( r \mid \mathbf{s}_i, \mathbf{a} \right), \widehat{P}\left( r \mid \mathbf{s}_j, \mathbf{a} \right) \right) \right| + 2\mathcal{E}_\phi$$

$$\leq \left| W_p(\|\widehat{d} - d\|_\infty + d) \left( \widehat{P}\left( r \mid \mathbf{s}_i, \mathbf{a} \right), \widehat{P}\left( r \mid \mathbf{s}_j, \mathbf{a} \right) \right) - W_p(d) \left( \widehat{P}\left( r \mid \mathbf{s}_i, \mathbf{a} \right), \widehat{P}\left( r \mid \mathbf{s}_j, \mathbf{a} \right) \right) \right| + 2\mathcal{E}_\phi$$

$$= \left| W_p(\|d - \widehat{d}\|_\infty + d) \left( \widehat{P}\left( r \mid \mathbf{s}_i, \mathbf{a} \right), \widehat{P}\left( r \mid \mathbf{s}_j, \mathbf{a} \right) \right) - W_p(d) \left( \widehat{P}\left( r \mid \mathbf{s}_i, \mathbf{a} \right), \widehat{P}\left( r \mid \mathbf{s}_j, \mathbf{a} \right) \right) \right| + 2\mathcal{E}_\phi$$

$$\leq \left| a_p \|d - \widehat{d}\|_\infty + a_p W_p(d) \left( \widehat{P}\left( r \mid \mathbf{s}_i, \mathbf{a} \right), \widehat{P}\left( r \mid \mathbf{s}_j, \mathbf{a} \right) \right) \right.$$

$$\left. - W_p(d) \left( \widehat{P}\left( r \mid \mathbf{s}_i, \mathbf{a} \right), \widehat{P}\left( r \mid \mathbf{s}_j, \mathbf{a} \right) \right) \right| + 2\mathcal{E}_\phi$$

$$\leq a_p \|d - \widehat{d}\|_\infty + (a_p - 1)\mathrm{diam}(\mathcal{S}; d) + 2\mathcal{E}_\phi. \tag{64}$$

We can then plug (63) and (64) into the difference between the true and approximate policy-dependent bisimulation distances:

$$|d(\mathbf{s}_i, \mathbf{s}_j) - \widehat{d}(\mathbf{s}_i, \mathbf{s}_j)|$$

$$\leq \max_{\mathbf{a} \in \mathcal{A}} \left( c_\mathrm{R} \left| W_p(d) \left( P\left( r \mid \mathbf{s}_i, \mathbf{a} \right), P\left( r \mid \mathbf{s}_j, \mathbf{a} \right) \right) - W_p(\widehat{d}) \left( \widehat{P}\left( r \mid \mathbf{s}_i, \mathbf{a} \right), \widehat{P}\left( r \mid \mathbf{s}_j, \mathbf{a} \right) \right) \right| \right)$$

$$+ \max_{\mathbf{a} \in \mathcal{A}} \left( c_\mathrm{T} \left| W_p(d) \left( P\left( \mathbf{s}' \mid \mathbf{s}_i, \mathbf{a} \right), P\left( \mathbf{s}' \mid \mathbf{s}_j, \mathbf{a} \right) \right) - W_p(\widehat{d}) \left( \widehat{P}\left( \mathbf{s}' \mid \mathbf{s}_i, \mathbf{a} \right), \widehat{P}\left( \mathbf{s}' \mid \mathbf{s}_j, \mathbf{a} \right) \right) \right| \right)$$

$$\leq c_\mathrm{R} \left| a_p \|d - \widehat{d}\|_\infty + (a_p - 1)\mathrm{diam}(\mathcal{S}; d) + 2\mathcal{E}_\phi \right|$$

$$+ c_\mathrm{T} \left| a_p \|d - \widehat{d}\|_\infty + (a_p - 1)\mathrm{diam}(\mathcal{S}; d) + 2\mathcal{E}_\theta \right|;$$

$$\|d - \widehat{d}\|_\infty \leq 2c_\mathrm{R}\mathcal{E}_\phi + 2c_\mathrm{T}\mathcal{E}_\theta + (c_\mathrm{R} + c_\mathrm{T})a_p\|d - \widehat{d}\|_\infty + (c_\mathrm{R} + c_\mathrm{T})(a_p - 1)\mathrm{diam}(\mathcal{S}; d);$$

$$\|d - \widehat{d}\|_\infty \leq \frac{2c_\mathrm{R}}{1 - (c_\mathrm{R} + c_\mathrm{T})a_p}\mathcal{E}_\phi + \frac{2c_\mathrm{T}}{1 - (c_\mathrm{R} + c_\mathrm{T})a_p}\mathcal{E}_\theta + \frac{(c_\mathrm{R} + c_\mathrm{T})(a_p - 1)}{1 - (c_\mathrm{R} + c_\mathrm{T})a_p}\mathrm{diam}(\mathcal{S}; d),$$

where the second-last inequality follows by taking the supremum over states for both sides.

### B.3.5 PROOF OF LEMMA 7

First, we prove the approximation theory for using ReLU neural networks to approximate the conditional score, that is

**Lemma 9** *Under Assumption 2, for sufficiently large $N$ and constant $C_\alpha > 0$, by taking terminal step $K = C_\alpha \log N$, there exists $\mathbf{s} \in \mathcal{F}(M_t, W, \kappa, L, P)$ such that for all $\mathbf{y} \in [0, 1]^{d_y}$ and $k \in [0, K]$, it holds that*

$$\int_{\mathbf{x}^{k_1}} \|\zeta(\mathbf{x}^{k_1}, \mathbf{y}, k_1) - \nabla \log p_{k_1}(\mathbf{x}^{k_1}|\mathbf{y})\|_2^2 \cdot p_{k_1}(\mathbf{x}^{k_1}|\mathbf{y}) d\mathbf{x}^{k_1}$$

$$+ \int_{\mathbf{x}^{k_2}} \|\zeta(\mathbf{x}^{k_2}, \mathbf{y}, k_2) - \nabla \log p_{k_2}(\mathbf{x}^{k_2}|\mathbf{y})\|_2^2 \cdot p_{k_2}(\mathbf{x}^k|\mathbf{y}) d\mathbf{x}^{k_2} = \mathcal{O}\left( \frac{B^2}{\sigma_k^2} \cdot N^{-\frac{2b}{d+d_y}} \cdot (\log N)^{b+1} \right), \tag{65}$$

*where $\delta \leq k_1 \leq K$ and $0 \leq k_2 \leq K$. The hyperparameters in the ReLU neural network class $\mathcal{F}$ satisfy*

$$M_k = \mathcal{O}\left(\sqrt{\log N}/\sigma_k\right), \; W = \mathcal{O}\left(N \log^7 N\right),$$

$$\kappa = \exp\left(\mathcal{O}(\log^4 N)\right), \; L = \mathcal{O}(\log^4 N), \; P = \mathcal{O}\left(N \log^9 N\right).$$

The proof of Lemma 9 is provided in Appendix B.3.6. According to Lemma 9, we have:

**Lemma 10** *Suppose that we configure the network parameters as Lemma 9*

$$M_k = \mathcal{O}\left(\sqrt{\log N}/\sigma_t\right), \; W = \mathcal{O}\left(N \log^7 N\right),$$

$$\kappa = \exp\left(\mathcal{O}(\log^4 N)\right), \; L = \mathcal{O}(\log^4 N), \; P = \mathcal{O}\left(N \log^9 N\right).$$

*We denote $m_k = M_k/\sqrt{\log N}$. Then for any $\mathbf{s} \in \mathcal{F}(M_k, W, \kappa, L, P)$ and $(\mathbf{x}, \mathbf{y}) \in \mathcal{D}$, we have $|\ell(\mathbf{s}, \mathbf{x}, \mathbf{y})| \lesssim \int_{k_0}^{K} m_k^2 dk \triangleq M$. In particular, if we take $k_0 = n^{-\mathcal{O}(1)}$ and $K = \mathcal{O}(\log n)$, we have $M = \mathcal{O}(\log k_0)$ for $m_k = \frac{1}{\sigma_k}, \delta \geq k_0$, and $M = O\left(\frac{1}{k_0}\right)$ for $m_k = \frac{1}{\sigma_k^2}$, respectively.*

The proof of the lemma is provided in Appendix B.3.8. Moreover, to convert our approximation guarantee to statistical theory, we need to calculate the covering number of the loss function class $\mathcal{S}(R)$, which is defined as follows.

**Definition 6** *We denote $\mathcal{N}(\varrho, \mathcal{F}, \|\cdot\|)$ to be the $\varrho-$covering number of any function class $\mathcal{F}$ w.r.t. the norm $\|\cdot\|$, i.e.,*

$$\mathcal{N}(\varrho, \mathcal{F}, \|\cdot\|) = \min\left\{N : \exists \{f_i\}_{t=1}^n \subseteq \mathcal{F}, s.t. \; \forall f \in \mathcal{F}, \exists i \in [N], \; \|f_i - f\| \leq \varrho\right\}.$$

The following lemma presents the covering number of $\mathcal{S}(R)$:

**Lemma 11** *Given $\varrho > 0$, when $\|\mathbf{x}\|_\infty \leq R$, the $\varrho-$covering number of the loss function class $\mathcal{S}(R)$ w.r.t. $\|\cdot\|_{L_\infty \mathcal{D}}$ satisfies*

$$\mathcal{N}\left(\varrho, \mathcal{S}(R), \|\cdot\|_{L_\infty \mathcal{D}}\right) \lesssim \left(\frac{2L^2(W \max(R, K) + 2)\kappa^L W^{L+1} \log N}{\varrho}\right)^{2P}. \tag{66}$$

*Here the norm $\|\cdot\|_{L_\infty \mathcal{D}}$ is defined as*

$$\|f(\cdot, \cdot)\|_{L_\infty \mathcal{D}} = \max_{\mathbf{x} \in [-R,R]^d, \mathbf{y} \in [0,1]^{d_y} \cup \{\emptyset\}} |f(\mathbf{x}, \mathbf{y})|.$$

The proof is provided in Appendix B.3.9. Particularly, under the network configuration in Lemma 9, we know that log covering number is bounded by

$$\log \mathcal{N} \lesssim N \log^9 N \left(\text{Poly}(\log \log N) + \text{Poly}(\log \log N) \log N \log R + \log^8 N + \log \frac{1}{\varrho}\right)$$

$$\lesssim N \log^9 N \left(\log^8 N + \log^2 N \log R + \log \frac{1}{\varrho}\right). \tag{67}$$

With the Lemmas 9, 10, and 11 introduced above, we now begin our proof of Lemma 7. We denote the true score by $\varphi^\star(\mathbf{x}, \mathbf{y}, k) = \nabla \log p(\mathbf{x}^k|\mathbf{y})$ if $\mathbf{y} \neq \emptyset$ and $\varphi^\star(\mathbf{x}, \emptyset, k) = \nabla \log p(\mathbf{x}^k)$. We create $n$ number of i.i.d ghost samples, as given by

$$(\mathbf{x}_1', \mathbf{y}_1'), (\mathbf{x}_2', \mathbf{y}_2'), ..., (\mathbf{x}_n', \mathbf{y}_n') \sim p_{\text{data}}(\mathbf{x}^\delta, \mathbf{y}).$$

Since $\mathcal{R}_\star(\varphi^\star) = 0$ and $\mathcal{R}_\star(\varphi)$ differs $\ell(\varphi)$ by a constant for any $\varphi$, it suffices to bound

$$\mathcal{R}_\star(\hat{\varphi}) = \mathcal{R}_\star(\hat{\varphi}) - \mathcal{R}_\star(\varphi^\star) = \ell(\hat{\varphi}) - \ell(\varphi^\star) = \mathbb{E}_{\{\mathbf{x}_i', \mathbf{y}_i'\}_{t=1}^n} \left[\frac{1}{n} \sum_{t=1}^{n} (\ell(\mathbf{x}_t', \mathbf{y}_t'; \hat{\varphi}) - \ell(\mathbf{x}_t', \mathbf{y}_t'; \varphi^\star))\right]. \tag{68}$$

Define

$$\ell_1 = \frac{1}{n}\sum_{t=1}^{n}\left(\ell(\mathbf{x}_t, \mathbf{y}_t; \hat{\varphi}) - \ell(\mathbf{x}_t, \mathbf{y}_t; \varphi^\star)\right), \quad \ell_1^{\mathrm{tr}} = \frac{1}{n}\sum_{t=1}^{n}\left(\ell^{\mathrm{tr}}(\mathbf{x}_t, \mathbf{y}_t; \hat{\varphi}) - \ell^{\mathrm{tr}}(\mathbf{x}_t, \mathbf{y}_t; \varphi^\star)\right)$$

and

$$\ell_2 = \frac{1}{n}\sum_{t=1}^{n}\left(\ell(\mathbf{x}_t', \mathbf{y}_t'; \hat{\varphi}) - \ell(\mathbf{x}_t', \mathbf{y}_t'; \varphi^\star)\right), \quad \ell_2^{\mathrm{tr}} = \frac{1}{n}\sum_{t=1}^{n}\left(\ell^{\mathrm{tr}}(\mathbf{x}_t', \mathbf{y}_t'; \hat{\varphi}) - \ell^{\mathrm{tr}}(\mathbf{x}_t', \mathbf{y}_t'; \varphi^\star)\right).$$

We decompose $\mathbb{E}_{\{\mathbf{x}_t, \mathbf{y}_t\}_{t=1}^{n}}[\mathcal{R}_\star(\hat{\varphi})]$ into

$$\mathbb{E}_{\{\mathbf{x}_t, \mathbf{y}_t\}_{t=1}^{n}}[\mathcal{R}_\star(\hat{\varphi})] = \underbrace{\mathbb{E}_{\{\mathbf{x}_t, \mathbf{y}_t\}_{t=1}^{n}}\left[\ell_1^{\mathrm{tr}} - \ell_1\right]}_{B_1} + \underbrace{\mathbb{E}_{\{\mathbf{x}_t, \mathbf{y}_t\}_{t=1}^{n}}\left[\mathbb{E}_{\{\mathbf{x}_t', \mathbf{y}_t'\}_{t=1}^{n}}\left[\ell_2 - \ell_2^{\mathrm{tr}}\right]\right]}_{B_2} \tag{69}$$

$$+ \underbrace{\mathbb{E}_{\{\mathbf{x}_t, \mathbf{y}_t\}_{t=1}^{n}}\left[\mathbb{E}_{\{\mathbf{x}_t', \mathbf{y}_t'\}_{t=1}^{n}}\left[\ell_2^{\mathrm{tr}}\right] - \ell_1^{\mathrm{tr}}\right]}_{C} \tag{70}$$

$$+ \underbrace{\mathbb{E}_{\{\mathbf{x}_t, \mathbf{y}_t\}_{t=1}^{n}}[\ell_1]}_{D}. \tag{71}$$

**Bounding Terms $B_1$ and $B_2$.** Since we have for any $\varphi \in \mathcal{F}$ ($\varphi$ can depend on $\mathbf{x}, \mathbf{y}$),

$$\mathbb{E}_{\mathbf{x}, \mathbf{y}}\left[\left|\ell(\mathbf{x}, \mathbf{y}; \varphi) - \ell^{\mathrm{tr}}(\mathbf{x}, \mathbf{y}; \varphi)\right|\right]$$

$$= \int_{k_0}^{K} \frac{1}{K - k_0} \int_{\mathbf{y}} \int_{\|\mathbf{x}\|>R} \mathbb{E}_{\tau, \mathbf{x}^k|\mathbf{x}^0=\hat{\mathbf{x}}^0}\left[\left\|\varphi(\mathbf{x}^k, \tau\mathbf{y}, k) - \nabla\log\zeta(\mathbf{x}^k|\mathbf{x}^0)\right\|_2^2\right] p(\mathbf{x}|\mathbf{y})p(\mathbf{y})d\mathbf{x}d\mathbf{y}dk$$

$$+ \int_{\delta}^{K} \frac{1}{K - \delta} \int_{\mathbf{y}} \int_{\|\mathbf{x}\|>R} \mathbb{E}_{\tau, \mathbf{x}^k|\mathbf{x}^\delta=\mathbf{x}}\left[\left\|\varphi(\mathbf{x}^k, \tau\mathbf{y}, k) - \nabla\log\zeta(\mathbf{x}^k|\mathbf{x}^\delta)\right\|_2^2\right] p(\mathbf{x}|\mathbf{y})p(\mathbf{y})d\mathbf{x}d\mathbf{y}dk$$

$$\leq 2\int_{k_0}^{K} \frac{1}{K - k_0} \int_{\mathbf{y}} \int_{\|\mathbf{x}\|>R} \mathbb{E}_{\tau, \mathbf{x}^k|\mathbf{x}^0=\hat{\mathbf{x}}^0}\left[\left\|\varphi(\mathbf{x}^k, \tau\mathbf{y}, k)\right\|_2^2 + \left\|\nabla\log\zeta(\mathbf{x}^k|\mathbf{x}^0)\right\|_2^2\right] p(\mathbf{x}|\mathbf{y})p(\mathbf{y})d\mathbf{x}d\mathbf{y}dk$$

$$+ 2\int_{\delta}^{K} \frac{1}{K - \delta} \int_{\mathbf{y}} \int_{\|\mathbf{x}\|>R} \mathbb{E}_{\tau, \mathbf{x}^k|\mathbf{x}^\delta=\mathbf{x}}\left[\left\|\varphi(\mathbf{x}^k, \tau\mathbf{y}, k)\right\|_2^2 + \left\|\nabla\log\zeta(\mathbf{x}^k|\mathbf{x}^\delta)\right\|_2^2\right] p(\mathbf{x}|\mathbf{y})p(\mathbf{y})d\mathbf{x}d\mathbf{y}dk$$

$$\lesssim \int_{k_0}^{K} \frac{1}{\log N} \int_{\|\mathbf{x}\|>R} \mathbb{E}_{\tau, \mathbf{x}^k|\mathbf{x}^0=\hat{\mathbf{x}}^0}\left[m_k^2\log N + \left\|\nabla\log\zeta(\mathbf{x}^k|\mathbf{x}^0)\right\|_2^2\right]\exp(-C_2\|\mathbf{x}\|_2^2/2)d\mathbf{x}dt$$

$$+ \int_{\delta}^{K} \frac{1}{\log N} \int_{\|\mathbf{x}\|>R} \mathbb{E}_{\tau, \mathbf{x}^k|\mathbf{x}^\delta=\mathbf{x}}\left[m_k^2\log N + \left\|\nabla\log\zeta(\mathbf{x}^k|\mathbf{x}^0)\right\|_2^2\right]\exp(-C_2'\|\mathbf{x}\|_2^2/2)d\mathbf{x}dt$$

$$\lesssim \exp\left(-C_2 R^2\right) R \int_{k_0}^{K} m_k^2 dk + \exp\left(-C_2 R^2\right) \int_{k_0}^{K} \frac{1}{\sigma_k^2} dk$$

$$+ \exp\left(-C_2' R^2\right) R \int_{\delta}^{K} m_k^2 dk + \exp\left(-C_2' R^2\right) \int_{\delta}^{K} \frac{1}{\sigma_k^2} dk$$

$$\lesssim \exp\left(-C_2 R^2\right) RM, \tag{72}$$

where the second inequality follows from the sub-Gaussian property of $p(\mathbf{x}|\mathbf{y})$ under Assumption 2, and the third inequality invokes the fact $\mathbb{E}_{\mathbf{x}^k|\mathbf{x}^0=\mathbf{x}}\left[\left\|\nabla\log p(\mathbf{x}^k|\mathbf{x}^0)\right\|_2^2\right] = 1/\sigma_k^2$ and $\mathbb{E}_{\mathbf{x}^k|\mathbf{x}^\delta=\mathbf{x}}\left[\left\|\nabla\log p(\mathbf{x}^k|\mathbf{x}^\delta)\right\|_2^2\right] = 1/\sigma_k^2$. Thus, both terms $B_1$ and $B_2$ are bounded by $\mathcal{O}\left(\exp\left(-C_2 R^2\right) RM\right)$.

**Bounding Term $C$.** For conciseness, we take $\mathbf{z} = (\mathbf{x}, \mathbf{y})$. We denote $\ell^{\mathrm{tr}}(\mathbf{x}, \mathbf{y}; \hat{\varphi})$ as $\hat{\ell}(\mathbf{z})$ and $\ell^{\mathrm{tr}}(\mathbf{x}, \mathbf{y}; \varphi^\star)$ as $\ell^\star(\mathbf{z})$. For $\varrho > 0$ to be chosen later, let $\mathcal{J} = \{\ell_1, \ell_2, ..., \ell_\mathcal{N}\}$ be a $\varrho$-covering of the loss function class $\mathcal{S}(R)$ with the minimum cardinality in the $L^\infty$ metric in the bounded space $\mathcal{D}$, and $J$ be a random variable such that $\left\|\hat{\ell} - \ell_J\right\|_\infty \leq \varrho$. Moreover, we define

$u_j = \max\left\{A, \sqrt{\mathbb{E}_{\mathbf{z}}\left[\ell_j(\mathbf{z}) - \ell^\star(\mathbf{z})\right]}\right\}$, where $\mathbf{z} \sim P_{\mathbf{x},\mathbf{y}}$ is independent of $\{\mathbf{z}_t, \mathbf{z}_t'\}_{t=1}^n$. Besides, we define

$$E = \max_{1 \leq j \leq \mathcal{N}} \left| \sum_{t=1}^n \frac{(\ell_j(\mathbf{z}_t) - \ell^\star(\mathbf{z}_t)) - (\ell_j(\mathbf{z}_t') - \ell^\star(\mathbf{z}_t'))}{u_j} \right|.$$

Then we can further bound term $C$ as follows:

$$|C| = \left| \mathbb{E}_{\{\mathbf{z}_i\}_{t=1}^n} \left[ \frac{1}{n} \sum_{t=1}^n \left( \hat{\ell}(\mathbf{z}_t) - \ell^\star(\mathbf{z}_t) \right) - \mathbb{E}_{\{\mathbf{z}_i'\}_{t=1}^n} \left[ \sum_{t=1}^n \left( \hat{\ell}(\mathbf{z}_t') - \ell^\star(\mathbf{z}_t') \right) \right] \right] \right|$$

$$= \left| \frac{1}{n} \mathbb{E}_{\{\mathbf{z}_t, \mathbf{z}_t'\}_{t=1}^n} \left[ \sum_{i=1}^n \left( \left( \hat{\ell}(\mathbf{z}_t) - \ell^\star(\mathbf{z}_t) \right) - \left( \hat{\ell}(\mathbf{z}_t') - \ell^\star(\mathbf{z}_t') \right) \right) \right] \right|$$

$$\leq \left| \frac{1}{n} \mathbb{E}_{\{\mathbf{z}_t, \mathbf{z}_t'\}_{t=1}^n} \left[ \sum_{i=1}^n \left( (\ell_J(\mathbf{z}_t) - \ell^\star(\mathbf{z}_t)) - (\ell_J(\mathbf{z}_t') - \ell^\star(\mathbf{z}_t')) \right) \right] \right| + 2\varrho$$

$$\leq \frac{1}{n} \mathbb{E}_{\{\mathbf{z}_t, \mathbf{z}_t'\}_{t=1}^n} \left[ u_J E \right] + 2\varrho$$

$$\leq \frac{1}{2} \mathbb{E}_{\{\mathbf{z}_t, \mathbf{z}_t'\}_{t=1}^n} \left[ u_J^2 \right] + \frac{1}{2n^2} \mathbb{E}_{\{\mathbf{z}_t, \mathbf{z}_t'\}_{t=1}^n} \left[ E^2 \right] + 2\varrho. \tag{73}$$

Denote $h_j(\mathbf{z}) = \ell_j(\mathbf{z}) - \ell^\star(\mathbf{z})$ and $\hat{h}(\mathbf{z}) = \hat{\ell}(\mathbf{z}) - \ell^\star(\mathbf{z})$. Moreover, we define the truncated population loss as $\mathcal{R}_\star^{\text{tr}}(\hat{\varphi}) = \mathbb{E}_{\mathbf{z}}\left[\hat{h}\right]$, and define the truncated empirical loss as $\hat{\mathcal{R}}_\star^{\text{tr}}(\hat{\varphi}) = \frac{1}{n} \sum_{t=1}^n \hat{h}(\mathbf{z}_t)$. By (72) we know that $|\mathcal{R}_\star^{\text{tr}}(\hat{\varphi}) - \mathcal{R}_\star(\hat{\varphi})| \lesssim \exp\left(-C_2 R^2\right) RM$. Now we bound $\mathbb{E}_{\{\mathbf{z}_t, \mathbf{z}_t'\}_{t=1}^n}\left[u_J^2\right]$ and $\mathbb{E}_{\{\mathbf{z}_t, \mathbf{z}_t'\}_{t=1}^n}\left[E^2\right]$ separately.

By the definition of $u_J$, we have

$$\mathbb{E}_{\{\mathbf{z}_t, \mathbf{z}_t'\}_{t=1}^n} \left[ u_J^2 \right] \leq A^2 + \mathbb{E}_{\{\mathbf{z}_t, \mathbf{z}_t'\}_{t=1}^n} \left[ \mathbb{E}_{\mathbf{z}}\left[h_J(\mathbf{z})\right] \right]$$

$$\leq A^2 + \mathbb{E}_{\{\mathbf{z}_t, \mathbf{z}_t'\}_{t=1}^n} \left[ \mathbb{E}_{\mathbf{z}}\left[\hat{h}(\mathbf{z})\right] \right] + 2\varrho$$

$$= A^2 + \mathbb{E}_{\{\mathbf{z}_t, \mathbf{z}_t'\}_{t=1}^n} \left[ \mathcal{R}_\star^{\text{tr}}(\hat{\varphi}) \right] + 2\varrho. \tag{74}$$

**Bounding term** $\mathbb{E}_{\{\mathbf{z}_t, \mathbf{z}_t'\}_{t=1}^n}\left[E^2\right]$. Denote $g_j = \sum_{t=1}^n \frac{h_j(\mathbf{z}_t) - h_j(\mathbf{z}_t')}{u_j}$. It is easy to observe that $\mathbb{E}_{\mathbf{z}_t, \mathbf{z}_t'}\left[\frac{h_j(\mathbf{z}_t) - h_j(\mathbf{z}_t')}{u_j}\right] = 0$ for any $t, j$. By independence of $\{g_j\}_{j=1}^\mathcal{N}$, we have

$$\mathbb{E}_{\{\mathbf{z}_t, \mathbf{z}_t'\}_{t=1}^n} \left[ \sum_{t=1}^n \left( \frac{h_j(\mathbf{z}_t) - h_j(\mathbf{z}_t')}{u_j} \right)^2 \right] \leq \sum_{t=1}^n \mathbb{E}_{\mathbf{z}_t, \mathbf{z}_t'} \left[ \left( \frac{h_j(\mathbf{z}_t)}{u_j} \right)^2 + \left( \frac{h_j(\mathbf{z}_t')}{u_j} \right)^2 \right]$$

$$\leq M \sum_{t=1}^n \mathbb{E}_{\mathbf{z}_t, \mathbf{z}_t'} \left[ \frac{h_j(\mathbf{z}_t)}{u_j^2} + \frac{h_j(\mathbf{z}_t')}{u_j^2} \right]$$

$$\leq 2nM.$$

Since $\left| \frac{h_j(\mathbf{z}_t) - h_j(\mathbf{z}_t')}{u_j} \right| \leq \frac{M}{A}$ and $g_j$ is centered, by Bernstein's Inequality, we have: $\forall j$, there exists

$$\Pr\left[g_j^2 \geq h\right] = 2\Pr\left[ \sum_{t=1}^n \frac{h_j(\mathbf{z}_t) - h_j(\mathbf{z}_t')}{u_j} \geq \sqrt{h} \right] \leq 2\exp\left( -\frac{h/2}{M(2n + \frac{\sqrt{h}}{3A})} \right).$$

Thus, we have

$$\Pr\left[E^2 \geq h\right] \leq \sum_{j=1}^\mathcal{N} \Pr\left[g_j^2 \geq h\right] \leq 2\mathcal{N} \exp\left( -\frac{h/2}{M(2n + \frac{\sqrt{h}}{3A})} \right).$$

Thus, $\forall h_0 > 0$, there is

$$\mathbb{E}_{\{\mathbf{z}_t, \mathbf{z}_t'\}_{t=1}^n} \left[ E^2 \right] = \int_0^{h_0} \Pr\left[ E^2 \geq h \right] dh + \int_{h_0}^{\infty} \Pr\left[ E^2 \geq h \right] dh$$

$$\leq h_0 + \int_{h_0}^{\infty} 2\mathcal{N} \exp\left( -\frac{h/2}{M(2n + \frac{\sqrt{h}}{3A})} \right) dh$$

$$\leq h_0 + 2\mathcal{N} \int_{h_0}^{\infty} \left[ \exp\left( -\frac{h}{8Mn} \right) + \exp\left( -\frac{3A\sqrt{h}}{4M} \right) \right] dh$$

$$\leq h_0 + 2\mathcal{N} \left[ 8Mn \exp\left( -\frac{h_0}{8Mn} \right) + \left( \frac{8M\sqrt{h_0}}{3A} + \frac{32M}{9A^2} \right) \exp\left( -\frac{3A\sqrt{h_0}}{4M} \right) \right].$$

Taking $A = \sqrt{h_0}/6n$ and $h_0 = 8Mn \log \mathcal{N}$, we have

$$\mathbb{E}_{\{\mathbf{z}_t, \mathbf{z}_t'\}_{t=1}^n} \left[ E^2 \right] \leq 8Mn \log \mathcal{N} + 2\left( 8Mn + 16Mn + \frac{16}{\log \mathcal{N}} \right)$$

$$\lesssim Mn \log \mathcal{N}. \tag{75}$$

By applying the bounds (74), (75) to (73), we obtain that

$$\left| \mathbb{E}_{\{\mathbf{z}_t\}_{t=1}^n} \left[ \hat{\mathcal{R}}_\star^{\mathrm{tr}}(\hat{\varphi}) - \mathcal{R}_\star^{\mathrm{tr}}(\hat{\varphi}) \right] \right| \lesssim \frac{1}{2}\left( A^2 + \mathbb{E}_{\{\mathbf{z}_t, \mathbf{z}_t'\}_{t=1}^n} \left[ \mathcal{R}_\star^{\mathrm{tr}}(\hat{\varphi}) + 2\varrho \right) + \frac{M}{n} \log \mathcal{N} + 2\varrho$$

$$= \frac{1}{2} \mathbb{E}_{\{\mathbf{z}_t\}_{t=1}^n} \left[ \mathcal{R}_\star^{\mathrm{tr}}(\hat{\varphi}) \right] + \frac{M}{n} \log \mathcal{N} + \frac{7}{2}\varrho.$$

Thus, we have

$$\mathbb{E}_{\{\mathbf{z}_t\}_{t=1}^n} \left[ \mathcal{R}_\star^{\mathrm{tr}}(\hat{\varphi}) \right] \lesssim 2\mathbb{E}_{\{\mathbf{z}_t\}_{t=1}^n} \left[ \hat{\mathcal{R}}_\star^{\mathrm{tr}}(\hat{\varphi}) \right] + \frac{M}{n} \log \mathcal{N} + 7\delta, \tag{76}$$

which means that

$$C \lesssim \mathbb{E}_{\{\mathbf{x}_t, \mathbf{y}_t\}_{t=1}^n} \left[ \ell_1^{\mathrm{tr}} \right] + \frac{M}{n} \log \mathcal{N} + 7\delta$$

$$\leq \mathbb{E}_{\{\mathbf{x}_t, \mathbf{y}_t\}_{t=1}^n} \left[ \ell_1 \right] + |A_1| + \frac{M}{n} \log \mathcal{N} + 7\delta$$

$$\lesssim D + \exp\left( -C_2 R^2 \right) RM + \frac{M}{n} \log \mathcal{N} + 7\delta.$$

**Bounding Term $D$**  For any $\varphi$, define $\hat{\mathcal{R}}_\star(\varphi) = \hat{\ell}(\varphi) - \hat{\ell}(\varphi^\star)$. Then we have $\ell_1 = \hat{\mathcal{R}}_\star(\hat{\mathbf{s}})$. Since $\hat{\mathbf{s}}$ minimizes $\hat{\ell}$, we obtain that

$$\hat{\mathcal{R}}_\star(\hat{\varphi}) = \hat{\ell}(\hat{\mathbf{s}}) - \hat{\ell}(\varphi^\star) \leq \hat{\ell}(\varphi) - \hat{\ell}(\varphi^\star) = \hat{\mathcal{R}}_\star(\varphi).$$

Thus, we have

$$D = \mathbb{E}_{\{\mathbf{z}_t\}_{t=1}^n} \left[ \hat{\mathcal{R}}_\star(\hat{\varphi}) \right] \leq \mathbb{E}_{\{\mathbf{z}_t\}_{t=1}^n} \left[ \hat{\mathcal{R}}_\star(\varphi) \right] = \mathcal{R}_\star(\varphi).$$

By taking minimum w.r.t. $\zeta \in \mathcal{F}$, we have $D \leq \min_{\mathbf{s} \in \mathcal{F}} \mathcal{R}_\star(\varphi)$.

**Balancing the error**  Now, combining the bounds for term $B_1$, $B_2$, $C$, and $D$ and plugging the log covering number (67), we have

$$\mathbb{E}_{\{\mathbf{z}_t\}_{t=1}^n} \left[ \mathcal{R}_\star(\hat{\varphi}) \right] \leq 2 \min_{\mathbf{s} \in \mathcal{F}} \int_{k_0}^K \frac{1}{K - k_0} \mathbb{E}_{\tau, \mathbf{x}^k, \mathbf{y}} \left\| \varphi(\mathbf{x}^k, \tau\mathbf{y}, k) - \nabla \log p(\mathbf{x}^k | \tau\mathbf{y}) \right\|_2^2 dk$$

$$+ 2 \min_{\mathbf{s} \in \mathcal{F}} \int_{\delta}^K \frac{1}{K - \delta} \mathbb{E}_{\tau, \mathbf{x}^k, \mathbf{y}} \left\| \varphi(\mathbf{x}^k, \tau\mathbf{y}, k) - \nabla \log p(\mathbf{x}^k | \tau\mathbf{y}) \right\|_2^2 dk$$

$$+ O\left( \frac{M}{n} N^{d + d_y} \log^9 N \left( \log^8 N + \log^2 N \log R + \log \frac{1}{\varrho} \right) \right)$$

$$+ O\left( \exp\left( -C_2 R^2 \right) RM \right) + 7\varrho. \tag{77}$$

By taking $R = \sqrt{\frac{(C_\sigma + 2b)\log N}{C_2(d+d_y)}}$ and $\varrho = N^{-2b/(d+d_y)}$ and under Assumption 2, we have

$$
\begin{aligned}
\mathbb{E}_{\{\mathbf{z}_t\}_{t=1}^n} [\mathcal{R}_\star(\hat{\varphi})] \leq & 2\min_{\mathbf{s}\in\mathcal{F}} \int_{k_0}^K \frac{1}{K-k_0} \mathbb{E}_{\tau,\mathbf{x}^k,\mathbf{y}} \left\| \varphi(\mathbf{x}^k, \tau\mathbf{y}, k) - \nabla \log p(\mathbf{x}^k|\tau\mathbf{y}) \right\|_2^2 dk \\
& + 2\min_{\mathbf{s}\in\mathcal{F}} \int_\delta^K \frac{1}{K-\delta} \mathbb{E}_{\tau,\mathbf{x}^k,\mathbf{y}} \left\| \varphi(\mathbf{x}^k, \tau\mathbf{y}, k) - \nabla \log p(\mathbf{x}^k|\tau\mathbf{y}) \right\|_2^2 dk \\
& + O\left( \frac{M}{n} N \log^{17} N \right) + O\left( M N^{-2b-C_\sigma} \right) \\
\leq & 2\min_{\mathbf{s}\in\mathcal{F}} \int_{k_0}^K \frac{1}{K-k_0} \mathbb{E}_{\tau,\mathbf{y}} \left[ \mathbb{E}_{\mathbf{x}^k} \left\| \varphi(\mathbf{x}^k, \tau\mathbf{y}, k) - \nabla \log p(\mathbf{x}^k|\tau\mathbf{y}) \right\|_2^2 \right] dk \\
& + 2\min_{\mathbf{s}\in\mathcal{F}} \int_\delta^K \frac{1}{K-\delta} \mathbb{E}_{\tau,\mathbf{y}} \left[ \mathbb{E}_{\mathbf{x}^k} \left\| \varphi(\mathbf{x}^k, \tau\mathbf{y}, k) - \nabla \log p(\mathbf{x}^k|\tau\mathbf{y}) \right\|_2^2 \right] dk \\
& + O\left( \frac{M}{n} N \log^{17} N \right) + O\left( N^{-\frac{2b}{d+d_y}} \right). \quad (78)
\end{aligned}
$$

We invoke the inequality $M \lesssim \frac{1}{\delta} \leq \frac{1}{k_0} = N^{C_\sigma}$ for the second inequality of (78). Recall that for any $k > 0$ and score approximator $\zeta(\cdot, \cdot, k)$, we have

$$
\begin{aligned}
\mathbb{E}_{\tau,\mathbf{x}^k,\mathbf{y}} \left\| \varphi(\mathbf{x}^k, \tau\mathbf{y}, k) - \nabla \log p(\mathbf{x}^k|\tau\mathbf{y}) \right\|_2^2 = & \frac{1}{2} \int_{\mathbb{R}^d} \left\| \varphi(\mathbf{x}, \varnothing, k) - \nabla \log p(\mathbf{x}) \right\|^2 p(\mathbf{x}) d\mathbf{x} \\
& + \frac{1}{2} \mathbb{E}_{\mathbf{y}} \left[ \int_{\mathbb{R}^d} \left\| \varphi(\mathbf{x}, \mathbf{y}, k) - \nabla \log p(\mathbf{x}|\mathbf{y}) \right\|^2 p(\mathbf{x}|\mathbf{y}) d\mathbf{x} \right].
\end{aligned}
$$

Therefore, we can invoke the score approximation error guarantee in Lemma 9 and Assumption 2 to bound the score estimation error. Particularly, under Assumption 2, we have $M = \mathcal{O}(1/k_0)$. By taking $N = n^{(d+d_y)/(d+d_y+b)}$ and invoking Lemma 9, the error is bounded by

$$
\mathbb{E}_{\{\mathbf{z}_t\}_{t=1}^n} [\mathcal{R}(\hat{\varphi})] \leq 2\mathbb{E}_{\{\mathbf{z}_t\}_{t=1}^n} [\mathcal{R}_\star(\hat{\varphi})] \lesssim \frac{1}{t_0} n^{-\frac{b}{d+d_y+b}} \log^{\max(17, d+b/2+1)} n. \quad (79)
$$

Similarly, under Assumption 2, we have $M = \mathcal{O}(\log \frac{1}{k_0})$. By taking $N = n^{(d+d_y)/(d+d_y+2b)}$ and invoking Lemma 9, the conditional score error is bounded by

$$
\mathbb{E}_{\{\mathbf{z}_t\}_{t=1}^n} [\mathcal{R}(\hat{\varphi})] \lesssim \log \frac{1}{t_0} n^{-\frac{2b}{d+d_y+2b}} \log^{\max(17, (b+1)/2)} n. \quad (80)
$$

We complete our proof.

### B.3.6 PROOF OF LEMMA 9

We start with the following assumption:

**Assumption 4** *Let $C$ and $C_2$ be two positive constants and function $f \in \mathcal{H}^b(\mathbb{R}^d \times [0,1]^{d_y}, B)$ for a constant radius $B$. We assume $f(\mathbf{x}, \mathbf{y}) \geq C$ for all $(\mathbf{x}, \mathbf{y})$ and the conditional density function $p(\mathbf{x}|\mathbf{y}) = \exp(-C_2 \|\mathbf{x}\|_2^2 /2) \cdot f(\mathbf{x}, \mathbf{y})$.*

Under Assumption 4, we have the following lemma paraphrased from Fu et al. (2024).

**Lemma 12 (Fu et al. (2024))** *For sufficiently large $N$ and constants $C_\sigma, C_\alpha > 0$, by taking early-stopping step $K_0 = N^{-C_\sigma}$ and terminal step $K = C_\alpha \log N$, there exists $\mathbf{s} \in \mathcal{F}(M_t, W, \kappa, L, P)$ such that for all $\mathbf{y} \in [0,1]^{d_y}$ and $k \in [K_0, K]$, it holds that*

$$
\int_{\mathbb{R}^d} \left\| \varphi(\mathbf{x}, \mathbf{y}, k) - \nabla \log p_k(\mathbf{x}|\mathbf{y}) \right\|_2^2 \cdot p_k(\mathbf{x}|\mathbf{y}) d\mathbf{x} = \mathcal{O}\left( \frac{B^2}{\sigma_k^2} \cdot N^{-\frac{2b}{d+d_y}} \cdot (\log N)^{b+1} \right).
$$

*The hyperparameters in the ReLU neural network class $\mathcal{F}$ satisfy*

$$
M_t = \mathcal{O}\left( \sqrt{\log N}/\sigma_t \right), \ W = \mathcal{O}\left( N \log^7 N \right),
$$

$$
\kappa = \exp\left( \mathcal{O}(\log^4 N) \right), \ L = \mathcal{O}(\log^4 N), \ P = \mathcal{O}\left( N \log^9 N \right).
$$

Based on the Assumptions 2, 4 and Lemma 12, the proposed loss function can be divided into the following two parts:

$$l_1 = \int_{\mathbf{x}^k} \left\| \varphi(\mathbf{x}^k, \mathbf{y}, k) - \nabla \log p_k(\mathbf{x}^k|\mathbf{y}) \right\|_2^2 \cdot p_k(\mathbf{x}^k|\mathbf{y}) d\mathbf{x}^k \quad \text{for all } \delta \leq k \leq K, \tag{81a}$$

$$l_2 = \int_{\mathbf{x}^k} \left\| \varphi(\mathbf{x}^k, \mathbf{y}, k) - \nabla \log p_k(\mathbf{x}^k|\mathbf{y}) \right\|_2^2 \cdot p_k(\mathbf{x}^k|\mathbf{y}) d\mathbf{x}^k \quad \text{for all } 0 \leq k \leq K. \tag{81b}$$

where $\mathbf{x}^\delta \sim p_{\text{data}}(\mathbf{x}^\delta|\mathbf{y})$ and $\mathbf{x}^0 = \frac{1}{\sqrt{\bar{\alpha}_\delta}} \left( \mathbf{x}^\delta - \sqrt{1 - \bar{\alpha}_\delta}\epsilon \right)$. Next, we analyze $l_1$ and $l_2$ separately.

**Part 1: Proof of $l_1$**   For $l_1$, each conditional distribution in the forward process satisfies Assumption 4, leading to the following lemma. The detailed proof can be found in Appendix B.3.7.

**Lemma 13** *Under Assumption 2, for any $k \geq 0$, let $C$ and $C_3$ be two positive constants and function $f \in \mathcal{H}^b(\mathbb{R}^d \times [0,1]^{d_y}, B)$ for a constant radius $B$. We assume $f(\mathbf{x}^k, \mathbf{y}) \geq C$ for all $(\mathbf{x}^k, \mathbf{y})$ and the conditional density function $p(\mathbf{x}^k|\mathbf{y}) = \exp(-C_3 \left\| \mathbf{x}^k \right\|_2^2 /2) \cdot f(\mathbf{x}^k, \mathbf{y})$, where*

$$p(\mathbf{x}^k|\mathbf{y}) = \int_{\mathbb{R}^d} p(\mathbf{x}_{\text{true}}|\mathbf{y}) \frac{1}{\sigma_k^d (2\pi)^{d/2}} \exp\left( -\frac{\left\| \sqrt{\alpha_k}\mathbf{x}_{\text{true}} - \mathbf{x}^k \right\|^2}{2\sigma_k^2} \right) d\mathbf{x}_{\text{true}}.$$

According to Lemma 13, Assumption 4 holds for any $k \geq 0$. Therefore, based on Lemma 12, by replacing $K_0$ with $\delta$ in (81a), we can derive the following corollary:

**Corollary 2** *Suppose Assumption 2 holds. For sufficiently large $N$ and constant $C_\alpha > 0$, by taking terminal step $K = C_\alpha \log N$, there exists $\mathbf{s} \in \mathcal{F}(M_t, W, \kappa, L, P)$ such that for all $\mathbf{y} \in [0,1]^{d_y}$ and $k \in [\delta, K]$, it holds that*

$$l_1 = \mathcal{O}\left( \frac{B^2}{\sigma_k^2} \cdot N^{-\frac{2b}{d+d_y}} \cdot (\log N)^{b+1} \right). \tag{82}$$

*The hyperparameters in the ReLU neural network class $\mathcal{F}$ satisfy*

$$M_t = \mathcal{O}\left( \sqrt{\log N}/\sigma_t \right), \ W = \mathcal{O}\left( N \log^7 N \right),$$

$$\kappa = \exp\left( \mathcal{O}(\log^4 N) \right), \ L = \mathcal{O}(\log^4 N), \ P = \mathcal{O}\left( N \log^9 N \right).$$

**Part 2: Proof of $l_2$**   Based on the diffusion model, we have $\mathbf{x}^0 = \frac{1}{\sqrt{\bar{\alpha}_\delta}} \left( \mathbf{x}^\delta - \sqrt{1 - \bar{\alpha}_\delta}\epsilon \right)$, where $\epsilon$ follows a standard normal distribution. Thus, we only need to prove that $p(\mathbf{x}^0|\mathbf{y})$ satisfies Assumption 4. Specifically, we can derive:

$$\epsilon = \frac{\mathbf{x}^\delta - \sqrt{\bar{\alpha}_\delta}\mathbf{x}^0}{\sqrt{1 - \bar{\alpha}_\delta}}, \tag{83}$$

where

$$p(\epsilon) = \frac{1}{(2\pi)^{d/2}} \exp\left( -\frac{\|\epsilon\|_2^2}{2} \right). \tag{84}$$

By substituting (83) into (84), we obtain

$$p(\epsilon|\mathbf{x}^0, \mathbf{x}^\delta, \mathbf{y}) = \frac{1}{(2\pi)^{d/2}} \exp\left( -\frac{\|\mathbf{x}^\delta - \sqrt{\bar{\alpha}_\delta}\mathbf{x}^0\|_2^2}{2(1 - \bar{\alpha}_\delta)} \right). \tag{85}$$

According to the change of variables formula, there is

$$p(\mathbf{x}^0|\mathbf{x}^\delta, \mathbf{y}) = \frac{1}{\sigma_\delta^d (2\pi)^{d/2}} \exp\left( -\frac{\|\mathbf{x}^0 - \mathbf{x}^\delta/\sqrt{\bar{\alpha}_\delta}\|_2^2}{2\sigma_\delta^2} \right). \tag{86}$$

Therefore, $p(\mathbf{x}^0|\mathbf{x}^\delta, \mathbf{y})$ follows a normal distribution with mean $\mathbf{x}^\delta/\sqrt{\bar{\alpha}_\delta}$ and covariance $\sigma_\delta \mathbf{I}$, where $\sigma_\delta = \sqrt{(1 - \bar{\alpha}_\delta)/(\bar{\alpha}_\delta)}$.

From (86), it readily follows that

$$
\begin{aligned}
p(\mathbf{x}^0|\mathbf{y}) &= \int_{\mathbb{R}^d} p(\mathbf{x}^0|\mathbf{x}^\delta, \mathbf{y}) p(\mathbf{x}^\delta|\mathbf{y}) d\mathbf{x}^\delta \\
&= \int_{\mathbb{R}^d} p(\mathbf{x}^\delta|\mathbf{y}) \frac{1}{\sigma_\delta^d (2\pi)^{d/2}} \exp\left( -\frac{\|\mathbf{x}^0 - \mathbf{x}^\delta/\sqrt{\bar{\alpha}_\delta}\|_2^2}{2\sigma_\delta^2} \right) d\mathbf{x}^\delta.
\end{aligned}
\tag{87}
$$

As a result, $p(\mathbf{x}^0|\mathbf{y})$ satisfies Assumption 4. Then, assuming $C_\sigma$ in Lemma (12) is sufficiently large, we can replacing $K_0$ with 0 and obtain the final conclusion, that is:

**Corollary 3** *Suppose Assumption 2 holds. For sufficiently large $N$ and constant $C_\alpha > 0$, by taking terminal step $K = C_\alpha \log N$, there exists $\mathbf{s} \in \mathcal{F}(M_t, W, \kappa, L, P)$ such that for all $\mathbf{y} \in [0,1]^{d_y}$ and $k \in [0, K]$, it holds that*

$$
l_2 = \mathcal{O}\left( \frac{B^2}{\sigma_k^2} \cdot N^{-\frac{2b}{d+d_y}} \cdot (\log N)^{b+1} \right).
\tag{88}
$$

*The hyperparameters in the ReLU neural network class $\mathcal{F}$ satisfy*

$$
M_t = \mathcal{O}\left( \sqrt{\log N}/\sigma_t \right), \; W = \mathcal{O}\left( N \log^7 N \right),
$$
$$
\kappa = \exp\left( \mathcal{O}(\log^4 N) \right), \; L = \mathcal{O}(\log^4 N), \; P = \mathcal{O}\left( N \log^9 N \right).
$$

**Part 3: Summing $l_1$ and $l_2$**   We obtain the final neural network approximation error by summing (82) and (88). Lemma 9 is proved.

### B.3.7   PROOF OF LEMMA 13

Under Assumption 1, we have

$$
p_t(\mathbf{x}^k|\mathbf{y})
$$
$$
= \int_{\mathbb{R}^d} p(\mathbf{x}_{\text{true}}|\mathbf{y}) \frac{1}{\sigma_k^d (2\pi)^{d/2}} \exp\left( -\frac{\left\| \sqrt{\alpha_k}\mathbf{x}_{\text{true}} - \mathbf{x}^k \right\|^2}{2\sigma_k^2} \right) d\mathbf{x}_{\text{true}}
$$
$$
= \int_{\mathbb{R}^d} \exp(-\frac{C_2 \|\mathbf{x}_{\text{true}}\|_2^2}{2}) \cdot f(\mathbf{x}_{\text{true}}, \mathbf{y}) \frac{1}{\sigma_k^d (2\pi)^{d/2}} \exp\left( -\frac{\left\| \sqrt{\alpha_k}\mathbf{x}_{\text{true}} - \mathbf{x}^k \right\|^2}{2\sigma_k^2} \right) d\mathbf{x}_{\text{true}}
$$
$$
= \int_{\mathbb{R}^d} f(\mathbf{x}_{\text{true}}, \mathbf{y}) \frac{1}{\sigma_k^d (2\pi)^{d/2}} \exp\left( -\frac{\left\| \sqrt{\alpha_k}\mathbf{x}_{\text{true}} - \mathbf{x}^k \right\|^2 + C_2 \sigma_k^2 \|\mathbf{x}_{\text{true}}\|_2^2}{2\sigma_k^2} \right) d\mathbf{x}_{\text{true}}
$$
$$
= \int_{\mathbb{R}^d} f(\mathbf{x}_{\text{true}}, \mathbf{y}) \frac{1}{\sigma_k^d (2\pi)^{d/2}} \exp\left( -\frac{\left\| (\alpha_k + C_2\sigma_k^2)\mathbf{x}_{\text{true}} - \sqrt{\alpha_k}\mathbf{x}^k \right\|^2 + C_2\sigma_k^2 \left\| \mathbf{x}^k \right\|^2}{2\sigma_k^2 (\alpha_k + C_2\sigma_k^2)} \right) d\mathbf{x}_{\text{true}}
$$
$$
= \exp\left( -\frac{C_2 \left\| \mathbf{x}^k \right\|_2^2}{2(\alpha_k + C_2\sigma_k^2)} \right) \int_{\mathbb{R}^d} \frac{f(\mathbf{x}_{\text{true}}, \mathbf{y})}{(2\pi)^{d/2}\sigma_k^d} \exp\left( -\frac{\left\| (\alpha_k + C_2\sigma_k^2)\mathbf{x}_{\text{true}} - \sqrt{\alpha_k}\mathbf{x}^k \right\|^2}{2\sigma_k^2 (\alpha_k + C_2\sigma_k^2)} \right) d\mathbf{x}_{\text{true}}
$$
$$
= \exp\left( -\frac{C_2 \left\| \mathbf{x}^k \right\|_2^2}{2(\alpha_k + C_2\sigma_k^2)} \right) \underbrace{\int_{\mathbb{R}^d} \frac{f(\mathbf{x}_{\text{true}}, \mathbf{y})}{(2\pi)^{d/2}\sigma_k^d} \exp\left( -\frac{\left\| \mathbf{x}_{\text{true}} - \sqrt{\alpha_k}\mathbf{x}^k/(\alpha_k + C_2\sigma_k^2) \right\|^2}{2\sigma_k^2/(\alpha_k + C_2\sigma_k^2)} \right) d\mathbf{x}_{\text{true}}}_{f^k(\mathbf{x}^k, \mathbf{y})} .
\tag{89}
$$

With $f \in \mathcal{H}^b(\mathbb{R}^d \times [0,1]^{d_y}, B)$ and $f(\mathbf{x}_{\text{true}}, \mathbf{y}) \geq C$ in Assumption 2, there exists two constants $B'$ and $C'$, such that $f^k \in \mathcal{H}^b(\mathbb{R}^d \times [0,1]^{d_y}, B')$ and $f(\mathbf{x}^k, \mathbf{y}) \geq C'$ holds. Let $C_3 = C_2/(\alpha_k + C_2\sigma_k^2)$, Lemma 13 is proved.

### B.3.8 PROOF OF LEMMA 10

By the definition of $\ell(\mathbf{x}, \mathbf{y}; \varphi)$, we have: $\forall \mathbf{x}, \mathbf{y}$ and $\mathbf{s} \in \mathcal{F}$

$$
\begin{aligned}
\ell(\mathbf{x}, \mathbf{y}; \varphi) \leq & 2 \int_{k_0}^{T} \frac{1}{K - k_0} \mathbb{E}_{\tau, \mathbf{x}^k | \mathbf{x}^0 = \hat{\mathbf{x}}^0} \left[ \left\| \varphi(\mathbf{x}^k, \tau \mathbf{y}, k) \right\|_2^2 + \left\| \nabla \log p_t(\mathbf{x}^k | \hat{\mathbf{x}}^0) \right\|_2^2 \right] dk \\
& + 2 \int_{\delta}^{T} \frac{1}{K - \delta} \mathbb{E}_{\tau, \mathbf{x}^k | \mathbf{x}^\delta = \mathbf{x}} \left[ \left\| \varphi(\mathbf{x}^k, \tau \mathbf{y}, k) \right\|_2^2 + \left\| \nabla \log p_t(\mathbf{x}^k | \mathbf{x}^\delta) \right\|_2^2 \right] dk \\
\lesssim & \int_{k_0}^{T} \frac{1}{K - k_0} \mathbb{E}_{\tau, \mathbf{x}^k | \mathbf{x}^0 = \hat{\mathbf{x}}^0} \left[ m_t^2 \log N + \left\| \nabla \log p_t(\mathbf{x}^k | \hat{\mathbf{x}}^0) \right\|_2^2 \right] dk \\
& + \int_{\delta}^{T} \frac{1}{K - \delta} \mathbb{E}_{\tau, \mathbf{x}^k | \mathbf{x}^\delta = \mathbf{x}} \left[ m_t^2 \log N + \left\| \nabla \log p_t(\mathbf{x}^k | \mathbf{x}^\delta) \right\|_2^2 \right] dk \\
\lesssim & \int_{k_0}^{K} M_k^2 dk + \int_{k_0}^{K} \frac{1}{K - k_0} \frac{1}{\sigma_k^2} dk + \int_{\delta}^{K} M_k^2 dk + \int_{\delta}^{K} \frac{1}{K - \delta} \frac{1}{\sigma_k^2} dk \\
\lesssim & \int_{k_0}^{K} M_k^2 dk + \int_{\delta}^{K} M_k^2 dk \lesssim \int_{k_0}^{K} M_k^2 dk = M,
\end{aligned}
$$

where we invoke $|\varphi| \lesssim m_k \sqrt{\log N}$ for the second inequality and $1/\sigma_k \lesssim m_k$ for the last inequality.

### B.3.9 PROOF OF LEMMA 11

We first introduce a standard result of bounding the covering number of a ReLU neural network.

**Lemma 14 (Chen et al. (2022), Lemma.7)** *Suppose $\varrho > 0$ and the input $\mathbf{z}$ satisfies $\|\mathbf{z}\|_\infty \leq R$, the $\varrho-$covering number of the neural network class $\mathcal{F}(W, \kappa, L, P)$ w.r.t. $\|\cdot\|_{L_\infty}$ satisfies*

$$
\mathcal{N}\left(\varrho, \mathcal{F}(W, \kappa, L, P), \|\cdot\|_{L_\infty}\right) \leq \left( \frac{2L^2(WR + 2)\kappa^L W^{L+1}}{\varrho} \right)^P. \tag{90}
$$

We remark that our input $(\mathbf{x}, \mathbf{y}, t)$ is uniformly bounded by $\mathcal{O}(\log N)$. Now we begin our proof of Lemma 11. For any two ReLU networks $\varphi_1$ and $\varphi_2$ such that $\|\varphi_1 - \varphi_2\|_{L_\infty \mathcal{D}} \leq \epsilon$, we can bound

the $L_\infty$ error between $\ell(\cdot, \cdot, \varphi_1)$ and $\ell(\cdot, \cdot, \varphi_2)$. For any $(\mathbf{x}, \mathbf{y}) \in \mathcal{D}$, we have

$$|\ell(\mathbf{x}, \mathbf{y}, \varphi_1) - \ell(\mathbf{x}, \mathbf{y}, \varphi_2)|$$

$$\leq \int_{k_0}^K \frac{1}{K-k_0} \mathbb{E}_{\tau, \mathbf{x}^k | \mathbf{x}^0 = \hat{\mathbf{x}}^0} \Big[ \big(\varphi_1(\mathbf{x}^k, \tau\mathbf{y}, k) - \varphi_2(\mathbf{x}^k, \tau\mathbf{y}, k)\big)^\top \big(\varphi_1(\mathbf{x}^k, \tau\mathbf{y}, k)$$
$$+ \varphi_2(\mathbf{x}^k, \tau\mathbf{y}, k) - 2p(\mathbf{x}^k|\hat{\mathbf{x}}^0)\big)\Big] dk$$

$$+ \int_\delta^K \frac{1}{K-\delta} \mathbb{E}_{\tau, \mathbf{x}^k | \mathbf{x}^\delta = \mathbf{x}} \Big[ \big(\varphi_1(\mathbf{x}^k, \tau\mathbf{y}, k) - \varphi_2(\mathbf{x}^k, \tau\mathbf{y}, k)\big)^\top$$
$$\big(\varphi_1(\mathbf{x}^k, \tau\mathbf{y}, k) + \varphi_2(\mathbf{x}^k, \tau\mathbf{y}, k) - 2p(\mathbf{x}^k|\mathbf{x}^\delta)\big)\Big] dk$$

$$\lesssim \epsilon \int_{k_0}^K \frac{1}{K-k_0} \mathbb{E}_{\tau, \mathbf{x}^k | \mathbf{x}^0 = \hat{\mathbf{x}}^0} \big[\|\varphi_1(\mathbf{x}^k, \tau\mathbf{y}, k) + \varphi_2(\mathbf{x}^k, \tau\mathbf{y}, k) - 2p(\mathbf{x}^k|\hat{\mathbf{x}}^0)\|\big] dk$$

$$+ \epsilon \int_\delta^K \frac{1}{K-\delta} \mathbb{E}_{\tau, \mathbf{x}^k | \mathbf{x}^\delta = \mathbf{x}} \big[\|\varphi_1(\mathbf{x}^k, \tau\mathbf{y}, k) + \varphi_2(\mathbf{x}^k, \tau\mathbf{y}, k) - 2p(\mathbf{x}^k|\mathbf{x}^\delta)\|\big] dk$$

$$\lesssim \epsilon \int_{k_0}^K \frac{1}{K-k_0} \mathbb{E}_{\tau, \mathbf{x}^k | \mathbf{x}^0 = \hat{\mathbf{x}}^0} \big[\|m_k \sqrt{\log N} + p(\mathbf{x}^k|\hat{\mathbf{x}}^0)\|\big] dk$$

$$+ \epsilon \int_\delta^K \frac{1}{K-\delta} \mathbb{E}_{\tau, \mathbf{x}^k | \mathbf{x}^\delta = \mathbf{x}} \big[\|m_k \sqrt{\log N} + p(\mathbf{x}^k|\mathbf{x}^\delta)\|\big] dk$$

$$\lesssim \frac{\epsilon}{K-k_0} \left(\sqrt{\log N} \int_{k_0}^K m_k dk + \int_{k_0}^K \frac{1}{\sigma_k} dk\right) + \frac{\epsilon}{K-\delta} \left(\sqrt{\log N} \int_\delta^K m_k dk + \int_\delta^K \frac{1}{\sigma_k} dk\right)$$

$$\lesssim \epsilon \log N, \tag{91}$$

where $\hat{\mathbf{x}}^0 = \frac{1}{\sqrt{\bar{\alpha}_\delta}}\left(\mathbf{x}^\delta - \sqrt{1 - \bar{\alpha}_\delta}\epsilon\right)$ and $\epsilon$ follows a standard normal distribution. For the second inequality, we invoke $|\varphi(\mathbf{x}^k, \tau\mathbf{y}, k)| \leq m_k \sqrt{\log N}$. In the last inequality, we invoke

$$m_k \leq \frac{1}{\sigma_k^2} \leq O\left(\frac{1}{k}\right) \quad \text{when } t = o(1) \text{ and } m_k = \mathcal{O}(1) \text{ when } k \gg 1,$$

and the inequality

$$\frac{1}{K-\delta} \lesssim \frac{1}{\log N} \quad \text{and} \quad \delta \geq k_0.$$

Since $\mathcal{F}$ is a concatenation of two ReLU neural networks of the same size and the domain of the input $\mathbf{z} = (\mathbf{x}, \mathbf{y}, k)$ (or $\mathbf{z} = (\mathbf{x}, k)$ for the unconditional score approximator) satisfies $\|(\mathbf{x}, \mathbf{y}, k)\|_\infty \leq \max(R, K)$, by Lemma 14 we have the covering number of $\mathcal{F}$ bounded as

$$\mathcal{N}\left(\varrho, \mathcal{F}, \|\cdot\|_{L_\infty \mathcal{D}}\right) \lesssim \left(\frac{2L^2(W \max(R, K) + 2)\kappa^L W^{L+1}}{\varrho}\right)^{2P}. \tag{92}$$

Combining this result with (91), we can bound the covering number of $\mathcal{S}(R)$ as

$$\mathcal{N}\left(\varrho, \mathcal{S}(R), \|\cdot\|_{L_\infty \mathcal{D}}\right) \lesssim \left(\frac{2L^2(W \max(R, K) + 2)\kappa^L W^{L+1} \log N}{\varrho}\right)^{2P}. \tag{93}$$

The proof is complete.

# C  ADDITIONAL EXPERIMENTS

In this section, we provide additional details about the experiments, including the introduction of environments and hyper-parameters of all algorithms.

## C.1  DETAILS FOR ENVIRONMENTS

To examine the performance of the proposed algorithm in more challenging control tasks with higher degrees of freedom (DOFs), we evaluated the performance of the proposed algorithm in the OpenAI

Roboschool environments(Brockman et al., 2016). The Roboschool environments include a number of continuous robotic control tasks, such as teaching a multiple-joint robot to walk as fast as possible without falling. The original Roboschool environments are nearly fully observable since observations include the robot's coordinates and (trigonometric functions of) joint angles, as well as (angular and coordinate) velocities.

As in the POMDP classic control tasks, we also performed experiments in the POMDP versions of the Roboschool environments. In the no-velocities (i.e., "-P") cases, velocity information was removed from raw observations; while in the velocities-only (i.e., "-V") cases, only velocity information was retained in raw observations. We summarize key information about each environment in Table 2 with a maximum of 1000 steps.

Table 2: Information of environments in this paper

| Name | Dim of observation space | DOF |
|---|---|---|
| RoboschoolAnt | 28 | 8 |
| RoboschoolAnt-V | 11 | 8 |
| RoboschoolAnt-P | 17 | 8 |
| RoboschoolHopper | 15 | 3 |
| RoboschoolHopper-V | 6 | 3 |
| RoboschoolHopper-P | 9 | 3 |
| RoboschoolWalker2d | 22 | 6 |
| RoboschoolWalker2d-V | 9 | 6 |
| RoboschoolWalker2d-P | 13 | 6 |

## C.2 HYPER-PARAMETERS

In this section, we describe the details of implementing our algorithm as well as its alternatives. Summaries of hyperparameters can be found in Tables 3 and 4.

Table 3: Shared hyperparameters for all algorithms and tasks in this paper

| Hyperparameter | Description | Value |
|---|---|---|
| / | Number of training iterates | 600 |
| $|\mathcal{D}|$ | The size of replay memory | $10^6$ |
| $|\mathcal{B}|$ | The number of samples for each update | 64 |
| $\gamma$ | Discount factor | 0.99 |
| $\tau$ | Fraction of updating the target network per gradient step | 0.005 |
| / | Learning rate for policy and value networks | 0.0003 |
| / | Learning rate for the entropy coefficient in SAC | 0.0003 |
| / | Target entropy in SAC | 0.2 |
| / | MLP layer sizes for policy network | 256,256 |
| / | MLP layer sizes for value network | 256,256 |

Table 4: Hyperparameters for CSR-ADM

| Hyperparameter | Description | Value |
|---|---|---|
| / | Learning rate of asynchronous diffusion model | 0.0003 |
| / | Learning rate of bisimulation metric learning | 0.0003 |
| / | Network for asynchronous diffusion model | UNet |
| / | MLP layer size for bisimulation metric learning | 256,256 |
| $K$ | Total diffusion step | 500 |
| $\beta$ | Beta schedule | linear |
| $\delta$ | noise intensity of observation and reward | 2 |
| / | The Variance of Gaussian noise | 0.5 |