# OpenReview forum: "Provable Causal State Representation under Asynchronous Diffusion Model for POMDPs"
_ICLR.cc/2025/Conference — Submitted to ICLR 2025_

### Official Review · Reviewer_Fvfk · 2024-11-01

**Soundness:** 3
**Presentation:** 3
**Contribution:** 3
**Rating:** 6
**Confidence:** 1

**Summary:**

This work considers the decision-making problem in POMDP with diffusion as an estimation tool. The authors adopt the diffusion model to utilize the causal graph under the POMDP for better value function estimation. They provide the theoretical analysis of the proposed algorithm, and the efficacy of the algorithm is also verified by the experimental results.

**Strengths:**

1.	The authors propose novel methods to make use of the causal structure under the POMDP environment, which achieves better performance in the simulations.
2.	This work contains the solid theoretical guarantee for the proposed methods.

**Weaknesses:**

1.	The implications of the assumptions adopted in this work are not clear. For Assumption 2, it is beneficial to justify when the lower boundness of $f$ holds in the real applications. For Assumption 3, I find that the statement of Definition 2 is not appropriate. When defining a mathematical notion, it is uncommon to state `` following metric exists and is unique’’, which looks like the statement of a theorem. In addition, I suggest the authors to discuss the sufficient conditions of Assumption 3. For example, if the state, action, and observation spaces are finite, will this metric exist and be unique?

2.	It will be helpful to discuss more about the results of diffusion model. In the existing analysis of diffusion models, the distribution estimation error usually consists of initialization error, score estimation error, and the discretization error. But such error decomposition structure is not presented in the current results.

3.	In addition, this is beneficial to discuss and explain each theorem below the statement of the results.

**Questions:**

Same as the weakness

---

> ### Author Response · Authors · 2024-11-22
>
> **Weakness 1:** We would like to express our sincere gratitude for your valuable comments.
>
> In Assumption 2, we assume the existence of a lower bound for the Holder continuous function to ensure the stability and validity of the density function. In practice, this assumption can be satisfied using methods, such as data normalization and function regularization. A similar lower-bound assumption was adopted and discussed in Ref. [1].
>
> Regarding Assumption 3, we apologize for any confusion caused by the definitions in Definition 2 and Assumption 3.
>
> * In Definition 2, we define a metric to measure the difference between the causal state extracted from observations and the true state underlying POMDPs. To clarify this, we have added the phrase "*for any pair of state and observation $\{\mathbf{s}_t \in \mathcal{S}, \mathbf{o}_t \in \mathcal{O}\}$*" to the revised version.
>
> * Assumption 3 refers to a common premise in the study of bisimulation metrics (see Ref. [2]), that is, the existence and uniqueness of the bisimulation metric for any pair of states. In other words, for any pair of states $(\mathbf{s}_i, \mathbf{s}_j)$, there exists a unique bisimulation metric to measure their similarity. In particular, we assume the existence of a unique bisimulation metric for $p$-Wasserstein metrics to facilitate analyzing value function approximation (VFA). In Remark 1, we validate Assumption 3 when $p=1$, with the proof proved in Appendix B 3.1. More general cases with $p>1$ will be studied in our future research, only after which Assumption 3, while it is reasonable and widely adopted, can be stated as an appropriate theorem.
>
> As suggested, we have also clarified in the revised version that Assumption 3 does not restrict the state, action, or observation spaces to be finite (or any other conditions).
>
> **Weakness 2:** We express our heartfelt gratitude for your feedback. We would like to clarify that we account for initialization error, score estimation error, and discretization error in our analysis of the diffusion model (like the existing studies).
>
> Specifically, in Appendix B2.3, we decompose the distribution estimation error into these three components (see Eq. (33)) and provide individual bounds for each of the three components. By appropriately setting the early-stopping step $k_0$ and the total diffusion model's step $K$, we can simplify the results, as presented in Theorem 3.
>
> To address this confusion, we have added the following clarification in the revised version:
>
> "*Under Assumption 2, we can measure the asynchronous diffusion model's distribution estimation by considering the initialization error, score estimation error, and discretization error, and provide the sample complexity bounds for each of the three errors using the Wasserstein-1 distance.*"
>
> **Weakness 3:** Sincere thanks to you for this valuable suggestion. We have added clarifying remarks following each theorem in the revised version, as follows.
>
> * Theorem 1: "*In this sense, the bisimulation metric in (12) represents the upper bound of the value gap.*"
>
> * Theorem 2: "*By Theorem 2, we can quantify the upper bound of the value gap under arbitrary model errors. This can be extended to different probability density estimation models to establish specific convergence properties. The theorem facilitates analyzing the impact of the proposed asynchronous diffusion model on the value gap.*"
>
> * Theorem 3: "*As $n\rightarrow\infty$, the distribution estimation measured by Wasserstein-1 distance converges, i.e., $\mathbb{E}_ {\{{\mathbf{x}_ t,\hat{\mathbf{s}}_ {t}, \mathbf{a}_ t}\}_ {t=1}^{n}} \[W_ 1(p(\mathbf{x}_ t|\hat{\mathbf{s}}_ {t}, \mathbf{a}_ t), \hat{p}({\mathbf{x}}_ t^{k_0}|\hat{\mathbf{s}}_ {t}, \mathbf{a}_ t))\]\to 0$, corroborating the effective distribution estimation capability offered by the proposed asynchronous diffusion model.*"
>
> * Theorem 4: "*As $ n \to \infty $, the estimated causal state $\widetilde{V}^\pi( \zeta( \mathbf{s} ) )$ in (15) converges to within $2\hat{\epsilon}$-neighborhood of the ground-truth causal state $V^\pi(\mathbf{s})$, i.e., the neighborhood region of the ground-truth causal state $V^\pi(\mathbf{s})$ with the radius of $\hat{\epsilon}$.*"
>
> **Reference**
>
> [1]. Fu, Hengyu, et al. "Unveil conditional diffusion models with classifier-free guidance: A sharp statistical theory." arXiv preprint arXiv:2403.11968 (2024).
>
> [2]. Kemertas, Mete, and Tristan Aumentado-Armstrong. "Towards robust bisimulation metric learning." Advances in Neural Information Processing Systems 34 (2021): 4764-4777.

---

### Official Review · Reviewer_xJtb · 2024-11-02

**Soundness:** 2
**Presentation:** 1
**Contribution:** 2
**Rating:** 3
**Confidence:** 2

**Summary:**

This paper proposes a method that incorporates the diffusion model in RL algorithms, along with the theoretical guarantees. Empirical evaluation is also included.

**Strengths:**

N/A

Caveat: As a researcher in learning theory, I am not familiar with the current line of research on RL with diffusion models. In below, I only evaluate this paper based on its math and theory, and my review should be taken with AC's discretion.

**Weaknesses:**

I find this paper difficult to understand.

1. The term "bisimulation" is never defined or well-explained. Definition 1 is also not rigorous: in POMDP, the distribution P(s_{t+1}|o_t,a) clearly depends on the history (and hence the policy). It is also not clear why there is a partition of the state space into the observation space.

2. Definition 2/Assumption 3 seems to define the bisimulation metric in terms of a fixed-point equation. While such a metric is shown to exist (at least when p=1), I find it difficult to interpret.

3. In Theorem 2, $\mathcal{E}_\zeta$ is defined twice.

4. Theorem 4 is claimed to establish the convergence of the proposed algorithm, but I can't see how. In. particular, the RHS of (15) is not vanishing when n tends to infinity. Further, the algorithm is based on gradient descent, but there is no analysis of GD here (it is vaguely mentioned that previous results can be invoked).

**Questions:**

See my discussion on the weakness.

---

> ### Author Response · Authors · 2024-11-22
> **Part I**
>
> **Weakness 1:** As an important concept in theoretical computational science, bisimulation is a method for identifying equivalent spaces under a given transition. For basic information about the definition of bisimulation, please refer to Ref. [1-3].
>
> Under MDP, bisimulation provides a framework to measure the equivalence of states based on their behavioral similarity. Specifically, bisimulation requires that if two states $\mathbf{s}_i$ and $\mathbf{s}_j$ are bisimilar, executing the same action $\mathbf{a}$ from these states should lead to statistically indistinguishable next-state distributions and reward distributions. This ensures that the dynamics of the system do not differentiate between $\mathbf{s}_i$ and $\mathbf{s}_j$, making them interchangeable in terms of their future trajectories.
>
> In reinforcement learning, the joint distribution $p(\mathbf{s}_ {t+1}, r_ {t+1} \mid \mathbf{s}_ {t}, \mathbf{a}_ {t})$ describes the transition dynamics and reward generation when action $\mathbf{a}_ {t}$ is taken in state $\mathbf{s}_ {t}$. The joint distribution $p(\mathbf{s}_ {t+1}, r_ {t+1} \mid \mathbf{s}_ {t}, \mathbf{a}_ {t})$ still governs the system's behavior. This joint distribution can be decomposed into the state transition probability $p(\mathbf{s}_ {t+1} \mid \mathbf{s}_ {t}, \mathbf{a}_ {t})$ and reward distribution $p(r_ {t+1} \mid \mathbf{s}_ {t}, \mathbf{a}_ {t})$ due to the previously rigorously proved independence of the future state $\mathbf{s}_ {t+1}$ and the reward $r_ {t+1}$ conditioned on the current state $\mathbf{s}_ {t}$ and action $\mathbf{a}_ {t}$ (See Ref. [4] for details).
>
> However, direct access to the underlying states is unavailable in POMDPs; instead, only the observations $\mathbf{o}_ {t}$ are accessible. Since $\mathbf{o}_ {t}$ reacts to $\mathbf{s}_ {t}$, we can establish the equivalence of states and observations even under partial observability. The combination of the state transition probability and reward distribution serves as a foundation for defining bisimulation in POMDPs, establishing a metric for state similarity.
>
> In the revised version, we have revised Definition 1 and added the following explanation: "*Based on the environment's dynamics $P(\mathbf{s}_ {t+1}, r_ {t+1}|\mathbf{s}_ t, \mathbf{a}_ t)$, the similarity between environments can be expressed by the similarity between their state transition and reward functions.*"
>
> **Weakness 2:** Sincere thanks for your insightful comment. We have rewritten Definition 2 and Assumption 3 in the revised version to help readers interpret the bisimulation metric.
>
> For Definition 2, we define the bisimulation metric between the causal state extracted from observations and the true state underlying the POMDPs. To clarify this, we have added the phrase "*for any pair of state and observation $\{\mathbf{s}_ t \in \mathcal{S}, \mathbf{o}_ t \in \mathcal{O}\}$*" in the revised version.
>
> For Assumption 3, we establish the existence and uniqueness of the bisimulation metric between any two states. Accordingly, we have included "$\forall (\mathbf{s}_i, \mathbf{s}_j) \in \mathcal{S} \times \mathcal{S}$" in the revised version.
>
> **Weakness 3:** We apologize for this typo, where $ \zeta $ was used to denote two different models. We have clarified this in the revised version by using $ \phi $ to indicate the reward noise model and $ \zeta $ to denote the bisimulation model.

---

> ### Author Response · Authors · 2024-11-22
> **Part II**
>
> **Weakness 4:** Sincere thanks to the reviewer for providing this insightful feedback.
>
> * Regarding the convergence:
>
> As $ n \to \infty $, the estimated causal state $\widetilde{V}^\pi( \zeta( \mathbf{s} ) )$ in Eq. (15) converges to within $2\hat{\epsilon}$-neighborhood of the ground-truth causal state $V^\pi(\mathbf{s})$, i.e., the neighborhood region of the ground-truth causal state $V^\pi(\mathbf{s})$ with the radius of $\hat{\epsilon}$, where $\hat{\epsilon}$ depends on the nature of the environment.
>
> Specifically, the term $\mathcal{O}(\mathrm{poly}(n, d_s))$ on the RHS of Eq. (15) was introduced by the use of the RNN model for bisimulation metric learning. Its convergence has already been proved in Ref. [5].
>
> To evaluate the convergence of the term $\mathcal{O}\left(n^{-\frac{b}{2d_s+d_a+2b}} (\log n)^{\max(\frac{19}{2},\frac{b+2}{2})} \right)$ on the RHS of Eq. (15) is, in essence, to analyze the convergence of $\lim_{n \to \infty} \frac{\ln^{c_1} n}{n^{c_2}}$ with $c_1 = \max(\frac{19}{2},\frac{b+2}{2})>0$ and $c_2=\frac{b}{2d_s+d_a+2b}>0$. By applying L'Hôpital's rule, it follows that $\lim_{n \to \infty} \frac{\ln^{c_1} n}{n^{c_2}} = \lim_{n \to \infty} \frac{c_1\ln n}{n\times c_2 n^{c_2-1}} = \lim_{n \to \infty} \frac{c_1\ln n}{c_2 n^{c_2}} = \lim_{n \to \infty} \frac{c_1}{n\times c_2^2n^{c_2-1}} = \lim_{n \to \infty} \frac{c_1}{c_2^2n^{c_2}} = 0$. As a result, the term $\mathcal{O}\left(n^{-\frac{b}{2d_s+d_a+2b}} (\log n)^{\max(\frac{19}{2},\frac{b+2}{2})} \right)$ on the RHS of Eq. (15) converge to zero.
>
> As a consequence, the RHS of Eq. (15) converges to $2\hat{\epsilon}$. When $\hat{\epsilon}$ is sufficiently small, the RHS of Eq. (15) vanishes, i.e., converging to zero, as $ n\rightarrow\infty $.
>
> In response to this comment, we have added this explanation of Theorem 4 to the revised version as
>
> "*As $ n \to \infty $, the estimated causal state $\widetilde{V}^\pi( \zeta( \mathbf{s} ) )$ in (15) converges to within $2\hat{\epsilon}$-neighborhood of the ground-truth causal state $V^\pi(\mathbf{s})$, i.e., the neighborhood region of the ground-truth causal state $V^\pi(\mathbf{s})$ with the radius of $\hat{\epsilon}$.*"
>
> * Regarding the analysis of gradient descent (GD):
>
> First of all, we analyze the upper bound of the value gap under arbitrary model approximation errors (as discussed in Theorem 2), which does not involve gradient descent. Subsequently, we measure the exact approximation errors of $\mathcal{E}_ \zeta$, $\mathcal{E}_ \phi$, and $\mathcal{E}_ \theta$.
>
> In the case of $\mathcal{E}_ \zeta$, we utilize an RNN model to learn the bisimulation metric and apply the existing results in Ref. [5] to establish the convergence analysis of our proposed algorithm (as described in Theorem 4). The convergence analysis of RNNs by Allen-Zhu et al. in Ref. [5] has already captured gradient descent.
>
> In the case of $\mathcal{E}_ \phi$ and $\mathcal{E}_ \theta$, we examine the approximation error of the asynchronous diffusion model (as shown in Theorem 3), which indeed needs to consider gradient descent. Nevertheless, our work builds on the conclusions of Ref. [6]. In particular, Fu et al. in Ref. [6] analytically established the convergence of the classical classifier-free diffusion model under neural network approximation and proved the error of distribution approximation can converge to zero. This is given as Lemma 12 in our paper and used for our analysis of the approximation errors in the asynchronous diffusion model.
>
> As a result, Theorem 4 rigorously asserts the convergence of our proposed algorithm.
>
> **Reference**
>
> [1]. Sangiorgi, Davide. Introduction to bisimulation and coinduction. Cambridge University Press, 2011.
>
> [2]. Van der Schaft, A. J. "Equivalence of dynamical systems by bisimulation." IEEE transactions on automatic control 49.12 (2004): 2160-2172.
>
> [3]. Hansen-Estruch, Philippe, et al. "Bisimulation makes analogies in goal-conditioned reinforcement learning." International Conference on Machine Learning. PMLR, 2022.
>
> [4]. Sutton, Richard S., and Andrew G. Barto. "Reinforcement learning: an introduction, 2nd edn. Adaptive computation and machine learning." (2018).
>
> [5]. Allen-Zhu, Zeyuan, Yuanzhi Li, and Zhao Song. "On the convergence rate of training recurrent neural networks." Advances in neural information processing systems 32 (2019).
>
> [6]. Fu, Hengyu, et al. "Unveil conditional diffusion models with classifier-free guidance: A sharp statistical theory." arXiv preprint arXiv:2403.11968 (2024).

---

> > ### Comment · Reviewer_xJtb · 2024-11-25
> >
> > Thank you for the response. There are still some confusing points in the revised paper:
> >
> > (1) While it is ok to cite the existing results of GD convergence, you should at least include a clear statement of the cited results and how it is specialized to this setting.
> >
> > (2) The upper bound in Theorem 4 still has a term O(poly(n,d)), which goes to infinity as n tends to infinity.
> >
> > Generally speaking, the readability of the revised paper does not improve much, so I will keep my score.

---

> > > ### Author Response · Authors · 2024-11-26
> > > **Response to Question (1)**
> > >
> > > (1) In terms of GD convergence, we apologize for confusion in our earlier response. We wish to clarify that our focus is on convergence from a statistical perspective. Statistical convergence refers to how the learned model's output achieves desired properties as the number of samples increases. By contrast, optimization convergence focuses on minimizing a loss function through iterative methods like GD (see Ref. [1] for details). Given the non-convex nature of our problem and our objective of bounding the value function rather than minimizing a loss function, it is appropriate to consider statistical convergence, which does not involve GD convergence analysis.
> > >
> > > As stated in Theorem 2, our analysis specifically addresses the statistical convergence of two components: (i) the asynchronous diffusion model used for reward approximation and state transition, and (ii) the RNN model for bisimulation metric learning.
> > >
> > > * For the diffusion model: The statistical convergence analysis  typically revolves around bounding the distribution estimation error, which can be decomposed into three key components: initialization error, score estimation error, and discretization error (see Refs. [2–4] for detailed discussions). By appropriately setting the early-stopping step $k_0$ and the total number of diffusion steps $K$, we show that the overall estimation error can be simplified to $\mathcal{O}\left(n^{-\frac{b}{2d_s+d_a+2b}} (\log n)^{\max(19/2,(b+2)/2)} \right)$, as stated in Theorem 3. Please refer to Appendix B2.3 (i.e., Eq. (37)) for the detailed decomposition of the distribution estimation error and individual bounds derived for each component.
> > >
> > > * For the RNN model: In our earlier version, we mistakenly cited the optimization convergence results of RNNs from Ref. [5], which were not applicable to our statistical convergence framework. Upon re-examining the literature, we have identified the appropriate statistical convergence properties of RNNs from Ref. [6], which proves that the statistical convergence rate of an RNN model is bounded as $\mathcal{O}(n^{-\frac{p_R}{2p_R + d_s + 1}} (\log n)^6)$, where $p_R$ corresponds to the RNN's size and $d_s$ denotes the state dimension.
> > >
> > > Consequently, GD convergence is not included in our statistical convergence analysis. Instead, our analysis of the diffusion model and RNN model aligns with established practices in the field.

---

> > > ### Author Response · Authors · 2024-11-26
> > > **Response to Question (2)**
> > >
> > > (2) In terms of Theorem 4, inspired by your insights, we have now adopted the statistical convergence analysis of RNNs from Ref. [6] and updated Eq. (15) in Theorem 4 as
> > > $$
> > >     \mathbb{E}_ {\\{\mathbf{o}_ t, \mathbf{a}_ t, r_ t, \mathbf{o}_ {t+1}\\}} | V^\pi(\mathbf{s}) - \widetilde{V}^\pi( \zeta( \mathbf{s} ) ) | \leq 2\widehat{\epsilon} + \frac{1}{c_ {\mathrm{R}}(1 - \gamma)} (\mathcal{O}(n^{-\frac{2p_ R}{2p_ R +d_ s+1}}(\log n)^6)+\frac{2c_ {\mathrm{R}}+2c_ {\mathrm{T}}}{1 - c_ {\mathrm{T}}-c_ {\mathrm{R}}} \mathcal{T}(\mathbf{s}^{\star}, \mathbf{a}^{\star})\mathcal{O}(n^{-\frac{b}{2d_ s+d_ a+2b}} (\log n)^{\max(19/2,(b+2)/2)} )).
> > > $$
> > > where term $\mathcal{O}(n^{-\frac{2p_R}{2p_R +d_s+1}}(\log n)^6)$ refers to the bisimulation metric learning error based on the RNN (See Ref. [1] for details), and term $\frac{2c_{\mathrm{R}}+2c_{\mathrm{T}}}{1 - c_{\mathrm{T}}-c_{\mathrm{R}}} \mathcal{T}(\mathbf{s}^{\star}, \mathbf{a}^{\star})\mathcal{O}\left(n^{-\frac{b}{2d_s+d_a+2b}} (\log n)^{\max\{19/2,(b+2)/2\}} \right)$ represents the errors in reward approximation and state transition modeling by the asynchronous diffusion models.
> > >
> > > To analyze the convergence of $ \mathbb{E}_ {\\{\mathbf{o}_ t, \mathbf{a}_ t, r_ t, \mathbf{o}_ {t+1}\\}}| V^\pi(\mathbf{s}) - \widetilde{V}^\pi( \zeta( \mathbf{s} ) ) |$ is in essence to analyze the convergence of $\frac{\ln^{c_1} n}{n^{c_2}}$. This is because $c_1=6$ and $c_2=\frac{2p_R}{2p_R +d_s+1}>0$ in $\mathcal{O}(n^{-\frac{2p_R}{2p_R +d_s+1}}(\log n)^6)$; and $c_1 = \max\{\frac{19}{2},\frac{b+2}{2}\}>0$ and $c_2=\frac{b}{2d_s+d_a+2b}>0$ in $\frac{2c_{\mathrm{R}}+2c_{\mathrm{T}}}{1 - c_{\mathrm{T}}-c_{\mathrm{R}}} \mathcal{T}(\mathbf{s}^{\star}, \mathbf{a}^{\star})\mathcal{O}\left(n^{-\frac{b}{2d_s+d_a+2b}} (\log n)^{\max(19/2,(b+2)/2)} \right)$.
> > >
> > > By applying L'Hôpital's rule, it follows that $\lim_{n \to \infty} \frac{\ln^{c_1} n}{n^{c_2}} = \lim_{n \to \infty} \frac{c_1\ln n}{n\times c_2 n^{c_2-1}} = \lim_{n \to \infty} \frac{c_1\ln n}{c_2 n^{c_2}} = \lim_{n \to \infty} \frac{c_1}{n\times c_2^2n^{c_2-1}} = \lim_{n \to \infty} \frac{c_1}{c_2^2n^{c_2}} = 0$, $\forall c_1,c_2>0$. As a result, both terms $\mathcal{O}(n^{-\frac{2p_R}{2p_R +d_s+1}}(\log n)^6)$ and $\frac{2c_{\mathrm{R}}+2c_{\mathrm{T}}}{1 - c_{\mathrm{T}}-c_{\mathrm{R}}} \mathcal{T}(\mathbf{s}^{\star}, \mathbf{a}^{\star})\mathcal{O}\left(n^{-\frac{b}{2d_s+d_a+2b}} (\log n)^{\max\{19/2,(b+2)/2\}} \right)$ on the RHS of Eq. (15) converge to zero, as $n\rightarrow \infty$. The estimated causal state $\widetilde{V}^\pi( \zeta( \mathbf{s} ) )$ in Eq. (15) converges to within $2\hat{\epsilon}$-neighborhood of the ground-truth causal state $V^\pi(\mathbf{s})$, i.e., the neighborhood region of the ground-truth causal state $V^\pi(\mathbf{s})$ with the radius of $\hat{\epsilon}$. In other words, the asymptotic convergence of the proposed algorithm is established, as $n \to \infty$.
> > >
> > > To clarify this, we have incorporated the above analysis into Appendix B2.4 of the revised version and added the following statement after Theorem 4:
> > > *"Therefore, we have established the asymptotic convergence of the proposed algorithm. See Appendix B2.4 for details."*
> > >
> > > **Reference**
> > >
> > > [1]. Bartlett, Peter L., Andrea Montanari, and Alexander Rakhlin. "Deep learning: a statistical viewpoint." Acta numerica 30 (2021): 87-201.
> > >
> > > [2]. Fu, Hengyu, et al. "Unveil conditional diffusion models with classifier-free guidance: A sharp statistical theory." arXiv preprint arXiv:2403.11968 (2024).
> > >
> > > [3]. Oko, Kazusato, Shunta Akiyama, and Taiji Suzuki. "Diffusion models are minimax optimal distribution estimators." International Conference on Machine Learning. PMLR, 2023.
> > >
> > > [4]. Chen, Minshuo, et al. "Score approximation, estimation and distribution recovery of diffusion models on low-dimensional data." International Conference on Machine Learning. PMLR, 2023.
> > >
> > > [5]. Allen-Zhu, Zeyuan, Yuanzhi Li, and Zhao Song. "On the convergence rate of training recurrent neural networks." Advances in neural information processing systems 32 (2019).
> > >
> > > [6]. Kohler, Michael, and Adam Krzyżak. "On the rate of convergence of a deep recurrent neural network estimate in a regression problem with dependent data." Bernoulli 29.2 (2023): 1663-1685.

---

> > > ### Author Response · Authors · 2024-11-28
> > >
> > > Thank you again for your time and valuable feedback during the discussion period. We have carefully addressed your concerns regarding the convergence in Theorem 4, and the clarifications have been incorporated into the revised version.
> > >
> > > As the discussion period has been extended to December 2nd (AOE), we hope this extension provides ample time for any final comments or follow-ups. Your continued input is invaluable in ensuring the rigor and clarity of the paper.
> > >
> > > Thank you once again for your attention and support.

---

> > > > ### Author Response · Authors · 2024-12-03
> > > >
> > > > Sincere thanks for your acknowledgment and kind approval of our response and revision. If there are no further questions or concerns, we kindly hope you might consider raising the score of our submission.

---

### Official Review · Reviewer_bMYC · 2024-11-03

**Soundness:** 2
**Presentation:** 1
**Contribution:** 2
**Rating:** 5
**Confidence:** 2

**Summary:**

In this paper, the authors propose a method of dealing with noisy observations in RL, which they call CSR-ADM. Intuitively, the algorithm uses both a denoise model and a bisimulation metric to find 'causal state representations' for a given observation, which can be used by off-the-shelf RL algorithms to compute policies. The authors provide a sub-optimality bound for their method under some (reasonable) assumptions on the dynamics of the environment. Empirically, the authors show that incorporating their method into SAC improves performance and outperforms methods that only consider denoising or finding causal representations.

**Strengths:**

* The topic of the paper is interesting and significant: dealing with partial observability is a key problem when applying RL in the real world.
* The paper combines both theory and application nicely. Moreover, the proposed method can easily be incorporated into off-the-shelf RL methods, which makes it easier to apply in practice.
* The authors compare their method with relevant baseline algorithms and use an ablation study to show the relevance of all proposed components.

**Weaknesses:**

The main weakness of this paper is its presentation. The intuition behind the methods is easy to follow, but details are often unclear: see some of my questions below. Because of this, I find it hard to determine the quality of the proposed method.

I'll note some other minor weaknesses:
* The method assumes Gaussian noise. Thus, the method may struggle with other noise (such as raindrops or colour shifts), and does not help with other types of partial observability (such as missing data).
* The paper does not quantify the additional computational cost of the method: this would be good to add.

**Questions:**

* In Eqs. 1a-1c, what exactly are the assumptions you make about the dynamics of the model? For example, must the state-, action- and observation spaces be continuous, or can they be discrete? What about the functions $f,g$ and $h$? Can two states give the same observation?
* In eq. 1, $f$ denotes the observation function. However, in Def. 1 and in Assumption 2, $f$ is also used to denote something that looks like an observation function but has a different number of inputs. How do these relate?
* $\zeta$ is overloaded in a confusing way: it is used to describe the predicted state (line 234), the noisy state (line 269), as well as the bisimulation model (Alg 1, line 2). Do these all represent the same thing?
* In Eqs. 7 and 8 and Alg 1, $\theta$ and $\zeta$ seem to be switched. Is this a typo?
* After eq. 6, the paper mentions a variables $n$, $\hat{s}_{t+1}$ and $r_{t+1}$. What do these refer to?
* In Alg. 1, what is the function of line 7? What do we use the sampled transitions for?
* In line 297, how can the diffusion model predict a future state? I thought it only removed noise?
* In Assumption 3, what does this assumption intuitively mean? Are $s_i, s_j \in S$, or in $S \cup O$ ?
* I do not understand Thm 1: it seems to me that if we pick $c_T \approx 0$ and $c_R \approx 1$, then any states that have the same immediate reward would have a value gap of $\approx 0$ as well, which clearly is not the case. Can you explain why it holds?
* In Fig 4 (App. A), how is $P(\hat{s}_t|o_t)$ computed? I thought this was what the diffusion model was used for.

---

> ### Author Response · Authors · 2024-11-22
> **Part I**
>
> **Weakness 1:** Sincere thanks for this insightful and inspirational comment. As pointed out in this comment, our current paper focuses on Gaussian noise for the reason that Gaussian noise is the most widely observed and considered in the literature. The consideration of Gaussian noise facilitates our analysis, helps shed insights, and can serve as a foundation for various noise distributions.
>
> While specific results or resultant mathematic expressions relying on the specific noise distribution could change under different noise distributions, the derivation steps and methodology developed in this paper would remain invariant under different noise distributions. Moreover, the Gaussian Mixture Model (GMM), created by superimposing Gaussian distributions, has demonstrated an excellent ability to approximate any target distribution (See Ref. [1] for details). In this sense, our consideration of Gaussian noise has the potential to be extended to accommodate any distribution.
>
> In terms of other types of partial observability (such as missing data), we would like to note that missing data imputation could be potentially solved by the diffusion model (See Ref. [2] for details). However, the detailed analysis of this lies beyond the scope of our study.
>
> **Weakness 2:** Thank you for your feedback regarding the computation cost. We would clarify that our proposed CSR-ADM framework involves primarily three computational components, i.e., denoising and fitting conditional probability distributions using the asynchronous diffusion model, learning the bisimulation metric through models such as RNNs or MLPs, and reinforcement learning decision-making. The additional computational cost of CSR-ADM compared to traditional reinforcement learning stems from the asynchronous diffusion model in the first computational component.
>
> Due to the introduction of noise intensity, the loss function Eq. (5) of the asynchronous diffusion model is twice that of a standard diffusion model, thereby doubling the associated computational cost. In Ref. [3], the computational cost of the diffusion model is $\widetilde{\mathcal{O}}(\mathrm{poly} \log d)$, where $d$ is the dimension of the input data. As a result, the computational cost of the causal state representation in our proposed CSR-ADM algorithm is $\widetilde{\mathcal{O}}(\mathrm{poly} \log \max\{|\mathcal{A}|, |\mathcal{O}|\})$, where $|\mathcal{A}|$ and $|\mathcal{O}|$ are the dimensions of the action and observation spaces, respectively.
>
> In the revised version, we have included the following analysis of computational cost in Section 4:
>
> "*We evaluate the additional computational cost of the CSR-ADM compared to typical RL algorithms. Chen et al. (2024) analyzed the computational cost of a diffusion model to be $\widetilde{\mathcal{O}}(\mathrm{poly} \log d)$, where $d$ is the dimension of the input data. Considering our definition of noise intensity, the loss function of the asynchronous diffusion model (see Eq. (5)) is twice that of a standard diffusion model, directly doubling the computational cost. Therefore, the computational cost of the causal state representation is $\widetilde{\mathcal{O}}(\mathrm{poly} \log \max\{|\mathcal{A}|, |\mathcal{O}|\})$ in CSR-ADM.*"
>
> **Q1:** Thank you for providing your insightful feedback. As clarified in the revised version, we make no assumptions about the dynamics of the environment; in other words, the state space, action space, and observation space can be either continuous or discrete. Similarly, the functions $ f $, $ g $, and $ h $ can be either continuous and discrete. As also clarified in the revised version, two distinct states may produce the same observation due to the presence of noise.
>
> **Q2:** We apologize for the confusion caused by this denoise definition. In the revised version, we define $ f $ as a $ b $-Hölder norm function and $ F $ as the observation function.
>
> **Q3:** We are extremely sorry for this typo. As classified in the revised version, $ \zeta $ denotes the bisimulation model (see Algorithm 1, line 2), which be interpreted as the function that denoises observations and extracts causal states (original manuscript, line 234; revised version, line 213). Moreover, the term "noisy model" has been corrected to "$p(\mathbf{x}^k | \mathbf{x})$'' to avoid confusion in the revised version (original manuscript, line 269; revised version, line 256).
>
> **Q4:** We apologize for this typo. We have corrected the typo in the revised version of Eqs. (7) and (8) and Algo. 1.

---

> ### Author Response · Authors · 2024-11-22
> **Part II**
>
> **Q5:** We have now clarified in the revised version that $ n $ denotes the number of samples used in training, which is now defined in line 266 of the revised version (original manuscript, line 278). $ \hat{\mathbf{s}}_ {t+1} $ represents the causal state obtained after denoising and extracting the causal information from the observation at time $ t+1 $, which is defined in line 205 of the revised version (original manuscript, line 212). Moreover, $ r_{t+1} $ denotes the reward received at time $ t+1 $, which is now defined in line 140 of the revised version (original manuscript, line 134).
>
> **Q6:** We would like to clarify that the purpose of line 7 in Algorithm 1 is to sample a batch of transitions from the replay buffer for the training of reinforcement learning. As a key feature of standard reinforcement learning, the replay buffer reduces correlations among samples and improves training efficiency, as described in Ref. [4]. The sampled transitions are used to evaluate the losses in Eqs. (7), (8), and (10), followed by gradient descent to optimize the model.
>
> **Q7:** Sincere thanks for this astute comment. As pointed out in this comment, the diffusion models remove noises, i.e., denoising. This is achieved by fitting the conditional probability distributions for the transition dynamics and rewards distribution (see Eqs. (1b) and (1c)) and then applying the distributions to estimate current causal states from the perturbed sample $(\mathbf{o}_ t, \mathbf{a}_ t, r_ {t+1}, \mathbf{o}_ {t+1})$. The diffusion models can effectively denoise states when the proposed asynchronous diffusion model fits the conditional probability distribution reasonably well in Eq. (1c).
>
> **Q8:** Intuitively, Assumption 3 refers to a common premise in the study of bisimulation metrics (e.g., see Ref. [5] and references therein), that is, the existence and uniqueness of the bisimulation metric for any pair of states. In other words, for any pair of states $(\mathbf{s}_i, \mathbf{s}_j)$, there exists a unique bisimulation metric to measure their similarity. Under this assumption, it becomes possible to use a model (e.g., the RNN model described in Theorem 4) to learn the bisimulation metric. In response to this suggestion, we have added the intuitive explanation of Assumption 3 in the revised version as
>
> "*To generalize the VFA bound, we assume the existence and uniqueness of $p$-Wasserstein bisimulation metric for any pair of states to measure their similarity.*"
>
> As also suggested, we have clarified "$ \forall(\mathbf{s}_i, \mathbf{s}_j) \in \mathcal{S} \times \mathcal{S} $" in the revised version of Assumption 3.
>
> **Q9:** We express our heartfelt gratitude for this astute and inspirational comment. As pointed out in this comment, if we choose $ c_T \approx 0 $ and $ c_R \approx 0 $, then the right-hand side (RHS) of Eq. (13) is indeed equal to zero. Since the left-hand side (LHS) of Eq. (13) is the product of $ c_R $ and the value gap with $ c_R \approx 0 $, the LHS of Eq. (13) is also zero, even though the value gap in the LHS of Eq. (13) is not necessarily zero. As a result, Theorem 1 still holds when $ c_T \approx 0 $ and $ c_R \approx 0 $.
>
> On the other hand, we are interested in the boundedness and the upper bound of the value gap. In Theorem 2, we further derive the upper bound of the value gap by explicitly imposing the condition that $ c_T $ and $ c_R $ are non-zero. This is because the bisimulation metric would be invariably zero and become useless, if $c_T$ and $c_R$ are zero. Nevertheless, even if $ c_T \approx 0 $ and $ c_R \approx 0 $, the value gap derived in Theorem 2 remains finite since the reward defined is bounded by $[0, 1]$ (revised version, line 140). In other words, the value gap is always bounded under the bisimulation metric.
>
> **Q10:** Sincere thanks for your insightful question. As clarified in the revised version, $ P(\hat{\mathbf{s}}_t|\mathbf{o}_t) $ represents the process of extracting denoised causal states from noisy observations, which can be divided into denoising and causal state extraction, and computed by the proposed asynchronous diffusion model and an RNN model, respectively.
>
> The denoising is achieved using a diffusion model, where we design a novel asynchronous diffusion model to effectively denoise perturbed observations. By contrast, the causal state extraction does not impose restrictions on the model used for fitting and extracting causal states. In Theorem 4, we employ an RNN model to learn the bisimulation metric for extracting causal states. Together, these two components jointly realize $ P(\hat{s}_t|o_t) $.
>
> To classify this, we have added the following explanation in Appendix A: "*It should be noted that the asynchronous diffusion model algorithm denoises observations, which are then input into the bisimulation metric learning model to extract causal states.*"

---

> ### Author Response · Authors · 2024-11-22
> **Reference**
>
> **Reference**
>
> [1]. Li, Jonathan, and Andrew Barron. "Mixture density estimation." Advances in neural information processing systems 12 (1999).
>
> [2]. Chen, Zhichao, et al. "Rethinking the Diffusion Models for Missing Data Imputation: A Gradient Flow Perspective." The Thirty-eighth Annual Conference on Neural Information Processing Systems. 2024.
>
> [3]. Chen, Haoxuan, et al. "Accelerating Diffusion Models with Parallel Sampling: Inference at Sub-Linear Time Complexity." The Thirty-eighth Annual Conference on Neural Information Processing Systems. 2024.
>
> [4]. Eysenbach, Ben, Russ R. Salakhutdinov, and Sergey Levine. "Search on the replay buffer: Bridging planning and reinforcement learning." Advances in neural information processing systems 32 (2019).
>
> [5]. Kemertas, Mete, and Tristan Aumentado-Armstrong. "Towards robust bisimulation metric learning." Advances in Neural Information Processing Systems 34 (2021): 4764-4777.

---

> > ### Comment · Reviewer_bMYC · 2024-11-24
> > **Follow up rebuttal**
> >
> > I'd like to thank the authors for their in-depth answers. I have a couple of follow-up comments/questions:
> >
> > **Q7:** Looking at the text again, I see that I was confused about what exactly the diffusion model does. This is explained differently in different places: line 206 describes that it both denoises a state and predicts a future state, while line 211 implies that it only predict future states, and line 221 and 234 imply it only denoises states (i.e. it predicts $s_t$ with $o_t$ without considering $s_{t-1}$). I assume the first explanation is the correct one, but then the explanation/notation in the other places is inconsistent.
> >
> > **Q9:** It seems like you misread my question: I suggested that if $c_R\approx1$ (not 0) the theorem seems incorrect: the lefthandside of eq. 13 would not be canceled out, but $d(s_i,s_j)$ would only depend on the direct rewards for $s_i$ and $s_j$ (eq. 12) (and in particular, becomes $\approx 0$ if the expected direct rewards are equal). This seems intuitively incorrect to me: can you explain why this holds?

---

> ### Author Response · Authors · 2024-11-25
> **Response**
>
> **Q7***: Thank you so much for your very careful review and intriguing comment. After carefully considering your comments, we confirm that our asynchronous diffusion model only removes noise. Specifically, given the perturbed and partially obstructed observation $\mathbf{o}_ {t+1}$, the problem of interest is to uncover the underlying causal state $\mathbf{s}_ {t+1}$ by leveraging the learning experience from the past state $\mathbf{s}_ {t}$ and action $\mathbf{a}_ {t}$. In essence, this is a denoising process.
>
> We have realized that our previous response to your previous question Q7 was inaccurate and misleading. We have now updated that response. Moreover, we have made the following modifications in the revised manuscript to ensure consistency:
>
> Line 206: "*Specifically, we design an asynchronous diffusion model to simultaneously denoise the states and rewards through the environment dynamics estimation*";
>
> Line 211: "*The objective of the asynchronous diffusion model is to derive $P(\hat{\mathbf{s}}_ {t+1}\mid \hat{\mathbf{s}}_ {t}, \mathbf{a}_ t)$ and $P(\widehat{r}_ {t+1}\mid \hat{\mathbf{s}}_ {t}, \mathbf{a}_ t)$ from perturbed sample $(\mathbf{o}_ t, \mathbf{a}_ t, r_ {t+1}, \mathbf{o}_ {t+1})$, where $\hat{\mathbf{s}}_ {t}$ and $\hat{\mathbf{s}}_ {t+1}$ denote the causal states estimated under denoised observations, and $\widehat{r}_ {t+1}$ represents the denoised reward at time $t+1$*";
>
> Line 221: "*Compute the (approximate) denoised causal state $\hat{\mathbf{s}}_t$ from $\mathbf{o}_t$ using observation denoise model $\theta$ and bisimulation model $\zeta$*";
>
> Line 234: "*To obtain the denoised causal state $\hat{\mathbf{s}}_ {t+1}$, we use $r_ {t+1}$ and $\tilde{\mathbf{s}}_ {t+1} = \zeta(\mathbf{o}_ {t+1})$ as part of the inputs to the asynchronous diffusion model, along with $\hat{\mathbf{s}}_ {t}$ and $\mathbf{o}_ {t}$, where $\tilde{\mathbf{s}}_ {t+1}$ represents the causal state with noise*".
>
> We hope these revisions address the inconsistencies and provide a clearer explanation. Thank you for pointing this out.
>
> **Q9***: Sincere thanks for this very insightful and intriguing comment. As clarified in the revised version, we set $c_ {\mathrm{T}}$ and $c_ {\mathrm{R}}$ to be non-zero in Assumption 3, due to the fact that both the state transition probability and reward distribution are indispensable for establishing the equivalence between any two causal states and, subsequently, for establishing a metric for state bisimilarity.
>
> On the one hand, the joint distribution $p(\mathbf{s}_ {t+1}, r_ {t+1} \mid \mathbf{s}_ {t}, \mathbf{a}_ {t})$ describes the transition dynamics and reward generation when action $\mathbf{a}_ {t}$ is taken in state $\mathbf{s}_ {t}$. It governs the system's behavior. The joint distribution $p(\mathbf{s}_ {t+1}, r_ {t+1} \mid \mathbf{s}_ {t}, \mathbf{a}_ {t})$ can be decomposed into the state transition probability $p(\mathbf{s}_ {t+1} \mid \mathbf{s}_ {t}, \mathbf{a}_ {t})$ and reward distribution $p(r_ {t+1} \mid \mathbf{s}_ {t}, \mathbf{a}_ {t})$ due to the previously rigorously proved independence of the future state $\mathbf{s}_ {t+1}$ and the reward $r_ {t+1}$ conditioned on the current state $\mathbf{s}_ {t}$ and action $\mathbf{a}_ {t}$ (See Ref. [1] for details). It is crucial to ensure the equivalence of two causal states in both state transition probabilities and reward distributions to satisfy the requirements of bisimulation.
>
> On the other hand, we use the $p$-Wasserstein distance to quantify the differences between two bisimilar states by measuring their differences in both state transition probabilities and reward distributions. If $c_ {\mathrm{R}} \approx 1$, the constraint $c_ {\mathrm{R}} + c_ {\mathrm{T}} < 1$ (needed to guarantee bounded difference between value functions of causal states under the bisimulation metric, as formally established in Theorem 1) implies that $c_ {\mathrm{T}} \approx 0$. In this case, the bisimulation metric fails to capture the impact of state transition probabilities on the equivalence relationship of the two states. This would invalidate the definition of bisimulation.
>
> To clarify this, we have highlighted the permissible ranges of $c_ {\mathrm{T}}$ and $c_ {\mathrm{R}}$ (neither of them can take the values of 0 or 1) in Assumption 3 and Theorem 1 in the revised version.
>
> **Reference**
>
> [1]. Sutton, Richard S., and Andrew G. Barto. "Reinforcement learning: an introduction, 2nd edn. Adaptive computation and machine learning." (2018).

---

> > ### Comment · Reviewer_bMYC · 2024-11-25
> > **Response**
> >
> > Thanks to the authors for their quick response to my comments. However, I do not feel like my concern about Thm. 1 (Q9) is sufficiently addressed. Even when restricting $c_T > 0$, choosing a sufficiently small value would still yield the problem that I described before, where future states are practically neglected when finding a value bound. Thus, to convince me that this theorem holds I’d need some (intuitive) explanation as to why the problem I describe is not a problem.

---

> > > ### Author Response · Authors · 2024-11-26
> > >
> > > Sincere thanks for your continued engagement with our work and for highlighting this important concern. We greatly appreciate your insightful and inspirational feedback.
> > >
> > > In light of this feedback, we have revisited Theorem 1 and its proof. We notice that the condition $ c_{\mathrm{T}} \geq \gamma $ is required for the validity of Theorem 1, where $\gamma$ is the discount factor in reinforcement learning within $(0, 1)$. This condition arises from the mathematical induction used to prove the theorem. The proof of Theorem 1 is provided in Appendix 2.1.
> > >
> > > Using mathematical induction, we define the update of the value function and bisimulation metric as
> > > $$
> > > V^{(t+1)}(\mathbf{s}_ i) = \max_{\mathbf{a}\in\mathcal{A}}(\int_ {r\in \mathcal{R}}r(\mathbf{s}_ i,\mathbf{a})P(r\mid \mathbf{s}_ {i}, \mathbf{a})dr + \gamma \int_ {\mathbf{s}' \in \mathcal{S}}P(\mathbf{s}'\mid \mathbf{s}_ {i}, \mathbf{a})V^{(t)}(\mathbf{s}')d\mathbf{s}')
> > > $$
> > > $$
> > > d^{(t+1)}(\mathbf{s}_ i, \mathbf{s}_ j) = \max_ {\mathbf{a}\in\mathcal{A}}(c_ {\mathrm{R}}W_ p(d^{(t)})(P(r\mid \mathbf{s}_ {i}, \mathbf{a}), P(r\mid \mathbf{s}_ {j}, \mathbf{a}))+c_ {\mathrm{T}} W_ p(d^{(t)})(P(\mathbf{s}'\mid \mathbf{s}_ {i}, \mathbf{a}), P(\mathbf{s}'\mid \mathbf{s}_ {j}, \mathbf{a}))).
> > > $$
> > > By assuming that Eq. (13) holds in the case of $t$, we can derive the inequality in the case of $t+1$: $c_ {\mathrm{R}}|V^{(t+1)}(\mathbf{s}_ i) - V^{(t+1)}(\mathbf{s}_ j)| \leq A_1 + A_2$ (see Eq. (30) for details), with $$A_1 = c_ {\mathrm{R}} \max_ {\mathbf{a} \in \mathcal{A}} | \int_ {r \in \mathcal{R}} r(\mathbf{s}_ i, \mathbf{a}) P(r \mid \mathbf{s}_ i, \mathbf{a}) \, dr - \int_ {r \in \mathcal{R}} r(\mathbf{s}_ j, \mathbf{a}) P(r \mid \mathbf{s}_ j, \mathbf{a}) \, dr |$$ and $$A_2 = c_ {\mathrm{T}} \max_ {\mathbf{a} \in \mathcal{A}} | \int_ {\mathbf{s}' \in \mathcal{S}} ( P(\mathbf{s}' \mid \mathbf{s}_ i, \mathbf{a}) - P(\mathbf{s}' \mid \mathbf{s}_ j, \mathbf{a}) ) \frac{c_ {\mathrm{R}} \gamma}{c_ {\mathrm{T}}} V^{(t)}(\mathbf{s}') \, d\mathbf{s}' |.$$
> > > Based on the definition of the Wasserstein-1 distance (Eq. (18)), $d(\mathbf{s}_ i, \mathbf{s}_ j)$ can be expressed as the sum of two parts, $B_1+B_2$, under the 1-Lipschitz assumption, i.e., $$B_1 = c_ {\mathrm{R}} \max_ {\mathbf{a} \in \mathcal{A}} | \int_ {r \in \mathcal{R}} r(\mathbf{s}_ i, \mathbf{a}) P(r \mid \mathbf{s}_ i, \mathbf{a}) \, dr - \int_ {r \in \mathcal{R}} r(\mathbf{s}_ j, \mathbf{a}) P(r \mid \mathbf{s}_ j, \mathbf{a}) \, dr |$$ and $$B_2 = c_ {\mathrm{T}} \max_ {\mathbf{a} \in \mathcal{A}} | \int_ {\mathbf{s}' \in \mathcal{S}} ( P(\mathbf{s}' \mid \mathbf{s}_ i, \mathbf{a}) - P(\mathbf{s}' \mid \mathbf{s}_ j, \mathbf{a}) ) c_ {\mathrm{R}} V^{(t)}(\mathbf{s}') \, d\mathbf{s}' |.$$
> > > Clearly, we obtain $A_1 = B_1$. Therefore, for Eq. (13) in Theorem 1 to hold, $A_2\le B_2$ must be satisfied, resulting in $\frac{c_{\mathrm{R}} \gamma}{c_{\mathrm{T}}}\le c_{\mathrm{R}}$ and subsequently $c_{\mathrm{T}} \geq \gamma$.
> > >
> > > Intuitively, $c_{\mathrm{T}} \geq \gamma$ prevents the bisimulation metric from overly emphasizing on rewards at the expense of state dynamics. As $c_{\mathrm{T}} \rightarrow \gamma$, the right-hand side of Eq. (13) tends to
> > > $$d(\mathbf{s}_ i, \mathbf{s}_ j):=\max_ {\mathbf{a}\in\mathcal{A}}(c_ {\mathrm{R}}W_ p(d)(P(r\mid \mathbf{s}_ {i}, \mathbf{a}), P(r\mid\mathbf{s}_ {j}, \mathbf{a}))+c_ {\mathrm{T}} W_ p(d)(P(\mathbf{s}'\mid \mathbf{s}_ {i}, \mathbf{a}), P(\mathbf{s}'\mid \mathbf{s}_ {j}, \mathbf{a}))).$$
> > > As $\gamma\rightarrow 0$, the left-hand side of Eq. (13) tends to 0 since the value function is given by $V^\pi(\mathbf{s})=\mathbb{E}_ {\pi}[\sum_ {i=0}^\infty \gamma^t r_ {t+i+1}|s_ t=s]$. In this case, Eq. (13) turns out to be $$0\le \max_ {\mathbf{a}\in\mathcal{A}}(c_ {\mathrm{R}}W_ p(d)(P(r\mid \mathbf{s}_ {i}, \mathbf{a}), P(r\mid \mathbf{s}_ {j}, \mathbf{a})))= d(\mathbf{s}_ i, \mathbf{s}_ j).$$
> > > Therefore, Eq. (13) remains valid, as $c_ {\mathrm{T}} \rightarrow \gamma$ with $\gamma\rightarrow 0$.
> > >
> > > On the other hand, as $c_{\mathrm{R}} \rightarrow 0$, the left-hand side of Eq. (13) tends towards 0, leading to $0\le d(\mathbf{s}_ i, \mathbf{s}_ j)$, and Eq. (13) also holds.
> > >
> > > Thank you once again for your valuable insights and inspiration. Hopefully, this response addresses your in-depth comment.

---

> > > > ### Comment · Reviewer_bMYC · 2024-11-26
> > > > **Response**
> > > >
> > > > Thanks to the authors for finding and fixing the error in their proof. Although I think the revised paper is a significant improvement, I still think the paper could use more revisions to increase clarity. Moreover, although the proposed fix to Thm 1 seems correct to me, I do not have time to check the details (nor how the changes effect the subsequent proofs) since the revision has only been provided on the last day of the discussion period. Thus, I'll keep my rating.

---

> > > > > ### Author Response · Authors · 2024-11-26
> > > > >
> > > > > Thank you for your generous approval of our revision, as well as your recognition of the correctness of Theorem 1 and significant improvement.
> > > > >
> > > > > We would like to clarify that the proof of Theorem 1 remains intact since the initial version, as can be verified by comparing the two versions. The modifications made to Theorem 1 in response to your specific questions were solely intended to provide you and readers a clearer explanation of the validity of the theorem. These modifications do not change the original finding stated in Theorem 1 and its validity. These modifications and clarifications do not impact the correctness of the original proofs or the overall conclusions of the paper.
> > > > >
> > > > > We would also note that the discussion period has been extended by six days, until December 2 (AoE), to provide sufficient time for further clarification and review. Please take your time to check the details and kindly reconsider your rating.
> > > > >
> > > > > Thank you once again for your in-depth and inspirational feedback and consideration.

---

> > > > > ### Author Response · Authors · 2024-11-28
> > > > >
> > > > > Thank you for your time and valuable feedback throughout the discussion period. We have addressed the concerns raised and provided additional clarifications regarding Theorem 1 in my recent responses. These changes are minor, aiming to improve clarity and further explain the points of concern. If possible, We would greatly appreciate any further comments or feedback.
> > > > >
> > > > > Since the discussion period has been extended to December 2nd (AOE), I hope this allows time for any remaining questions or follow-ups. Your input would be invaluable in ensuring the clarity and rigor of the revised paper.
> > > > >
> > > > > Thank you again for your attention and assistance.

---

> > > > > > ### Comment · Reviewer_bMYC · 2024-12-02
> > > > > >
> > > > > > Thanks to the authors for their continued interactions. I'd like to point out that the claim made in the author's previous responses is incorrect: the original finding in Thm. 1 is changed (an additional assumption is added: $c_T \geq \gamma$). This is a relatively minor change, but not simply 'aiming to improve clarity': it is fixing a mistake.
> > > > > >
> > > > > > I think the paper is still somewhat lacking in terms of clarity. However, it has sufficiently improved that I will increase my score to a 5. I feel confident that with minor revisions, this work can be a valuable contribution to the field of causal RL in the future.

---

### Official Review · Reviewer_h5Gh · 2024-11-05

**Soundness:** 3
**Presentation:** 3
**Contribution:** 3
**Rating:** 8
**Confidence:** 2

**Summary:**

The paper considers causal state representations of partially observable environments for approximately solving POMDPs. Specifically, the authors propose an approach to find bisimulation-based causal state models by denoising observations and rewards via an asynchronous diffusion model. The offers also do analysis and give some theoretical guarantees under some assumptions.

**Strengths:**

This is a generally well-written paper based on a novel idea (as far as I can judge). It combines a proposal of an algorithm with theoretical analysis and reasonable, well-designed experiments

**Weaknesses:**

A few points could be clearer in the manuscript, some of which I listed in the questions below. Part of my confusion could be my lack of in-depth knowledge of Causal State Representations.

**Questions:**

Upon initial reading I was unclear what was meant by “POMDPS with perturbed inputs” in the abstract. Isn’t the whole point of any POMDP that its inputs (observations of the environment) are subject to noise?
Line 151: “the action a_{t-1} directly affects the state s_t rather than the observation signal o_t”: Why then is the probability of o_t defined conditional on a_{t-1} in eq. 1a.
Algorithm 1: Why does \zeta denote both the reward noise model and the bisimulation model?

---

> ### Author Response · Authors · 2024-11-22
>
> We express our heartfelt gratitude for your feedback.
>
> * As suggested, we have clarified in the revised version that a POMDP with perturbed inputs stands for a Markov decision process (MDP) with only partially observable, perturbed states (due to partial obstruction and noises/perturbation of the observation). By contrast, standard POMDPs do not require observations to be noisy or perturbed, as discussed in Ref. [1-3].
>
> * Sincere thanks for pointing out a typo in Eq. (1a). As pointed out by the reviewer, the observation $\mathbf{o}_ {t}$ depends solely on the state $\mathbf{s}_ t$ at each time step $t$. In response to this comment, we have corrected Eq. (1a) to "*$\mathbf{o}_ {t} = F\left(\mathbf{s}_ {t}, \mathbf{e}_ {t}\right) \iff P\left(\mathbf{o}_ {t}\mid \mathbf{s}_ {t}\right)$*" in the revised version.
>
> * We apologize for double-defining $\zeta$ to denote two different models in the previously submitted version. In the revised version, we have now used $\phi$ to indicate the reward noise model and $\zeta$ to denote the bisimulation model.
>
> Once again, we sincerely appreciate your approval and valuable comments.
>
> **Reference**
>
> [1]. Barenboim, Moran, and Vadim Indelman. "Online POMDP planning with anytime deterministic guarantees." Advances in Neural Information Processing Systems 36 (2024).
>
> [2]. Lev-Yehudi, Idan, Moran Barenboim, and Vadim Indelman. "Simplifying complex observation models in continuous POMDP planning with probabilistic guarantees and practice." Proceedings of the AAAI Conference on Artificial Intelligence. Vol. 38. No. 18. 2024.
>
> [3]. Singh, Gautam, et al. "Structured world belief for reinforcement learning in POMDP." International Conference on Machine Learning. PMLR, 2021.

---

### Meta-Review · Area_Chair_Dhwy · 2024-12-23

**Metareview:**

This paper proposes an that incorporates the diffusion model in RL algorithms, and learn causal state representation. The paper provides related theoretical guarantees, and include empirical evaluation to justify the advantage of the proposed methods. While being mostly a theoretical paper, most reviewers raised clarity issues, and several reviewers remain not convinced on the validity a few technical details after a few iterations of discussion. We thus recommend rejection on the current form, and suggest authors to improves the clarity and rigor of the paper, and fix typos especially those math critical ones.

**Additional Comments On Reviewer Discussion:**

No strong reason / support for acceptance. Overall rigor / technical solidness is questionable.

---

### Decision · Program_Chairs · 2025-01-22

Reject